# A compendium of genetic regulatory effects across pig tissues

The Farm Animal Genotype-Tissue Expression (FarmGTEx) project has been established to develop a public resource of genetic regulatory variants in livestock, which is essential for linking genetic polymorphisms to variation in phenotypes, helping fundamental biological discovery and exploitation in animal breeding and human biomedicine. Here we show results from the pilot phase of PigGTEx by processing 5,457 RNA-sequencing and 1,602 whole-genome sequencing samples passing quality control from pigs. We build a pig genotype imputation panel and associate millions of genetic variants with five types of transcriptomic phenotypes in 34 tissues. We evaluate tissue specificity of regulatory effects and elucidate molecular mechanisms of their action using multi-omics data. Leveraging this resource, we decipher regulatory mechanisms underlying 207 pig complex phenotypes and demonstrate the similarity of pigs to humans in gene expression and the genetic regulation behind complex phenotypes, supporting the importance of pigs as a human biomedical model.

Genome-wide association studies (GWAS) reveal genomic variants associated with complex phenotypes at an unprecedented speed and scale in both plants[1] and animals[2], but particularly in humans[3,4]. However, most of the variants fall in noncoding regions, putatively contributing to phenotypic variation by regulating gene activity at different biological levels[5,6]. The systematic characterization of genetic regulatory effects on transcriptome (for example, expression quantitative trait loci (eQTLs)) across tissues, as carried out in the Genotype-Tissue Expression (GTEx) project in humans[7], has proven to be a powerful strategy for connecting GWAS loci to gene regulatory mechanisms at large scale[6,8,9].

To sustain food and agriculture production while minimizing associated negative environmental impacts, it is crucial to identify molecular mechanisms that underpin complex traits of economic importance to enable biology-driven selective breeding in farm animals. However, the annotation of regulatory variants in farm animals has so far been limited by small sample size, few tissue/cell type assayed, and in restricted genetic background[10–12]. We thus launched the international Farm Animal GTEx (FarmGTEx) project to build a comprehensive atlas of regulatory variants in domestic animal species. This resource along with the functional annotation of animal genomes project will not only facilitate fundamental biology discovery but also enhance the genetic improvement of farm animals[13].

Pigs are an important agricultural species by supplying meat for humans, and serve as an important biomedical model for studying human development, disease and organ xenotransplantation, due to their similarity to humans in multiple attributes such as anatomical structure, physiology and immunology[14]. Here we report the results of the pilot PigGTEx, which is underpinned by 5,457 RNA-seq data and 1,602 whole-genome sequence (WGS) samples (Supplementary Tables 1 and 2). We test the association of transcriptomic phenotypes with 3,087,268 DNA variants in 34 pig tissues and then evaluate tissue-sharing patterns of regulatory effects. We examine multi-omics data to identify putative molecular mechanisms underlying regulatory variants and then apply this resource to dissect GWAS associations for 268 complex traits. Finally, we leverage the human GTEx resource and GWAS of 136 human complex phenotypes to assess the similarity between pigs and humans in genetic regulation of gene expression and complex phenotypes. We make the PigGTEx resources freely accessible via http://piggtex.farmgtex.org.

## Results

### Data summary

After filtering out the low-quality samples from the initial set of 9,530, we retained 7,095 RNA-seq profiles for downstream analysis (Supplementary Fig. 1 and Supplementary Note). We quantified expression

✉e-mail: albert.tenesa@ed.ac.uk; likui@caas.cn; george.liu@usda.gov; zhezhang@scau.edu.cn; lingzhao.fang@qgg.au.dk

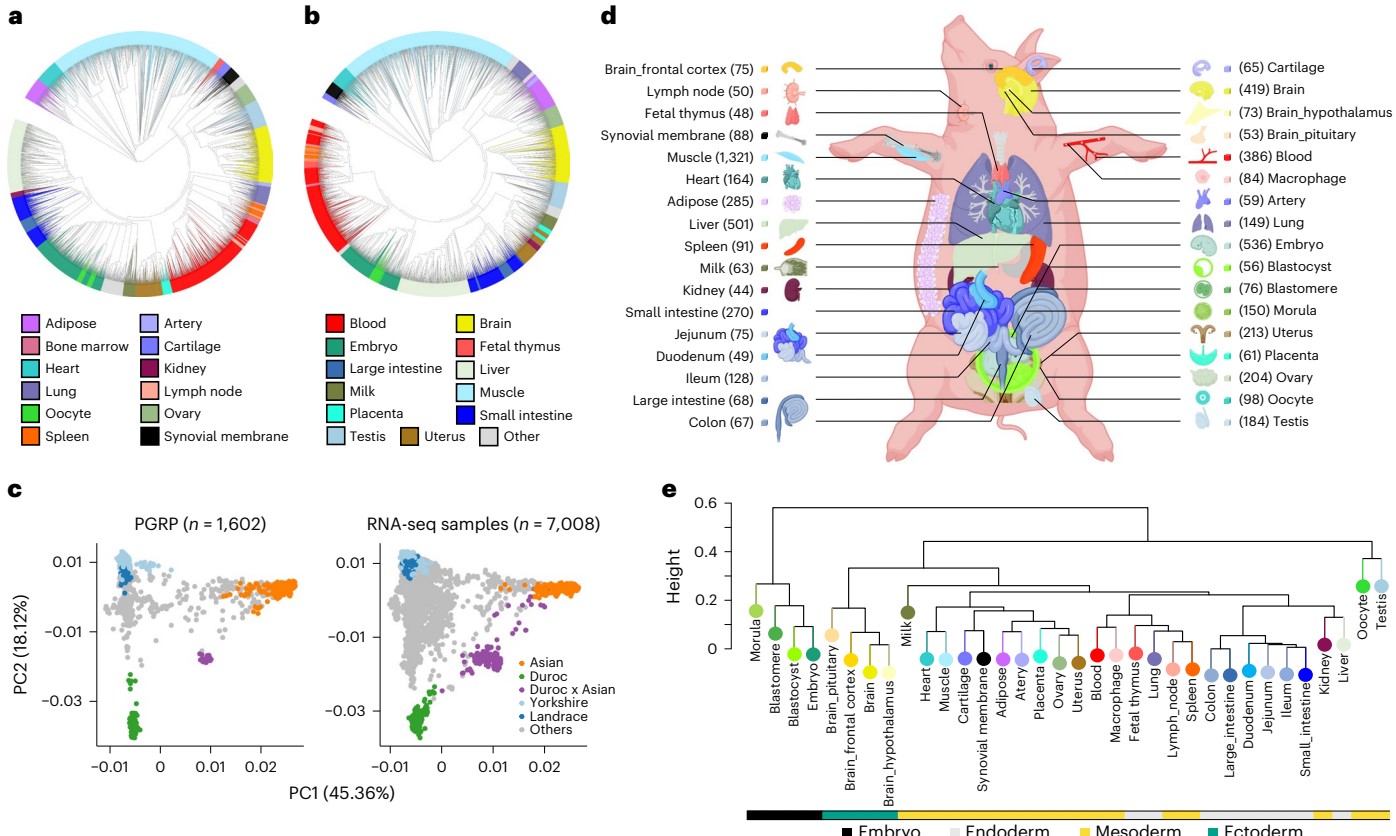

**Fig. 1 | Characteristics of samples in the pilot phase of PigGTEx project.**
**a**, Clustering of 7,095 RNA-seq samples based on the normalized expression ($\log_{10}$-transformed TPM) of 6,500 highly variable genes, defined as the top 20% of genes with the largest s.d. of TPM across samples. **b**, The same sample clustering as **a** but based on normalized alternative splicing values (PSI) of 6,500 highly variable spliced introns, defined as the top 13% of spliced introns with the largest s.d. of PSI across samples. **c**, Principal component analysis of samples based

on 12,207 LD-independent ($r^2 < 0.2$) SNPs. The left panel is for whole-genome sequencing samples ($n = 1,602$) in the PGRP, while the right one is for RNA-seq samples ($n = 7,008$) with successful genotype imputations. **d**, Sample sizes of 34 tissues, cell types and organ systems (all referred to as 'tissues') used for molQTLs mapping. **e**, Clustering of 34 tissues based on the median expression of all 31,871 Ensembl annotated genes (v100) across samples within tissues, representing embryo, endodermal, mesodermal and ectodermal lineages.

levels for protein-coding genes (PCG), lncRNA, exons and enhancers, and alternative splicing events in these samples. Sample clustering based on the five transcriptomic phenotypes recapitulated tissue types well (Fig. 1a,b and Supplementary Fig. 2). We called a median number of 74,347 single-nucleotide polymorphisms (SNPs) from these RNA-seq samples (Extended Data Fig. 1a,b). Leveraging a multibreed pig genomics reference panel (PGRP) consisting of 1,602 WGS samples (Supplementary Fig. 3), we imputed genotypes of RNA-seq samples with an imputation accuracy of 0.94 (concordance rate) and 0.82 (genotype correlation, $r^2$; Extended Data Fig. 1c–n and Supplementary Table 3). The population structure of the RNA-seq samples was similar to the PGRP (Fig. 1c). After removing duplicated RNA-seq samples, we retained 5,457 samples representing 34 tissues, cell types or organ systems (all referred to as 'tissues' hereafter), with at least 40 samples per tissue, for subsequent analysis (Fig. 1d–e, Extended Data Fig. 2a–e and Supplementary Table 4). We further analyzed 270 multi-omics datasets in pigs, including 245 whole-genome bisulfite sequencing (WGBS; Supplementary Figs. 4 and 5 and Supplementary Tables 5–7), 20 single-cell RNA-seq (Supplementary Fig. 6 and Supplementary Table 8) and five Hi-C samples (Supplementary Tables 9 and 10).

**The gene expression atlas empowers functional annotation**
Gene expression was either tissue-specific or ubiquitous (Supplementary Fig. 7a and Extended Data Fig. 3a). We detected between 145 (morula) and 5,180 (frontal cortex) tissue-specific genes across 34 tissues (Extended Data Fig. 3b and Supplementary Fig. 7b).

Tissue-specific genes showed a higher enrichment of active regulatory elements and a higher depletion of repressed polycomb regions in matching tissues than in nonmatching tissues[15] (Extended Data Fig. 3c–e and Supplementary Fig. 7c,d). In addition, tissue-specific genes exhibited distinct patterns of evolutionary DNA sequence constraints across tissues (Supplementary Fig. 7e), in agreement with the hypothesis of tissue-driven evolution[16]. To assign function to pig genes, we performed a gene co-expression analysis in each of the 34 tissues (Supplementary Fig. 8a–c). In total, we detected 5,309 co-expression modules across tissues and assigned 25,023 genes to at least one module (Supplementary Fig. 8d–f and Supplementary Table 11). Among them, 13,266 (42.57%) genes had no functional annotation in the Gene Ontology (GO) database (Extended Data Fig. 3f and Supplementary Fig. 8d); these are referred to as 'unannotated genes' hereafter. For instance, 42 unannotated genes were co-expressed with 59 functional annotated genes in the pituitary, which were substantially enriched in neuron apoptotic processes (Extended Data Fig. 3g). Unannotated genes were less expressed, showed weaker DNA sequence conservation, lower proportion of orthologous genes and higher tissue specificity than genes with functional annotations (Extended Data Fig. 3f). The proportion of expressed unannotated genes varied across tissues, indicating differences in functional annotation between tissues (Extended Data Fig. 3h).

**MolQTL mapping**
In total, 93% of tested genes had significant *cis*-heritability (*cis-h²*; within ±1 Mb of transcription start sites (TSS)) estimates in at least one

tissue while accounting for hidden factors (Extended Data Fig. 2f–h and Extended Data Fig. 4a,b). We mapped molecular quantitative trait loci (molQTLs) for five molecular phenotypes, including *cis*-eQTL for PCG expression, *cis*-eeQTL for exon expression, *cis*-lncQTL for lncRNA expression, *cis*-enQTL for enhancer expression and *cis*-sQTL for alternative splicing. In total, 86%, 67%, 46%, 27% and 64% of all tested PCGs (n = 17,431), lncRNAs (n = 7,374), exons (n = 82,678), enhancers (n = 3,353) and genes with alternative splicing events (n = 18,331) had at least one significant variant (eVariant) detected in at least one tissue; hence, they were defined as eGenes, eLncRNAs, eExons, eEnhancers and sGenes, respectively (Supplementary Fig. 9 and Supplementary Table 12). The proportion of eGenes detected was positively correlated with sample size across tissues, similar to the other four molecular phenotypes (Fig. 2a, Extended Data Fig. 4c and Supplementary Fig. 10). The top *cis*-e/sQTL centered around TSS of genes (Supplementary Fig. 11a–e). Tissues with a larger sample size yielded a larger proportion of *cis*-eQTL with smaller effects (Supplementary Fig. 11f–g). PCG had the highest proportion of detected eGenes across tissues, followed by lncRNA, enhancer, splicing and finally exon (Fig. 2b). Notably, molecular phenotypes exhibited a high proportion (an average of 70%) of their own specific molQTL after taking linkage disequilibrium (LD) between SNPs into account (Fig. 2b), indicative of their distinct underlying genetic regulation. On average, 20% of eGenes, 13.5% of sGenes, 21.2% of eExons, 23.5% of eLncRNAs and 21% of eEnhancers had more than one independent eVariant across tissues, and the proportion increased with an increasing sample size of tissues (Fig. 2c and Extended Data Fig. 5a). Down-sampling analysis in three major tissues further confirmed an impact of sample size on the statistical power for *cis*-eQTL discovery (Fig. 2d). Approximately half of the independent *cis*-eQTL were located within ±182 kb of TSS, and those with larger effect size were closer to TSS (Extended Data Fig. 5b–d). The eGenes with more independent *cis*-eQTL have a higher *cis-h²*, but no significant differences for the median gene expression level (Fig. 2e).

We applied four distinct strategies to validate the *cis*-eQTL. First, the summary statistics of *cis*-eQTL derived from the linear regression model[17] had a strong correlation with those from a linear mixed model (Extended Data Fig. 6a–e). Second, the internal validation yielded a high replication rate (measured by $\pi_1$) of *cis*-eQTL, with an average $\pi_1$ value of 0.92 (range: 0.80–1.00) and an average of 0.56 (range 0.36–0.89) for Pearson's *r* between effect sizes across tissues (Fig. 2f). Third, 92%, 74%, 73% and 69% of *cis*-eQTL in blood, liver, duodenum and muscle, respectively, were replicated in independent datasets (Extended Data Fig. 6f–h). Fourth, effects derived from allele-specific expression (ASE) analysis were correlated with those from *cis*-eQTL mapping (Fig. 2g and Extended Data Fig. 6i–k). In addition, we conducted an exploratory analysis of *trans*-eQTL in 12 tissues with over 150 individuals and detected an average of 80 *trans*-eGenes (false discovery rate, FDR < 0.05) across tissues (Supplementary Fig. 12a,b). We took the muscle that had the largest sample size (n = 1,321) as an example to conduct an internal validation of *trans*-eQTL by randomly and evenly dividing samples into two groups. We observed that the replication rate ($\pi_1$) between the two groups was 0.4 and the Pearson's correlation of effect sizes of significant *trans*-eQTL between groups was 0.5 (Supplementary Fig. 12c).

To understand how *cis*-eQTL are shared across pig breeds, we considered muscle as an example. We divided muscle samples into eight breed groups (all referred to as 'breeds' hereafter) and performed *cis*-eQTL mapping separately (Extended Data Fig. 7a and Supplementary Table 13). Across all eight breeds, we detected 9,548 unique *cis*-eGenes, of which 97.1% could be replicated in at least two of these breeds (Fig. 2h and Extended Data Fig. 7b,c). The replication rates were higher in breeds with more samples (Extended Data Fig. 7d). For instance, the Landrace × Yorkshire cross-breed had the largest sample size (n = 374) replicated on average 95.6% of the *cis*-eQTL detected in

the other seven breeds (Extended Data Fig. 7d). The *cis*-eQTL effects were positively correlated between breeds and clearly separated from other tissues (Fig. 2i and Extended Data Fig. 7e). In addition, the effects of *cis*-eQTL from the multibreed meta-analysis were correlated with those from the combined muscle population (Extended Data Fig. 7f). Compared to the single-breed meta-analysis, the combined population detected 86.2% more *cis*-eQTL (Extended Data Fig. 7g). To explore whether breed interacts with genotype to modulate expression of some genes, we conducted breed-interaction *cis*-eQTL (bieQTL) mapping. In total, 589 genes had at least one significant bieQTL in 13 tissues (Fig. 2j,k, Extended Data Fig. 7h,i and Supplementary Table 14). Furthermore, we conducted a cell-type deconvolution analysis in seven tissues, demonstrating the variation of cell-type composition across bulk tissue samples (Extended Data Fig. 8a). A total of 376 genes had at least one significant cell-type interaction *cis*-eQTL (cieQTL) in three tissues (Fig. 2l–m, Extended Data Fig. 8b,c and Supplementary Table 14). In addition, we validated half of bieQTL and cieQTL with the ASE approach[18] (Fig. 2j,l and Extended Data Fig. 8d–g).

## Tissue-sharing patterns of molQTL

Tissues with similar functions clustered together, and the tissue relationship was consistent across all ten data types, including the five types of molQTL and the respective molecular phenotypes (Fig. 3a,b and Extended Data Fig. 9a,d). The most easily accessible samples, that is, blood and milk cells, showed an average correlation of 0.51 *cis*-eQTL effects with other tissues. Both had the highest similarity to immune tissues, followed by intestinal tissues, and finally testis and embryonic tissues. The overall tissue-sharing of molQTL showed a U-shaped curve (Fig. 3c). Among them, *cis*-eQTL of PCG had the highest degree of tissue-sharing, followed by *cis*-lncQTL, *cis*-sQTL, *cis*-eeQTL and finally *cis*-enQTL (Fig. 3c and Extended Data Fig. 9e). An eGene tended to be regulated by *cis*-eQTL of smaller effect if it showed a higher level of tissue-sharing or was expressed in more tissues (Fig. 3d and Extended Data Fig. 9f). The higher the tissue-sharing of eGenes, the larger the minor allele frequency (MAF) of their *cis*-eQTL, and the closer the distance of their *cis*-eQTL to TSS (Fig. 3d). In addition, eGenes that were active in more tissues had a decreased PhastCons score (that is, less sequence constraint), while genes that were not eGenes (non-eGenes) in more tissues had an increased PhastCons score (Fig. 3e). The shared non-eGenes in the 34 tissues were substantially enriched in fundamental biological processes (Supplementary Table 15). We summarized four types of SNP–gene pairs and observed that 1.8% (1,166/64,250) of top *cis*-eQTL of the same eGenes had an opposite effect in at least one tissue pair, representing 3.1% (467/14,988) of all detected eGenes (Fig. 3f). Compared to other tissue pairs, blood and testis showed the highest proportion (25%) of eGenes with opposite *cis*-eQTL effects (Fig. 3g). For example, *ODF2L*, which showed the opposite direction of eQTL effect (rs329043485) between blood and testis (Fig. 3h and Extended Data Fig. 9g–h), is involved in negative regulation of cilium assembly and spermatogenesis[19].

## Functional annotation of molQTL

Compared to other molQTL, *cis*-sQTL had a higher enrichment for missense variants, variants with a high impact on protein sequence and variants in splice region and acceptor sites (Fig. 4a and Supplementary Fig. 13a). Although there was a significant enrichment of molQTL in exonic annotations (for example, synonymous and missense), the proportion of such variants over all the molQTL was around 5.4%, that is, 5.4% for eQTL, 5.5% for sQTL, 5.2% for eeQTL, 5.4% for lncQTL and 5.8% for enQTL. This finding was consistent with human GTEx[7,20] and RatGTEx[21]. Looking at chromatin states, these five types of molQTL showed the highest enrichment in active promoters, followed by those proximal to TSS and ATAC islands (Fig. 4b and Supplementary Fig. 13b). The molQTL with higher causality scores showed a higher enrichment in functional features (Supplementary Fig. 13c,d). Among

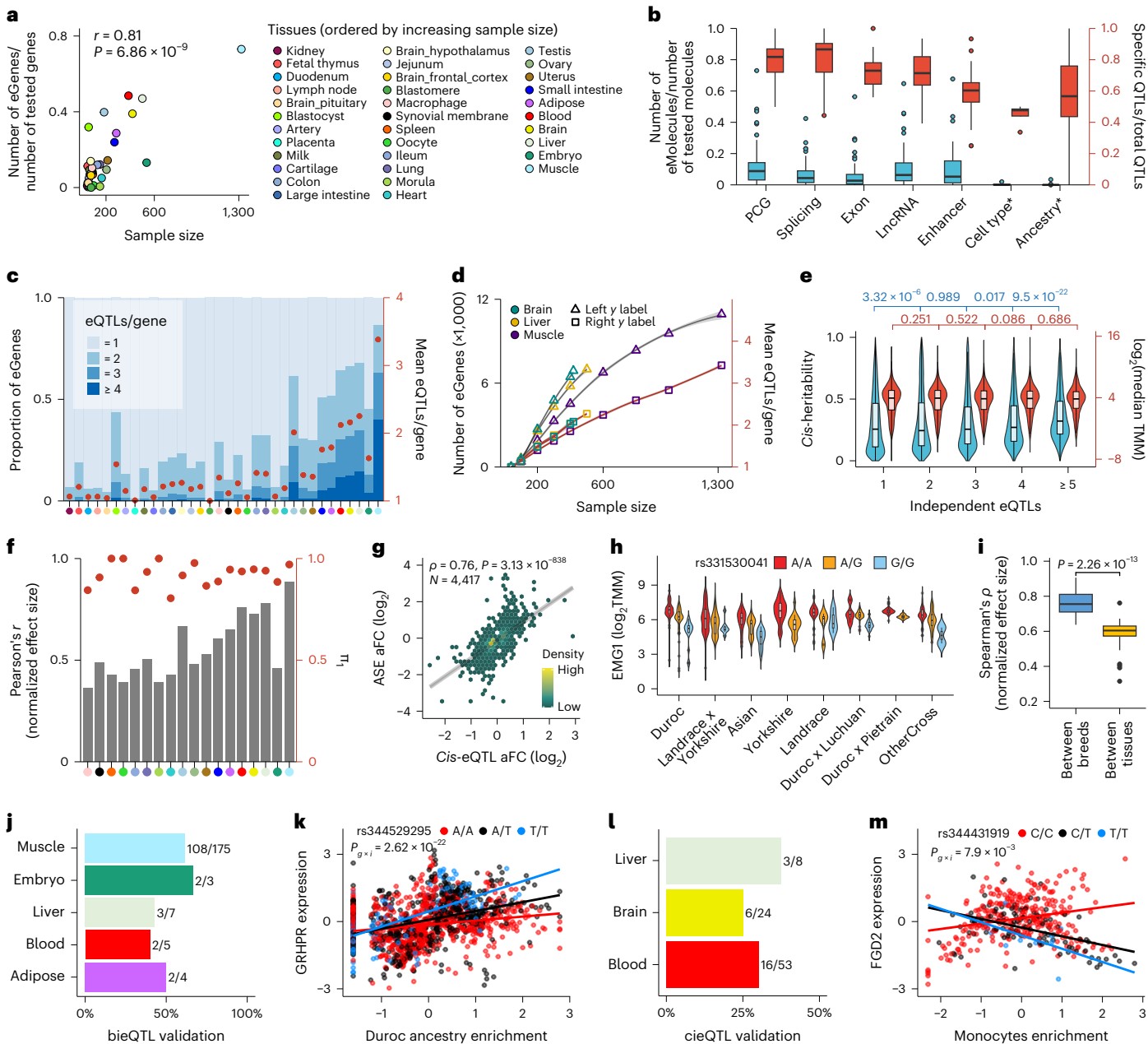

**Fig. 2 | molQTL discovery. a**, Pearson's *r* between the proportion of detectable eGenes and sample size across 34 tissues. **b**, Proportions of detectable eMolecule (blue) and specific molQTL (red) for different molecular phenotypes in 34 tissues. * indicates the interaction of *cis*-eQTLs (ieQTL). Cell type* and Ancestry* are for cell-type ieQTL (cieQTL) and breed/ancestry ieQTLs (bieQTL), respectively. **c**, Distribution and the average number of independent *cis*-eQTL per gene. Tissues (*x* axis) are ordered by increasing sample size. The color key is the same as in **a**. **d**, Number of eGenes (triangle) and average number of independent *cis*-eQTL (square). **e**, The comparison of *cis*-$h^2$ (blue) and median expression levels (red) of genes with different numbers of detectable independent *cis*-eQTL across tissues. The top labels show nominal *P* values (uncorrected for multiple testing) from one-sided Student's *t* tests. **f**, Internal validation of *cis*-eQTL. Bars represent Pearson's *r* of the normalized effects of *cis*-eQTL between validation and discovery groups. Points represent the $\pi_1$ statistic measuring the replication rate of *cis*-eQTL. **g**, Spearman's $\rho$ of effect sizes (aFC in log₂ scale) between

*cis*-eQTL and ASE at matched loci (*n* = 4,417) in muscle. **h**, A *cis*-eQTL (rs331530041) of *EMG1* in muscle is shared across eight ancestry groups. **i**, Spearman's correlation of the *cis*-eQTL effects between eight breeds of the muscle (left) and between muscle and other 33 tissues (right). The *P* value is obtained from a two-sided Wilcoxon rank-sum test. **j**, Proportion of bieQTL that are validated with the ASE approach. The number of validated bieQTLs out of the total number of bieQTLs tested is shown to the right of each bar. **k**, Effect of eVariant (rs344529295) of *GRHPR* interacted with the Duroc ancestry enrichment in muscle. The two-sided *P* value is calculated by the linear regression bieQTL model. The lines are fitted by a linear regression model using the geom_smooth function from ggplot2 (v3.3.2) in R (v4.0.2). **l**, Proportion of cieQTL that are validated by the ASE approach. **m**, Effect of eVariant (rs344431919) of *FGD2* interacted with monocyte enrichment in blood. The two-sided *P* value is calculated by the linear regression cieQTL model. The lines are fitted using the same method as in **k**. aFC, allelic fold change.

all the five types of molQTL, *cis*-enQTL with high causality scores had the highest enrichment for enhancer-like chromatin states (Supplementary Fig. 13d). An average of 64% of *cis*-eQTL could potentially modify transcription factor binding sites (Supplementary Table 16). Although they

showed a weak enrichment for molQTL (except for *cis*-enQTL; Fig. 4b), enhancers had a higher enrichment for *cis*-eQTL in the matching tissue compared to nonmatching tissues (Fig. 4c). Notably, the top *cis*-eQTL tended to be enriched in promoters rather than enhancers, whereas

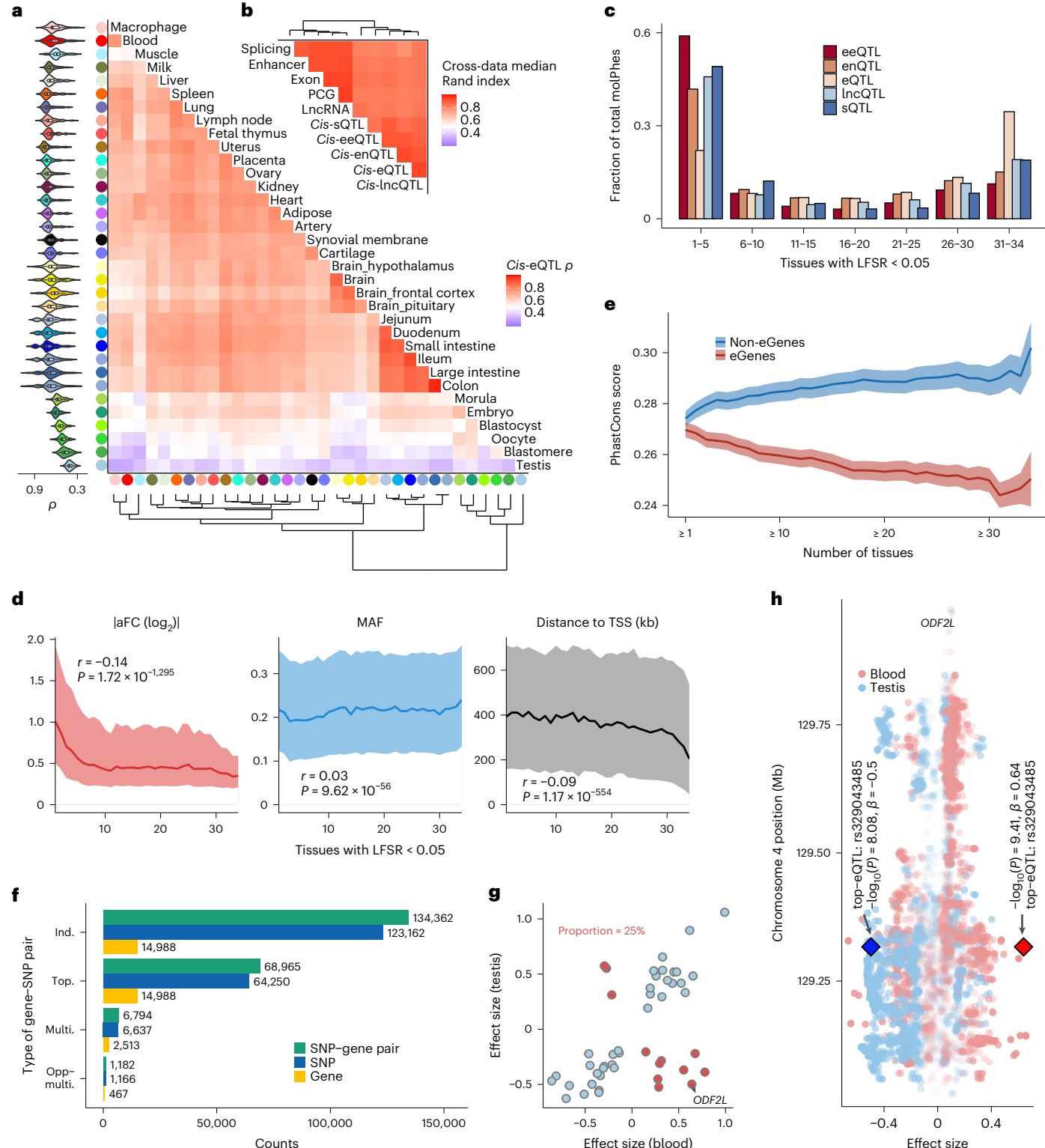

**Fig. 3 | Tissue-sharing pattern of regulatory effects. a**, Heatmap of tissues depicting the corresponding pairwise Spearman's correlation ($\rho$) of *cis*-eQTL effect sizes. Tissues are grouped by hierarchical clustering (bottom). Violin plots (left) represent Spearman's $\rho$ between the target tissue and other tissues. **b**, Similarity (measured by the median pairwise Rand index) of tissue-clustering patterns across ten data types. **c**, The overall tissue-sharing pattern of five molQTL types at LFSR < 5% obtained by MashR (v0.2-6). **d**, Relationships between the magnitude of tissue-sharing of *cis*-eQTL and their effect sizes (aFC, left), MAFs (middle) and distances to the TSS (right). The *P* values are obtained by Pearson's correlation (*r*) test. The line and shading indicate the median and interquartile range, respectively. **e**, Conservation of DNA sequence (measured by the PhastCons score

of 100 vertebrate genomes) of eGenes and non-eGenes regarding tissue-sharing. The line and shading indicate the mean and standard error, respectively. **f**, Counts of four types of SNP–gene pairs across 34 tissues. Ind., independent *cis*-eQTL; top., top *cis*-eQTL; multi., eGenes have identical or high LD ($r^2 > 0.8$) *cis*-eQTL in any two tissues; opp-multi., eGenes have an opposite direction of *cis*-eQTL effect between any two tissues. **g**, Scatter plots of *cis*-eQTL effect sizes of 48 common multi-eGenes in blood and testis. *cis*-eQTL with the same directional effect are colored blue (*n* = 36), and those with the opposite direction are colored red (*n* = 12). **h**, The *cis*-eQTL effects of *ODF2L* on chromosome 4 in blood and testis. Diamond symbols represent the top *cis*-eQTL of *ODF2L*. The two-sided *P* value is calculated by the linear regression *cis*-eQTL model.

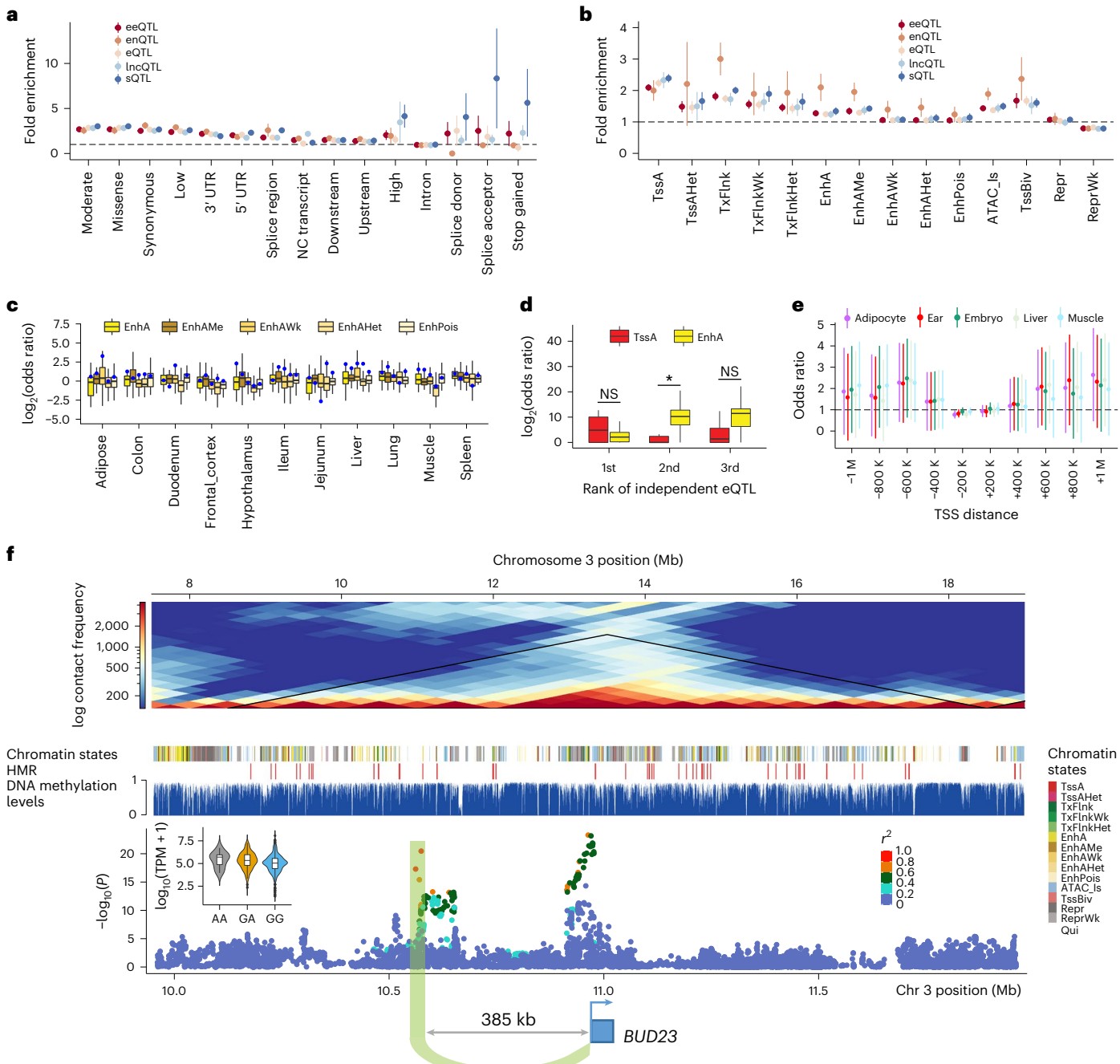

**Fig. 4 | Functional characterization of regulatory variants. a,b,** Fold enrichment (mean ± s.d.) for fine-mapped molQTLs in sequence ontologies (**a**) and 14 chromatin states[15] (**b**). **c,** Enrichment of *cis*-eQTL in five types of enhancers. Each box includes enrichment of *cis*-eQTL from 34 tissues across enhancers. Blue dots represent enrichments from matching tissues. **d,** Enrichment of top three independent *cis*-eQTL in two chromatin states. TssA is for active TSS, while EnhA is for active enhancers. The *P* values are obtained by the two-sided Student *t* test. *\*P* < 0.05 and NS indicates not significant. **e,** Enrichment (mean ± s.d.) of *cis*-eQTL within the same topologically associating domain of TSS of target genes. TADs

are obtained from Hi-C data of five tissues. The *cis*-eQTL are grouped according to their distance to TSS. – and + means upstream and downstream, respectively. **f,** The landscape of *BUD23* at multiple genomic features in muscle. The top plot shows that *BUD23* and its second independent eVariant (rs790620973) are located within a TAD (the black triangle). The bottom is the Manhattan plot showing *cis*-eQTL results of *BUD23*. The violin plot shows the expression levels (log₁₀-transformed TPM) of *BUD23* across three genotypes (AA, *n* = 9; GA, *n* = 131; GG, *n* = 1,181) of this eVariant in muscle. The two-sided *P* value is obtained from the linear regression *cis*-eQTL model.

the reverse was observed for the second- and third-ranked *cis*-eQTL (Fig. 4d). In addition, molQTL showed tissue-specific enrichment for hypomethylated regions (HMRs) and allele-specific methylation loci (Supplementary Fig. 13e). In muscle, 2,016 *cis*-eQTL, 4,694 *cis*-eeQTL, 524 *cis*-lncQTL, 5,174 *cis*-enQTL and 1,590 *cis*-sQTL were mediated by methylation QTL (Supplementary Fig. 13f,g and Supplementary Table 17). The long-distance *cis*-eQTL were substantially enriched in

the same topologically associating domain (TAD) as TSS of target genes after accounting for the *cis*-eQTL-TSS distance (Fig. 4e). This suggests that long-range *cis*-eQTL may affect gene expression by mediating 3D genome interactions[22]. For instance, in muscle, the second independent *cis*-eQTL of *BUD23* was 385 kb upstream of its TSS, and located within the same TAD of the TSS, as well as was surrounded by HMRs and enhancers (Fig. 4f).

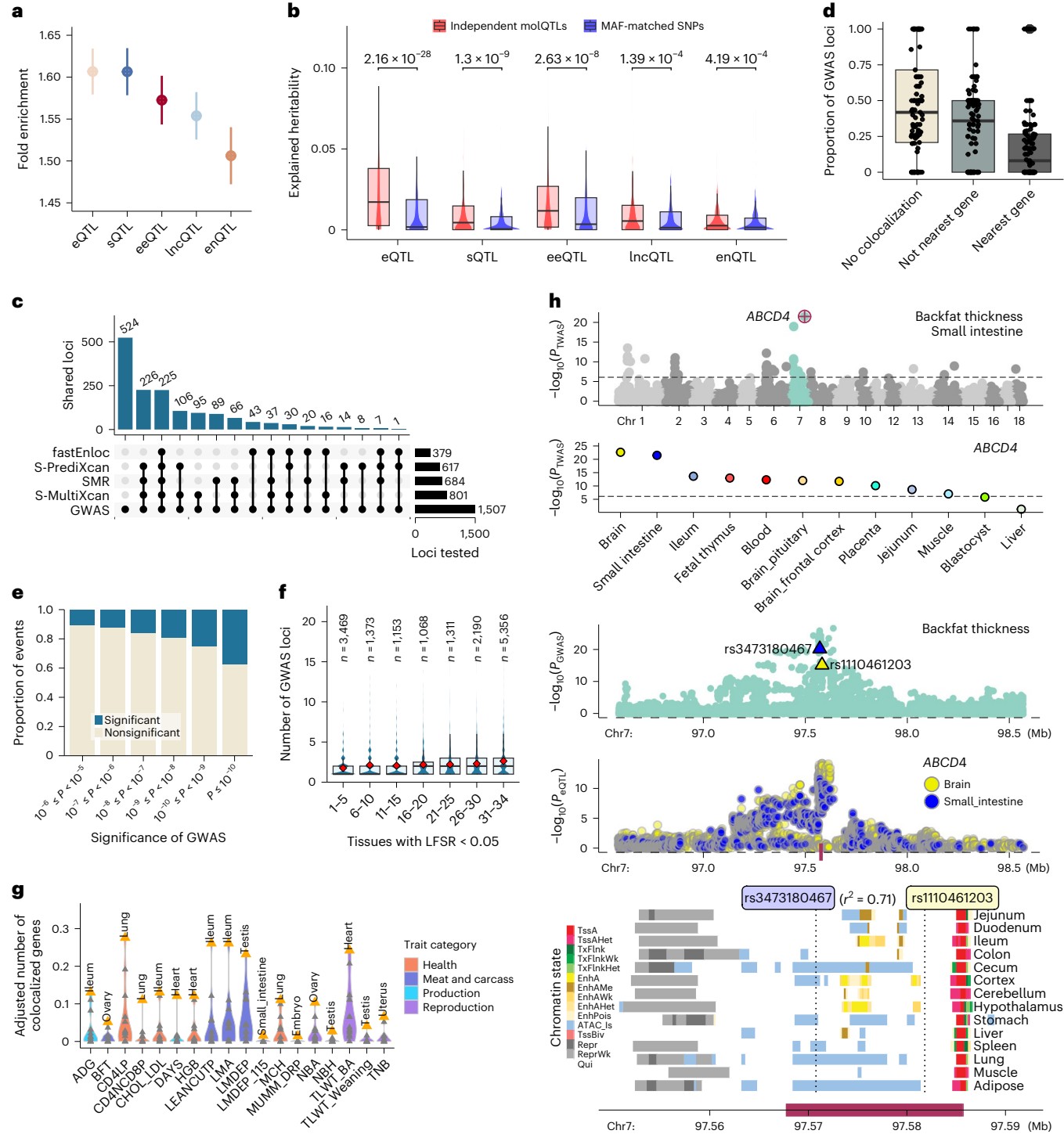

**Fig. 5 | Interpreting GWAS loci of complex traits using molQTL. a,** Enrichment (mean and 95% confidence interval) of GWAS variants with five types of molQTL in 34 tissues. **b,** Heritability of 16 complex traits of pig explained by independent molQTLs and those MAF-matched SNPs across 34 tissues. The top numerical labels are the nominal *P* values (uncorrected for multiple testing) based on the two-sided paired Student's *t* test. **c,** Number of GWAS loci linked to eGenes through fastEnloc, SMR, S-PrediXcan and S-MultiXcan. The bottom point-line combinations of the upset plot represent the intersections of GWAS loci linked to eGenes by different methods. **d,** Proportion of three types of GWAS loci regarding the colocalization results, where 105 GWAS traits are shown in each category. No colocalization, GWAS loci that are not colocalized with any eGenes in 34 tissues. Not nearest gene, GWAS loci whose colocalized eGenes are not nearest genes to GWAS lead SNPs. Nearest gene, GWAS loci whose colocalized eGenes are the nearest ones. Each dot represents a complex trait. **e,** Proportion of significant

colocalizations of GWAS loci with *cis*-eQTL at various significance levels of GWAS. **f,** The number of colocalized GWAS loci per eGene across 105 traits above. eGenes are classified into seven groups regarding the tissue-sharing pattern. Diamond indicates the mean value. **g,** The number of colocalized genes adjusted for tissue sample size and eGene discovery ratio in 14 tissues across 18 GWAS traits (detailed abbreviations in Supplementary Table 18). Top tissues are labeled. **h,** The association of *ABCD4* with the average BFT. The top Manhattan plot represents the TWAS results of BFT in the small intestine, followed by the TWAS results of *ABCD4* for BFT in 12 tissues being tested. The two following Manhattan plots show the colocalization of BFT GWAS (top) and *cis*-eQTL (bottom) of *ABCD4* on chromosome 7 (chr 7) in both the brain and small intestine. The blue and yellow triangles indicate the top variants of *ABCD4* in the small intestine (rs3473180467) and brain (rs1110461203), respectively. These two variants are in high LD ($r^2$ = 0.71). The bottom panel is for chromatin states around *ABCD4*.

## Interpreting GWAS loci with molQTL

To study the regulatory mechanisms underlying complex traits in pigs, we examined 268 GWAS summary statistics of 207 complex traits (Supplementary Table 18) and found that GWAS signals were enriched in molQTL (Fig. 5a and Supplementary Fig. 14a–e). Among them, *cis*-eQTL/*cis*-sQTL showed the highest enrichment (-1.61-fold, s.e. = 0.014), followed by *cis*-eeQTL (1.57-fold, s.e. = 0.015), *cis*-lncQTL (1.55-fold, s.e. = 0.014) and *cis*-enQTL (1.51-fold, s.e. = 0.017; Fig. 5a and Supplementary Fig. 14f). Averaging across 198 traits, approximately half of the heritability was mediated by PCG expression and alternative splicing, followed by exon expression (46.4%), enhancer expression (29.5%) and lncRNA expression (28.5%; Supplementary Fig. 14g). The amounts of heritability of complex traits explained by molQTL were higher than those explained by MAF-matched random SNPs (Fig. 5b and Supplementary Fig. 14h).

Furthermore, we employed four complementary approaches to detect shared regulatory variants/genes associated with both molecular phenotypes and complex traits, including colocalization via fastENLOC[23], Mendelian randomization via SMR[24], single-tissue transcriptome-wide association studies (TWAS) via S-PrediXcan[25] and multi-tissue TWAS via S-MultiXcan[26]. Of 1,507 significant GWAS loci that were tested in the *cis*-eQTL mapping, 983 (65%) were interpreted with *cis*-eQTL in at least one tissue (Fig. 5c and Supplementary Table 19). Among them, only 33% were colocalized with the nearest genes of the lead GWAS SNP (Fig. 5d). GWAS loci mapped with higher significance levels were more likely to be colocalized with *cis*-eQTL (Fig. 5e). The eGenes shared by more tissues tended to be colocalized with more GWAS loci (Fig. 5f). The number of colocalization events of a trait was determined by the statistical power of both GWAS and *cis*-eQTL mapping (Supplementary Fig. 14i–o).

To prioritize tissues relevant for complex trait variation, we defined a 'tissue relevance score' through the number of colocalization events adjusted by sample size and eGene discovery ratio of a tissue (Supplementary Table 20). We only considered 14 tissues with over 100 samples and found that, for instance, the ileum was the most relevant tissue for both average daily gain (ADG) and loin muscle area (Fig. 5g). For instance, *ABCD4* was the top associated gene in the small intestine TWAS of the average backfat thickness (BFT; Fig. 5h). It also had a significant association with BFT in the brain. The GWAS loci of BFT were colocalized with *cis*-eQTL of *ABCD4* in both the brain and small intestine. Although these lead SNPs were different in these two tissues, they had a relatively high LD ($r^2 = 0.71$), potentially tagging the same underlying causal variant. The fine-mapped SNP (rs1114012229) of the BFT GWAS was in a high LD ($r^2 = 0.85$) with the fine-mapped SNP (rs1107405934) of the *ABCD4* eQTL (Supplementary Fig. 15a). In addition, rs1107405934 was specifically associated with the expression of *ABCD4* in both intestinal tissues and the brain (Supplementary Fig. 15b, c).

Furthermore, we employed the same GWAS integrative analysis for other molQTL (Supplementary Tables 21–24). Around

80% (1,204/1,507) of significant GWAS loci could be explained by at least one molQTL in the 34 tissues. Of note, 8.2%, 3.8%, 3.5%, 1.9% and 0.4% of all 1,507 GWAS loci were only explained by *cis*-eQTL, *cis*-sQTL, *cis*-eeQTL, *cis*-lncQTL and *cis*-enQTL, respectively (Extended Data Fig. 10a,b). For example, a GWAS signal of ADG on chromosome 13 was only colocalized with *cis*-eQTL of *CFAP298-TCP10L* in the colon, but not with its *cis*-sQTL or *cis*-eeQTL (Extended Data Fig. 10c). The GWAS signal for BFT on chromosome 15 was exclusively colocalized with *cis*-sQTL of *MYO7B* in small intestine, while the GWAS signal of litter weight was exclusively colocalized with *cis*-eeQTL of *FBXL12* in uterus (Extended Data Fig. 10d–e). In addition, 63% of GWAS loci were colocalized with more than one type of molQTL (Extended Data Fig. 10a and Supplementary Fig. 16). In addition, we detected 512 lncRNA-PCG-trait trios with significant pleiotropic associations (Supplementary Table 25 and Extended Data Fig. 10f).

## The shared genetic regulation between humans and pigs

By examining GTEx (v8) in humans[7], we found that one-to-one orthologous genes ($n = 15,944$) contributed to an average of 82% and 87% of overall expression across 17 common tissues in pigs and humans, respectively (Supplementary Fig. 17a,b). The visualization of variation in gene expression among all 12,453 samples clearly recapitulated tissue types rather than species (Supplementary Fig. 17c–h). The number of tissues in which an eGene was active was correlated between species (Supplementary Fig. 17i). The eGenes in a pig tissue generally had a higher enrichment for eGenes in the matching tissue in humans compared to other tissues (Fig. 6a). Furthermore, we observed a significant correlation ($r = 0.56$) of averaged eQTL effect between humans and pigs (Fig. 6b), which was higher than that ($r = 0.24$) observed between humans and rats previously[21]. In general, matching tissues had a higher correlation of eQTL effect compared to nonmatching tissues (Supplementary Fig. 18a,b and Supplementary Table 26). We observed a significant but weak correlation ($r = 0.09$) of *cis*-$h^2$ between humans and pigs (Supplementary Fig. 18c), similar to that between humans and rats ($r = 0.10$)[21]. In addition, tissue-specific expression of genes was more similar between pigs and humans than that between cattle and humans (Supplementary Fig. 19a–c). Similarly, the eQTL effects of orthologous genes in pigs were more correlated with those in humans than with those in cattle (Supplementary Fig. 19d–f).

We divided orthologous genes into four groups (that is, 'neither', 'human-specific', 'pig-specific' and 'shared') in each of the 17 matching tissues and observed a significant difference in expression levels among them. The shared eGenes had a lower tissue specificity in expression levels and regulatory effects, compared to genes in the other three groups (Fig. 6c and Supplementary Fig. 18d). A total of 783 eGenes were active in all tissues in both species, which were substantially enriched in metabolic processes (Supplementary Table 27). A total of

**Fig. 6 | Conservation of gene expression, *cis*-eQTL and complex trait genetics between pigs and humans. a**, Enrichment (Fisher's exact test) of pig eGenes with human eGenes across 17 matching tissues. Red triangles: matching tissues. **b**, Pearson's correlation of eQTL effect size in orthologous genes ($n = 15,944$) between pigs and humans. **c**, Expression levels, TAU values and tissue-sharing levels for four groups of orthologous genes across 17 tissues in pigs. Neither, 3,993 non-eGenes in both species; human-specific, 8,174 eGenes; pig-specific, 3,882 eGenes; shared, 10,574 eGenes in both species. Two-sided Wilcoxon rank-sum test, ***$P < 0.001$. Diamond, median; error bar, upper/lower quartiles. **d**, LOEUF in the four groups of orthologous genes in ten evenly spaced expression level bins. One-sided Wilcoxon rank-sum test, NS $P > 0.05$, *$P < 0.05$, **$P < 0.01$ and ***$P < 0.001$. The diamond and error bar are the same as in **c**. **e**, Significance ($-\log_{10}(P)$) of Pearson's r of orthologous gene effect size between pig ($n = 268$) and human ($n = 136$) traits derived from TWAS. Each bar represents a pig–human trait pair in the same tissue ($n = 11$) and the within-domain blocks

of color correspond to different human traits. The number of tested genes for each of the pairs is shown in Supplementary Table 30. The text in the middle of the circle represents the significant examples of pig–human trait pairs in different thresholds. For each example, it includes human trait (top), pig trait (bottom) and TWAS tissue (left). $P_{cutoff 1}$: FDR < 10% across all tested combinations. $P_{cutoff 2}$: Bonferroni-corrected $P < 5\%$ within each trait–tissue pair of humans. **f**, Differences in the number of significant genes (FDR < 5%) from cross-species (pig and human) meta-TWAS, compared to those from human TWAS. Supplementary Tables 18 and 29 present a detailed description of pig traits and human traits, respectively. **g**, FDR of discovered genes in human TWAS (RawTWAS) and cross-species meta-TWAS in the brain for BFT (pig) and weight (human). **h**, Pearson's r between TWAS significances (color bar) of genes in pig BFT and their heritability enrichments (mean ± s.e.) in human weight. The orthologous genes were divided into ten evenly spaced bins by sorting the $P$ values of TWAS in the brain of pig BFT. Shading: standard error of the fitting line.

194 genes were not eGenes in any tissues in both species, and these were substantially enriched in essential biological functions (Supplementary Table 28). Expression levels of genes were negatively correlated with LOEUF scores, which was consistent across the four groups of

genes (Supplementary Fig. 18e). Among them, 'Shared' eGenes had the weakest negative correlation of expression levels and LOEUF scores, while 'neither' eGenes had the strongest negative correlation (Supplementary Fig. 18e). Of specific note, although they had the

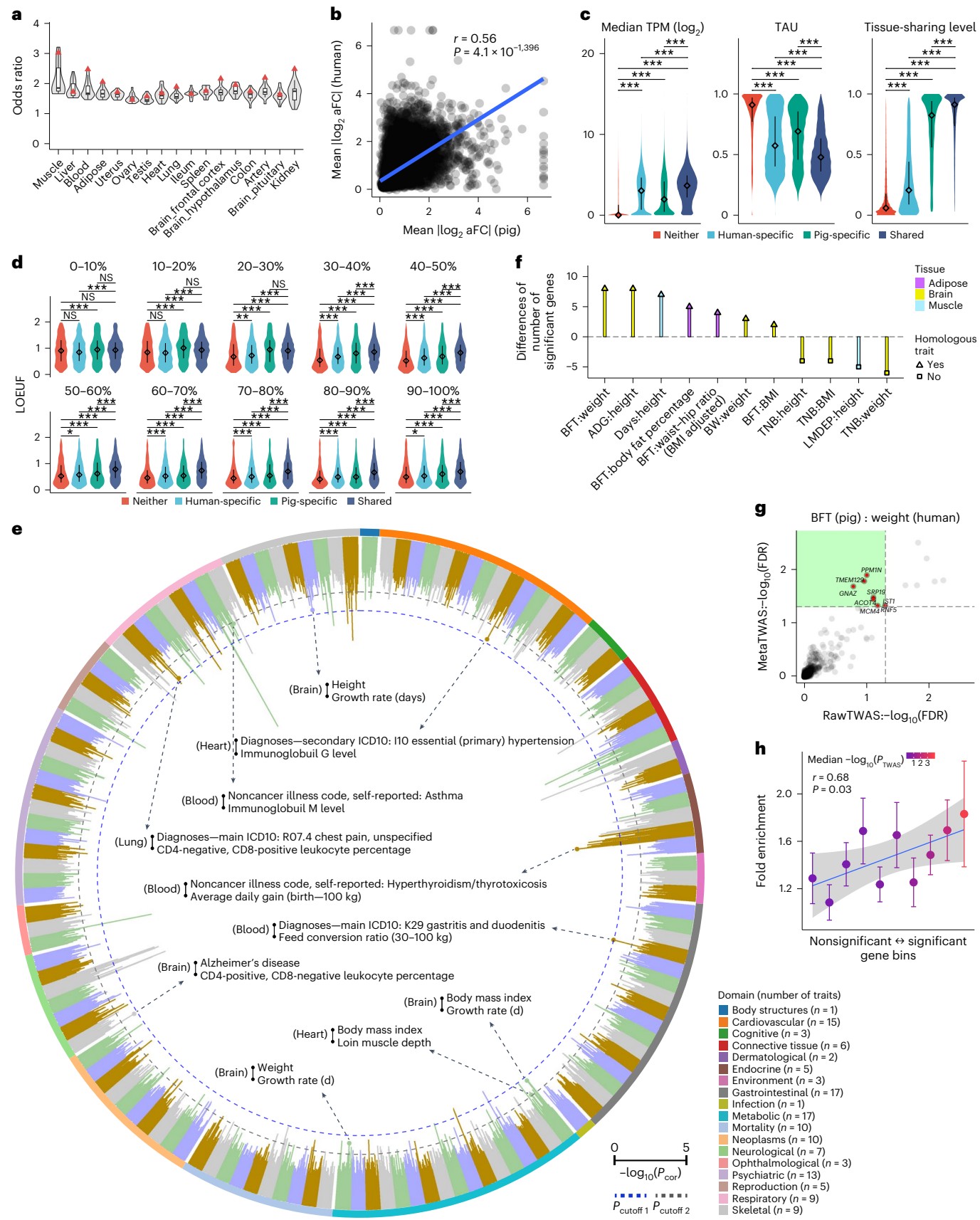

highest expression levels, 'Shared' eGenes showed the strongest tolerance to loss of function mutations among the four gene groups (Fig. 6d). Compared to other genes, eGenes shared in both species had the lowest evolutionary DNA sequence constraints, whereas shared non-eGenes showed the opposite trend (Supplementary Fig. 18f). The expression levels of most genes were weakly or even not correlated with their PhastCons scores, eQTL detection and $cis$-$h^2$ estimates across tissues (Supplementary Fig. 18g–i).

To investigate whether the regulatory mechanism of complex phenotypes was conserved between humans and pigs, we compared the effect sizes of orthologous genes between 268 pig and 136 human complex phenotypes based on the summary statistics of TWAS (Supplementary Table 29). We observed a clear deviation (Wilcoxon rank-sum test $P = 2.16 \times 10^{-62}$) of the observed $P$ values of TWAS correlations from the permutation-based null distribution (Supplementary Fig. 20a), and a total of 89 pig–human trait pairs were significant (FDR < 0.1; Supplementary Table 30, Fig. 6e and Supplementary Fig. 20b–e). We then chose several well-recognized homologous trait pairs between humans and pigs to perform the meta-TWAS, with several nonhomologous trait pairs as negative controls. For homologous trait pairs, cross-species meta-TWAS improved the discovery of trait-associated genes in humans (Fig. 6f). For instance, cross-species meta-TWAS analysis of pig average BFT and human body weight (BW) revealed eight new genes (FDR < 0.05) associated with BW in humans (Fig. 6g). Based on GWAS of 3,302 traits in humans[27], phenome-wide association studies (PheWAS) showed that five of these eight genes were associated with other BW-relevant traits, such as height, birth weight and BMI (Supplementary Table 31). Furthermore, gene groups with higher significance in the pig BFT TWAS showed a higher enrichment for heritability of human BW (Fig. 6h).

## Discussion

The pilot PigGTEx offers a deep survey of genetic regulatory effects across a wide range of tissues, representing a substantial advance in the understanding of the gene regulation landscape in pigs. This multi-tissue catalog of regulatory variants further advances our understanding of biological mechanisms underlying complex traits of economic importance in pigs. On average, about 80% of GWAS loci tested in pigs are linked to candidate target genes by molQTL in the PigGTEx, comparable with 78% of GWAS loci linked by GTEx in humans[7]. The PigGTEx will eventually enhance genetic improvement programs through the development of advanced biology-driven genomic prediction models that depend on informative SNPs[28]. We also demonstrate the level of similarity between pigs and humans in gene expression, gene regulation and complex trait genetics. This extensive comparison of the pig and human genomes at multiple biological levels will be instructive for deciding which human diseases and complex traits make the pig the most suitable animal model.

Although a fraction of regulatory effects are shared across tissues, we note that some tissues, like the testis and those from early developmental stages, are distinct from other primary tissues. Due to the differences in sample size and other biological factors (for example, breed and cell-type composition) across tissue types in the current phase of PigGTEx, underrepresented tissues at multiple development stages are still required to gain a more comprehensive view of tissue-specific gene regulation and to refine the tissue-trait map in pigs. To elucidate gene regulation at single-cell resolution, we conducted an exploratory analysis to discover cell-type-interaction regulatory effects through an in silico cell-type deconvolution[18]. The cieQTL identified for several cell types indicate that a vast majority of cell-type-specific $cis$-QTL remain to be detected[29,30]. Compared to $cis$-eQTL, $trans$-eQTL often have smaller effect sizes and thus require hundreds of thousands of samples to be discovered[22,31]. Although integrating multi-omics data provides insight into the molecular mechanisms underlying regulatory variants, experimental follow-ups are necessary to functionally validate and characterize these regulatory variants at large scale[32,33].

## Online content

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

Jinyan Teng [1,54], Yahui Gao[1,2,3,54], Hongwei Yin [4,54], Zhonghao Bai[5,6,54], Shuli Liu[2,7,54], Haonan Zeng[1,54], The PigGTEx Consortium*, Lijing Bai[4], Zexi Cai [5], Bingru Zhao[8], Xiujin Li[9], Zhiting Xu[1], Qing Lin[1], Zhangyuan Pan[10,11], Wenjing Yang[8,10], Xiaoshan Yu[6], Dailu Guan [10], Yali Hou [12], Brittney N. Keel[13], Gary A. Rohrer [13], Amanda K. Lindholm-Perry[13], William T. Oliver[13], Maria Ballester [14], Daniel Crespo-Piazuelo [14], Raquel Quintanilla[14], Oriol Canela-Xandri[6], Konrad Rawlik [15], Charley Xia[16,17], Yuelin Yao [6,18], Qianyi Zhao[4], Wenye Yao[4,19], Liu Yang[4], Houcheng Li[5], Huicong Zhang[5], Wang Liao[6], Tianshuo Chen[6], Peter Karlskov-Mortensen [20], Merete Fredholm[20], Marcel Amills [21,22], Alex Clop [21,23], Elisabetta Giuffra [24], Jun Wu[1], Xiaodian Cai[1], Shuqi Diao[1], Xiangchun Pan[1], Chen Wei[1], Jinghui Li[10], Hao Cheng[10], Sheng Wang[25], Guosheng Su[5], Goutam Sahana [5], Mogens Sandø Lund[5], Jack C. M. Dekkers[26], Luke Kramer[26], Christopher K. Tuggle [26], Ryan Corbett[26], Martien A. M. Groenen [19], Ole Madsen[19], Marta Gòdia[19,21], Dominique Rocha[24], Mathieu Charles [27], Cong-jun Li [2], Hubert Pausch [28], Xiaoxiang Hu[29], Laurent Frantz[30,31], Yonglun Luo [32,33,34], Lin Lin [32,33], Zhongyin Zhou[25], Zhe Zhang[35], Zitao Chen[35], Leilei Cui[36,37,38], Ruidong Xiang[39,40], Xia Shen [41,42,43], Pinghua Li[44], Ruihua Huang[44], Guoqing Tang[45], Mingzhou Li [45], Yunxiang Zhao[46], Guoqiang Yi [4], Zhonglin Tang [4], Jicai Jiang[47], Fuping Zhao[11], Xiaolong Yuan[1], Xiaohong Liu[48], Yaosheng Chen[48], Xuewen Xu[49], Shuhong Zhao [49], Pengju Zhao[50], Chris Haley[6,51], Huaijun Zhou [10], Qishan Wang [35], Yuchun Pan[35], Xiangdong Ding[8], Li Ma [3], Jiaqi Li[1], Pau Navarro [6,51], Qin Zhang[52], Bingjie Li [53], Albert Tenesa [6,51]✉, Kui Li [4]✉, George E. Liu [2]✉, Zhe Zhang[1]✉ & Lingzhao Fang [5,6]✉

[1]State Key Laboratory of Swine and Poultry Breeding Industry, National Engineering Research Center for Breeding Swine Industry, Guangdong Provincial Key Lab of Agro-Animal Genomics and Molecular Breeding, College of Animal Science, South China Agricultural University (SCAU), Guangzhou, China. [2]Animal Genomics and Improvement Laboratory, Henry A. Wallace Beltsville Agricultural Research Center, Agricultural Research Service (ARS), U.S. Department of Agriculture (USDA), Beltsville, MD, USA. [3]Department of Animal and Avian Sciences, University of Maryland, College Park, MD, USA. [4]Shenzhen Branch, Guangdong Laboratory of Lingnan Modern Agriculture, Key Laboratory of Livestock and Poultry Multi-Omics of MARA, Agricultural Genomics Institute at Shenzhen, Chinese Academy of Agricultural Sciences, Shenzhen, China. [5]Center for Quantitative Genetics and Genomics, Aarhus University, Aarhus, Denmark. [6]MRC Human Genetics Unit at the Institute of Genetics and Cancer, The University of Edinburgh, Edinburgh, UK. [7]School of Life Sciences, Westlake University, Hangzhou, China. [8]College of Animal Science and Technology, China Agricultural University, Beijing, China. [9]Guangdong Provincial Key Laboratory of Waterfowl Healthy Breeding, College of Animal Science and Technology, Zhongkai University of Agriculture and Engineering, Guangzhou, China. [10]Department of Animal Science, University of California, Davis, Davis, CA, USA. [11]Institute of Animal Science, Chinese Academy of Agricultural Sciences, Beijing, China. [12]Beijing Institute of Genomics, Chinese Academy of Sciences and China National Center for Bioinformation, Beijing, China. [13]ARS, USDA, U.S. Meat Animal Research Center, Clay Center, NE, USA. [14]Animal Breeding and Genetics Programme,

Institut de Recerca i Tecnologia Agroalimentàries (IRTA), Torre Marimon, Caldes de Montbui, Spain. [15]Baillie Gifford Pandemic Science Hub, University of Edinburgh, Edinburgh, UK. [16]Lothian Birth Cohort studies, University of Edinburgh, Edinburgh, UK. [17]Department of Psychology, University of Edinburgh, Edinburgh, UK. [18]School of Informatics, The University of Edinburgh, Edinburgh, UK. [19]Animal Breeding and Genomics, Wageningen University and Research, Wageningen, The Netherlands. [20]Animal Genetics, Bioinformatics and Breeding, Department of Veterinary and Animal Sciences, University of Copenhagen, Copenhagen, Denmark. [21]Department of Animal Genetics, Centre for Research in Agricultural Genomics (CRAG), CSIC-IRTA-UAB-UB, Campus de la Universitat Autònoma de Barcelona, Bellaterra, Spain. [22]Departament de Ciència Animal i dels Aliments, Universitat Autònoma de Barcelona, Bellaterra, Spain. [23]Consejo Superior de Investigaciones Científicas, Barcelona, Spain. [24]Paris-Saclay University, INRAE, AgroParisTech, GABI, Jouy-en-Josas, France. [25]State Key Laboratory of Genetic Resources and Evolution, Kunming Institute of Zoology, Chinese Academy of Sciences, Kunming, China. [26]Department of Animal Science, Iowa State University, Ames, IA, USA. [27]Paris-Saclay University, INRAE, AgroParisTech, GABI, SIGENAE, Jouy-en-Josas, France. [28]Animal Genomics, ETH Zurich, Universitaetstrasse 2, Zurich, Switzerland. [29]State Key Laboratory of Agrobiotechnology, College of Biological Sciences, China Agricultural University, Beijing, China. [30]Palaeogenomics Group, Department of Veterinary Sciences, Ludwig Maximilian University, Munich, Germany. [31]School of Biological and Behavioural Sciences, Queen Mary University of London, London, UK. [32]Department of Biomedicine, Aarhus University, Aarhus, Denmark. [33]Steno Diabetes Center Aarhus, Aarhus University Hospital, Aarhus, Denmark. [34]Lars Bolund Institute of Regenerative Medicine, Qingdao-Europe Advanced Institute for Life Sciences, BGI-Research, Qingdao, China. [35]Department of Animal Science, College of Animal Sciences, Zhejiang University, Hangzhou, China. [36]School of Life Sciences, Nanchang University, Nanchang, China. [37]Human Aging Research Institute and School of Life Science, Nanchang University, and Jiangxi Key Laboratory of Human Aging, Jiangxi, China. [38]UCL Genetics Institute, University College London, London, UK. [39]Faculty of Veterinary and Agricultural Science, The University of Melbourne, Parkville, Victoria, Australia. [40]Agriculture Victoria, AgriBio, Centre for AgriBiosciences, Bundoora, Victoria, Australia. [41]State Key Laboratory of Genetic Engineering, School of Life Sciences, Fudan University, Shanghai, China. [42]Center for Intelligent Medicine Research, Greater Bay Area Institute of Precision Medicine, Fudan University, Guangzhou, China. [43]Centre for Global Health Research, Usher Institute, University of Edinburgh, Edinburgh, UK. [44]Institute of Swine Science, Nanjing Agricultural University, Nanjing, China. [45]Farm Animal Genetic Resources Exploration and Innovation Key Laboratory of Sichuan Province, Sichuan Agricultural University, Chengdu, China. [46]College of Animal Science and Technology, Guangxi University, Nanning, China. [47]Department of Animal Science, North Carolina State University, Raleigh, NC, USA. [48]State Key Laboratory of Biocontrol, School of Life Sciences, Sun Yat-sen University, Guangzhou, China. [49]Key Laboratory of Agricultural Animal Genetics, Breeding and Reproduction, Ministry of Education and College of Animal Science and Technology, Huazhong Agricultural University, Wuhan, China. [50]Hainan Institute, Zhejiang University, Yongyou Industry Park, Yazhou Bay Sci-Tech City, Sanya, China. [51]The Roslin Institute, Royal (Dick) School of Veterinary Studies, The University of Edinburgh, Midlothian, UK. [52]College of Animal Science and Technology, Shandong Agricultural University, Tai'an, China. [53]Scotland's Rural College (SRUC), Roslin Institute Building, Midlothian, UK. [54]These authors contributed equally: Jinyan Teng, Yahui Gao, Hongwei Yin, Zhonghao Bai, Shuli Liu, Haonan Zeng. *A list of authors and their affiliations appears at the end of the paper. ✉e-mail: albert.tenesa@ed.ac.uk; likui@caas.cn; george.liu@usda.gov; zhezhang@scau.edu.cn; lingzhao.fang@qgg.au.dk

## The PigGTEx Consortium

Jinyan Teng[1,54], Yahui Gao[1,2,3,54], Hongwei Yin[4,54], Zhonghao Bai[5,6,54], Shuli Liu[2,7,54], Haonan Zeng[1,54], Lijing Bai[4], Zexi Cai[5], Bingru Zhao[8], Xiujin Li[9], Zhiting Xu[1], Qing Lin[1], Zhangyuan Pan[10,11], Wenjing Yang[8,10], Xiaoshan Yu[6], Dailu Guan[10], Yali Hou[12], Brittney N. Keel[13], Gary A. Rohrer[13], Amanda K. Lindholm-Perry[13], William T. Oliver[13], Maria Ballester[14], Daniel Crespo-Piazuelo[14], Raquel Quintanilla[14], Oriol Canela-Xandri[6], Konrad Rawlik[53], Charley Xia[16,17], Yuelin Yao[6,18], Qianyi Zhao[4], Wenye Yao[4,19], Liu Yang[4], Houcheng Li[5], Huicong Zhang[5], Wang Liao[6], Tianshuo Chen[6], Peter Karlskov-Mortensen[20], Merete Fredholm[20], Marcel Amills[21,22], Alex Clop[21,23], Elisabetta Giuffra[24], Jun Wu[1], Xiaodian Cai[1], Shuqi Diao[1], Xiangchun Pan[1], Chen Wei[1], Jinghui Li[10], Hao Cheng[10], Sheng Wang[25], Guosheng Su[5], Goutam Sahana[5], Mogens Sandø Lund[5], Jack C. M. Dekkers[26], Luke Kramer[26], Christopher K. Tuggle[26], Ryan Corbett[26], Martien A. M. Groenen[19], Ole Madsen[19], Marta Gòdia[19,21], Dominique Rocha[24], Mathieu Charles[52], Cong-jun Li[2], Hubert Pausch[27], Xiaoxiang Hu[28], Laurent Frantz[29,30], Yonglun Luo[31,32,33], Lin Lin[31,32], Zhongyin Zhou[25], Zhe Zhang[34], Zitao Chen[34], Leilei Cui[35,36,37], Ruidong Xiang[38,39], Xia Shen[40,41,42], Pinghua Li[43], Ruihua Huang[43], Guoqing Tang[44], Mingzhou Li[44], Yunxiang Zhao[45], Guoqiang Yi[4], Zhonglin Tang[4], Jicai Jiang[46], Fuping Zhao[11], Xiaolong Yuan[1], Xiaohong Liu[47], Yaosheng Chen[47], Xuewen Xu[48], Shuhong Zhao[48], Pengju Zhao[49], Chris Haley[6,15], Huaijun Zhou[10], Qishan Wang[34], Yuchun Pan[34], Xiangdong Ding[8], Li Ma[3], Jiaqi Li[1], Pau Navarro[6,15], Qin Zhang[50], Bingjie Li[51], Albert Tenesa[6,15], Kui Li[4], George E. Liu[2], Zhe Zhang[1] & Lingzhao Fang[5,6]

## Methods

### Ethics

It is not applicable because no biological samples were collected and no animal handling was performed for this study.

### RNA-seq data analysis and molecular phenotype quantification

In total, we gathered 11,323 publicly accessible raw RNA-seq datasets, representing 9,530 distinct samples (downloaded from NCBI SRA by 26 February 2021), of which 98.13% were generated using the Illumina platform. We removed 121 embargoed RNA-seq samples and then processed all the remaining RNA-seq samples using a uniform pipeline. Briefly, we first trimmed adaptors and discarded reads with poor quality using Trimmomatic (v0.39)[34]. We then aligned clean reads to the Sscrofa11.1 (v100) pig reference genome using STAR (v2.7.0)[35]. We kept 8,262 samples with more than 500K clean reads and uniquely mapping rates ≥ 60% for subsequent analysis (Supplementary Table 1). We extracted the raw read counts of 31,871 Ensembl (Sscrofa11.1 v100) genes by featureCounts (v1.5.2)[36] and obtained their normalized expression (that is, transcripts per million (TPM)) using Stringtie (v2.1.1)[37]. We removed 544 samples in which less than 20% of all annotated genes were expressed (TPM ≥ 0.1), resulting in 7,597 samples. We then visualized the variance in gene expression among samples using $t$-distributed stochastic neighbor embedding ($t$-SNE)[38]. After filtering out outliers within each of the tissues, we eventually kept 7,095 samples for subsequent analysis (Supplementary Table 1). We employed MEGA (vX)[39] to build a neighbor-joining tree of these samples based on TPM and then visualized it by iTOL (v6)[40].

For PCG expression, we considered 21,280 PCGs from the Ensembl annotation (Sscrofa11.1 v100). For exon expression of PCGs, we extracted raw read counts of 290,536 exons by featureCounts (v1.5.2)[36] and normalized them as TPM. To explore enhancer expression, we downloaded the previously predicted enhancers (strong active enhancers, EnhA) from 14 pig tissues[15]. We merged these enhancer regions across tissues using bedtools (v2.30.0)[41], resulting in 158,998 nonredundant enhancer regions. To control the potential contamination of transcribed genes, we only focused on transcribed enhancers that were not overlapped with any known gene regions (including protein-coding gene, lncRNA, pseudogene, tRNA, miRNA and snoRNA)[42–44], resulting in 3,679 enhancers. We obtained raw read counts of these nonredundant enhancer regions from all 7,095 RNA-seq samples by featureCounts (v1.5.2)[36], followed by TPM normalization. For lncRNA expression, we obtained 17,162 lncRNAs predicted from 33 Iso-Seq datasets, representing ten tissues from four animals by using FEELnc[45]. We applied the same approach to extract and normalize lncRNA expression as above.

For alternative splicing, we used Leafcutter (v0.2.9)[46] to quantify excision levels of introns and then to identify splicing events within each tissue as described in the following: (1) converting aligned bam files from STAR (v2.7.0) into junction files using the script bam2junc.sh; (2) generating intron clusters using the script leafcutter_cluster.py, and then mapping them to genes by the map_clusters_to_genes.R script with exon coordinates extracted from the Ensembl annotation file (v100); (3) discarding introns without any read count in more than 50% of samples or with fewer than max(10, 0.1$n$) unique values, where $n$ is the sample size; (4) filtering out introns with low complexity: $\sum_i(|z_i| < 0.25) \geq n\text{-}3$ and $\sum_i(|z_i| > 6) \leq 3$, where $z_i$ is the $z$ score of the $i$th cluster read fraction across individuals; (5) using prepare_phenotype_table.py script to normalize filtered counts and convert them into BED format, where start/end positions correspond to the TSS of corresponding genes. Furthermore, we normalized excision levels of introns as percent spliced-in (PSI) values.

### MolQTL mapping

For molQTL mapping within each of the 34 tissues, we only considered SNPs with MAF ≥ 5% and minor allele count ≥ 6, resulting in an average of 2,705,637 SNPs (ranging from 1,815,729 in synovial membrane to 3,004,852 in muscle). We computed genotype PCs based on the filtered SNPs within each of the tissues using SNPRelate (v1.26.0)[47]. We used the top five and ten genotype PCs to account for the population structure among samples in tissues with <200 and ≥200 samples, respectively (Extended Data Fig. 2f). To account for technical confounders among RNA-seq samples, we used the probabilistic estimation of expression residual (PEER) method, implemented in PEER (v1.0) R package[48], to estimate a set of latent covariates within each tissue based on gene expression matrices. We obtained a total of 60 PEER factors in each tissue and assessed their relative contributions (that is, factor weight variance) to gene expression variation using the PEER_getAlpha function. We decided to use the top ten PEER factors for each tissue as covariates when conducting molQTL mapping for PGC, exon, lncRNA and enhancer expression (Extended Data Fig. 2g). For $cis$-sQTL mapping, we estimated and fitted ten PEER factors from the splicing quantifications of genes within each tissue. To understand whether known covariates can be captured by PEER factors, we fitted a linear regression model to estimate the proportion of variance in known confounders that were explained by the top ten PEER factors.

For $cis$-eQTL mapping, we first normalized the PCGs expression across samples within each tissue using the trimmed mean of M-value (TMM) method, implemented in edgeR[49], followed by inverse normal transformation of the TMM. We performed $cis$-eQTL mapping using a linear regression model, implemented in TensorQTL (v1.0.3)[17], while accounting for the estimated covariates. Within each tissue, we filtered out genes with TPM < 0.1 and/or raw read counts < 6 in more than 80% of samples. We defined the $cis$-window of PCG as ±1 Mb of TSS and obtained the nominal $P$ values of $cis$-eQTL with the parameter mode cis_nominal in TensorQTL. We then employed two layers of multiple testing corrections based on the permutation approach[50], implemented in the TensorQTL. In the first layer, we applied an adaptive permutation approach to calculate the empirical $P$ values of variants within each gene and obtained the permutation $P$ value of the lead variant for each gene. In the second layer, we conducted the Benjamini–Hochberg correction for the permutation $P$ values of lead variants across all tested genes and considered genes with FDR < 5% as the genome-wide significant eGenes and genes without significant $cis$-eQTL as non-eGenes. To identify significant $cis$-eQTL associated with eGenes, we defined the empirical $P$ value of the gene that was closest to an FDR of 0.05 as the genome-wide empirical $P$ value threshold (pt). We obtained the gene-level threshold for each gene from the beta distribution by qbeta (pt, beta_shape1, beta_shape2) in R (v4.0.2), where beta_shape1 and beta_shape2 were derived using TensorQTL. We considered SNPs with a nominal $P$ value below the gene-level threshold as significant $cis$-eQTL for a given gene–tissue pair.

Similarly, we normalized the expression of exons, lncRNAs and enhancers to inverse normal transformed TMM across samples and excluded lowly expressed elements using the same approach as for PCG. We conducted $cis$-QTL mapping for exons ($cis$-eeQTL), lncRNAs ($cis$-lncQTL) and enhancers ($cis$-enQTL) using TensorQTL. For $cis$-eeQTL mapping, we defined the $cis$-window of an exon as the ±1 Mb region of its source gene's TSS. For exons, lncRNA and enhancer $cis$-QTL mapping, we defined the $cis$-window as the ±1 Mb region of the TSS of the source gene, of its TSS and its TSS, respectively. We declared significant $cis$-QTL for exons, lncRNAs and enhancers using the same approach as done for the $cis$-eQTL mapping. We defined exons, lncRNAs and enhancers with at least one significant $cis$-QTL as eExon, eLncRNA and eEnhancer, respectively.

We performed $cis$-sQTL mapping for genes with splicing quantifications (PSI values) and tested SNPs within ±1 Mb of TSS using TensorQTL (v1.0.3)[17] while accounting for the estimated covariates. To compute the empirical $P$ value of $cis$-sQTL, we grouped all intron clusters of a gene with the parameter: --phenotype_groups option in the permutation mode of TensorQTL (v1.0.3)[17]. We defined sGene and significant $cis$-sQTL using the same approach as used for $cis$-eQTL

mapping. We refer to the eGene, eExon, eLncRNA and eEnhancer above, as well as sGene collectively as eMolecule.

## Conditionally independent molQTL mapping

To identify the multiple independent *cis*-QTL signals of a given eMolecular, we applied a forward–backward stepwise regression approach[7], using TensorQTL (v1.0.3) with the parameter: --mode cis_independent[17]. We set the gene-level significance threshold to be the maximum β-adjusted *P* value for eMolecules within each tissue after correcting for multiple testing as described above. At each iteration, we scanned the new *cis*-QTL after adjusting for all previously discovered *cis*-QTL and covariates. In addition, we further employed SuSiE-inf (v1.2)[51] to fine-map the potential causal *cis*-QTL for each eMolecule.

## The tissue-sharing patterns of molQTL

To understand the shared or specific genetic regulatory mechanisms between tissues, we performed a meta-analysis of molQTL across all 34 tissues using MashR (v0.2–6)[52] and METASOFT (v2.0.1)[53] as described above. For MashR (v0.2-6), we only considered the *z* scores from TensorQTL (v1.0.3; slope/slope_se) of the top *cis*-molQTL. We obtained the estimated effect sizes (that is, posterior means) and the corresponding significance levels (that is, local false sign rate (LFSR)) from the mash function. We defined a molQTL with LFSR < 0.05 as active in a given tissue. To estimate the pairwise tissue similarity with regard to genetic regulation of gene expression, we calculated the pairwise Spearman's correlation of effect size estimates of *cis*-molQTL between any tissue pairs, focusing on SNPs with LFSR < 0.05 in at least one tissue. For METASOFT (v2.0.1), we used summary statistics (that is, slope and slope_se) from TensorQTL (v1.0.3) of molQTL across all tissues. We estimated the meta-analytic effect size using a fixed effect model and calculated *M* values (posterior probabilities) using the MCMC method. We considered a molQTL with *M* > 0.7 active in tissue. To evaluate the similarity of tissue-clustering patterns across different data types (that is, PCG expression, splicing quantifications, exon expression, lncRNA expression, enhancer expression, *cis*-eQTL, *cis*-sQTL, *cis*-lncQTL, *cis*-eeQTL and *cis*-enQTL), we performed *k*-means clustering using the *k*-means function in the stats R package (v4.0.2), in which parameter *k* was allowed to range from 2 to 20 and the maximum number of iterations was 1,000,000. We calculated the pairwise Rand index to measure the clustering similarity using the rand.index function in the fossil (v0.4.0) R package (v4.0.2)[54].

## GWAS summary statistics

To investigate the regulatory mechanisms underpinning complex traits in pigs, we systematically integrated the identified molQTL with summary statistics of 268 meta-GWAS from 207 complex traits of economic importance, representing five trait domains (Supplementary Table 18). In total, we performed 2,056 separate GWAS and conducted the meta-GWAS analysis for the same traits across different populations based on GWAS summary statistics using METAL (v2011-03-25)[55], resulting in 268 meta-GWAS results. To perform the integrative analysis of GWAS and molQTL, we overlapped significant GWAS loci with the 3,087,268 SNPs tested in the molQTL mapping, resulting in 1,507 GWAS loci with lead SNP *P* < 1 × 10⁻⁵.

## Enrichment of molQTL and trait-associated variants

To examine whether molQTL was enriched among the significant GWAS variants, we applied three distinct approaches as described in the following. First, we used a simple overlapping approach to examine whether a significant molQTL is more likely to be a significant trait-SNP as described in ref. 9 Briefly, for each tissue, we kept SNPs with the most significant nominal *P* value for a gene and scaled *P* values to a comparable level (λ = 10) across 34 tissues. We selected the minimum *P* value of each SNP in the 34 tissues as the background set, from which we extracted *P* values for SNPs that overlapped with significant GWAS loci.

Second, we applied QTLEnrich (v2)[7] to quantify the enrichment degree between significant molQTL and GWAS loci. We only used summary statistics of 198 GWAS for which ≥80% of SNPs were also tested in the molQTL mapping. Third, we applied the mediated expression score regression method to estimate the heritability of complex trait that was mediated by the *cis*-genetic component of different molecular phenotypes ($h^2_{med}$)[56].

## *Cis*-molQTL-GWAS colocalization

To identify shared genetic variants between the molecular phenotypes and complex traits, we performed a colocalization analysis of molQTL and GWAS loci using fastENLOC (v1.0)[23]. Briefly, we obtained the probabilistic annotation of molQTL from the DAP-G (v1.0.0)[57] and then used the summarize_dap2enloc.pl script to generate the annotation file of multi-tissue molQTLs. We generated approximate LD blocks across the entire genome based on the PGRP using PLINK (v1.90)[58]. We applied the TORUS tool to generate the posterior inclusion probability of each LD block based on GWAS *z* scores[59], followed by the colocalization analysis with fastENLOC (v1.0). We obtained the regional colocalization probability (RCP) of each LD-independent genomic region using a natural Bayesian hierarchical model[60] and considered a gene with RCP > 0.9 as significant. To identify the trait-relevant tissues, we calculated a 'relevance score' between a tissue and a trait by dividing the number of colocalized genes by the product of sample size and eGene proportion in this tissue. We only considered 14 tissues with ≥100 samples.

## TWAS of complex traits

To explore whether the overall *cis*-genetic component of a molecular phenotype is associated with complex traits, we conducted single- and multi-tissue TWAS using S-PrediXcan[25] and S-MultiXcan in MetaXcan (v0.6.11)[26], respectively, based on the summary statistics of the meta-GWAS. Briefly, we employed the nested cross-validated elastic net model implemented in S-PrediXcan to predict the five types of molecular phenotypes in all 34 tissues. To train the predictive model, we used the confounder-corrected expression or PSI values as phenotypes and SNPs within the *cis*-windows of genes as genotypes. We kept only predictive models with cross-validated correlation ρ > 0.1 and prediction performance *P* < 0.05 for further TWAS analysis. We ran S-PrediXcan on all 268 GWAS to obtain gene–trait associations at a single-tissue level. Based on results from S-PrediXcan, we ran S-MultiXcan to integrate predictions from multiple tissues, yielding the multi-tissue TWAS results. We applied Bonferroni correction and considered a corrected *P* < 0.05 as significant.

## MR analysis between molQTL and GWAS loci

We conducted MR analysis to infer the causality between molecular phenotypes and complex traits using the SMR (v1.03)[24]. We first converted the summary statistics of molQTL from TensorQTL (v1.0.3) to BESD format using SMR with the options: --fastqtl-nominal-format --make-besd. We only considered eMolecules with top nominal *P* value < 1 × 10⁻⁵ for the SMR test. We defined gene–trait pairs to pass the SMR test if the Benjamini–Hochberg-adjusted *P*SMR < 0.05 and *P*GWAS < 1 × 10⁻⁵. For gene–trait pairs that passed the SMR test, we performed the heterogeneity in dependent instruments (HEIDI) test, with *P*HEIDI ≥ 0.05 reflecting that we could not reject a single causal variant with effects on both molecular phenotype and complex trait. As a *cis*-regulator, lncRNA can regulate the expression of neighboring PCGs and then can influence complex traits. To understand this etiology of complex traits, we performed an integrative SMR analysis that used three layers of summary-level information from *cis*-lncQTL, *cis*-eQTL and GWAS. We used the summary statistics of *cis*-lncQTL and *cis*-eQTL as the exposure and the outcome input for SMR (v1.03)[61], respectively, which detected pleiotropic effects between lncRNA and PCG expression. We used Bonferroni correction within each tissue and defined a corrected *P* < 0.05 as significant.

## Comparative analysis between pigs and humans

To explore the genetic similarity of complex traits between pigs and humans, we performed a comparative analysis of TWAS summary statistics. We downloaded public human GWAS summary statistics for 136 complex traits, representing 18 trait domains (Supplementary Table 29). Based on the predictive models in human GTEx v8 (ref. [62]), we applied the S-PrediXcan to conduct TWAS for all 136 complex traits across 49 human tissues. We only kept TWAS results from 11 major tissues in humans that had matched tissues with ≥100 samples in pigs. We only considered 15,944 one-to-one orthologous genes. For a trait pair, we calculated the Pearson's correlation of absolute effect size estimated of orthologous genes between pigs and humans within the matching tissue. We applied Benjamini–Hochberg correction for *P* values of all tested correlations and defined an FDR < 10% as significant. To investigate whether GTEx-like resources can facilitate cross-species gene mapping of complex traits through borrowing 'information' at the level of orthologous genes instead of individual variants, we performed a cross-species meta-TWAS analysis through modifying a multi-ancestry meta-TWAS method in humans[63]. We calculated the *z* statistics of meta-TWAS as follows: $z_{meta} = \frac{n_i z_{TWAS,i} + n_j z_{TWAS,j}}{\sqrt{n_i^2 + n_j^2}}$, where $z_{TWAS,i}$ and $z_{TWAS,j}$ were the *z* statistics from pig TWAS and human TWAS results, respectively; $n_i$ and $n_j$ were the population size of pig TWAS and human TWAS, respectively. If the tested trait is a case–control study, we adjusted the sample size as $4/(\frac{1}{n_{cases}} + \frac{1}{n_{controls}})$. We chose several well-recognized homologous trait pairs between humans and pigs to perform the meta-TWAS, and we also selected several nonhomologous trait pairs as negative controls. We divided orthologous genes into ten bins sorted by *P* values of pig TWAS and estimated the heritability enrichment of different gene bins in homologous trait of humans using LD score regression implemented in LDSC[64]. We performed the PheWAS based on 4,756 GWAS, including 3,302 traits in GWAS ATLAS[27].

## Statistics and reproducibility

No statistical method was used to predetermine the sample size. The details of data exclusions for each specific analysis are available in the Methods section. For all the boxplots, the horizontal lines inside the boxes show the medians. Box bounds show the lower quartile (*Q*1, the 25th percentile) and the upper quartile (*Q*3, the 75th percentile). Whiskers are minima (*Q*1 − 1.5× IQR) and maxima (*Q*3 + 1.5× IQR), where IQR is the interquartile range (*Q*3−*Q*1). Outliers are shown in the boxplots unless otherwise stated. The experiments were not randomized, as all the datasets are publicly available from observational studies. The investigators were not blinded to allocation during experiments and outcome assessment, as the data were not from controlled randomized studies.

## Reporting summary

Further information on research design is available in the Nature Portfolio Reporting Summary linked to this article.

## Data availability

All raw data analyzed in this study are publicly available for download without restrictions from SRA (https://www.ncbi.nlm.nih.gov/sra/) and BIGD (https://bigd.big.ac.cn/bioproject/) databases. Details of RNA-seq, WGS, WGBS, single-cell RNA-seq and Hi-C datasets can be found in Supplementary Tables 1, 2, 5, 8 and 9, respectively. All the WGS data newly generated in this study are available under CNCB GSA (https://ngdc.cncb.ac.cn/) accessions PRJCA016120, PRJCA016130, PRJCA017284, PRJCA016012 and PRJCA016216. All processed data and the full summary statistics of molQTL mapping are available at http://piggtex.farmgtex.org/.

## Code availability

All the computational scripts and codes for RNA-seq, WGS, WGBS, single-cell RNA-seq and Hi-C dataset analyses, as well as the respective quality control, molecular phenotype normalization, genotype imputation, molQTL mapping, functional enrichment, colocalization, SMR and TWAS, are available at the FarmGTEx GitHub website (https://github.com/FarmGTEx/PigGTEx-Pipeline-v0, https://doi.org/10.6084/m9.figshare.24247771)[65].

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

## Acknowledgements

Zhe Zhang (SCAU) acknowledges funding from the National Natural Science Foundation of China (32022078), the National Key R&D Program of China (2022YFF1000900) and the Local Innovative and Research Teams Project of Guangdong Province (2019BT02N630), and support from National Supercomputer Center in Guangzhou, China. Y.C., Zhe Zhang (SCAU), Jiaqi Li, X. Liu, X.D. and S.Z. acknowledge funding from the China Agriculture Research System (CARS-35). L. Fang acknowledges funding from HDR-UK under award HDR-9004 and the European Union's Horizon 2020 research and innovation program under the Marie Skłodowska-Curie grant agreement 801215. G.E.L. was supported by United States Department of Agriculture (USDA) National Institute of Food and Agriculture (NIFA) AFRI under grant numbers 2019-67015-29321 and 2021-67015-33409 and the appropriated project 8042-31000-112-00-D, 'Accelerating Genetic Improvement of Ruminants Through Enhanced Genome Assembly, Annotation, and Selection' of the USDA Agricultural Research Service (ARS). This research used resources provided by the SCINet project of the USDA ARS under project 0500-00093-001-00-D. Mention of trade names or commercial products in this article is solely for the purpose of providing specific information and does not imply recommendation or endorsement by the USDA. The USDA is an equal opportunity provider and employer. A.T. acknowledges funding from the BBSRC through program grants BBS/E/D/10002070 and BBS/E/D/30002275, MRC research grant MR/P015514/1 and HDR-UK award HDR-9004. P.N. and C.H. were supported by the Medical Research Council, UK (grant MC_UU_00007/10). O.C.-X. was supported by MR/R025851/1. M.B. and D.C.-P. belonged to a Consolidated Research Group AGAUR, ref. 2017SGR-1719, and D.C.-P. was supported by the GENE-SWitCH project (https://www.gene-switch.eu), which is funded by the European Union's Horizon 2020 research and innovation program under the grant agreement 817998. R.X. was supported by the Australian Research Council's Discovery Projects (DP200100499). L.M. was supported in part by AFRI under grants 2020-67015-31398 and 2021-67015-33409 from the USDA NIFA. B.N.K. and G.A.R. were supported by appropriated project 3040-31000-099-000-D, 'Identifying Genomic Solutions to Improve Efficiency of Swine Production' of the ARS of the USDA. A.K.L.P. and W.T.O. were supported by appropriated project 3040-31000-102-000-D, 'Optimizing Nutrient Management and Efficiency of Beef Cattle and Swine' of the ARS of the USDA. Z.P., D.G. and H. Zhou, and computational resource were supported in part by Agriculture and Food Research Initiative Competitive grants 2018-67015-27501 and 2015-67015-22940. All the funders had no role in study design, data collection and analysis and decision to publish or prepare the manuscript.

We thank all the researchers who have contributed to the publicly available data used in this research. We thank the valuable comments and suggestions from D. Speed, G. Paul Ramstein (QGG, Aarhus University, Denmark), M. E. Goddard (The University of Melbourne, Australia), C. Ponting (IGC, The University of Edinburgh, UK) and G. Larson (The University of Oxford, UK). Figure 1d was created with BioRender.com. For the purpose of open access, the author has applied a CCBY public copyright license to any author-accepted manuscript version arising from this submission.

## Author contributions

L. Fang, Zhe Zhang (SCAU), G.E.L., A.T. and K.L. conceived and designed the project. Y.G., S.L., X. Li, H.Y., B.Z., W. Yang, W. Yao, Y.Y., H.L., H. Zhang and X.P. performed bioinformatic analyses of RNA-seq data analysis. H.Y., S.D., L.B., S.W., D.G., L.Y. and Z.Chen conducted whole-genome sequence data analysis. Y.G., Q. Zhao and Z.P. performed omics data analysis. J.T. conducted genotype imputation and molQTL mapping. Z.X., H. Zeng, C.W., W.L., T.C. and X. Yu prepared the summary statistics of GWAS in pigs and humans. J.T., Q.L., X.C. and J.W. integrated molQTL with GWAS. Z.B., J.T., C.X. and Jinghui Li led the comparison of PigGTEx and human GTEx. B.N.K., G.A.R., A.K.L.P., W.T.O., M.B., D.C., M.C. and L.K. contributed to the validation and functional annotation of molQTL. P.N., Y.H., B.L., Z. Cai, P.Z., D.R., C.L., H.P., X.H., L. Frantz, Y.L., L.L., L.C., J.J., R.H., Z.T., M.L., S.Z. and Y.C. contributed to the critical interpretation of analytical results before and during manuscript preparation. H. Zeng, J.T., Zhe Zhang (SCAU) and L. Fang built the PigGTEx web portal. L. Fang, Zhe Zhang (SCAU), G.E.L., K.L., M.B., R.Q., O.C.-X., K.R., P.K.M., M.F., M.A., A.C., E.G., H.C., G. Su, G. Sahana, M.S.L., J.C.M.D., C.K.T., R.C., M.A.M.G., O.M., M.G., Z. Zhou, Z. Zhang, R.X., X.S., P.L., G.T., Y.Z., G.Y., F.Z., P.N., X. Yuan, X. Liu, L.M., H.S., X.X., Q.W., X.D., H. Zhou, Jiaqi Li, C.H., Y.P., B.L. and Q. Zhang contributed to the data and computational resources. L. Fang, J.T., Y.G. and Z.B. drafted the manuscript. All authors read, edited and approved the final manuscript.

## Competing interests

The authors declare no competing interests.

## Additional information

**Extended data** is available for this paper at https://doi.org/10.1038/s41588-023-01585-7.

**Correspondence and requests for materials** should be addressed to Albert Tenesa, Kui Li, George E. Liu, Zhe Zhang or Lingzhao Fang.

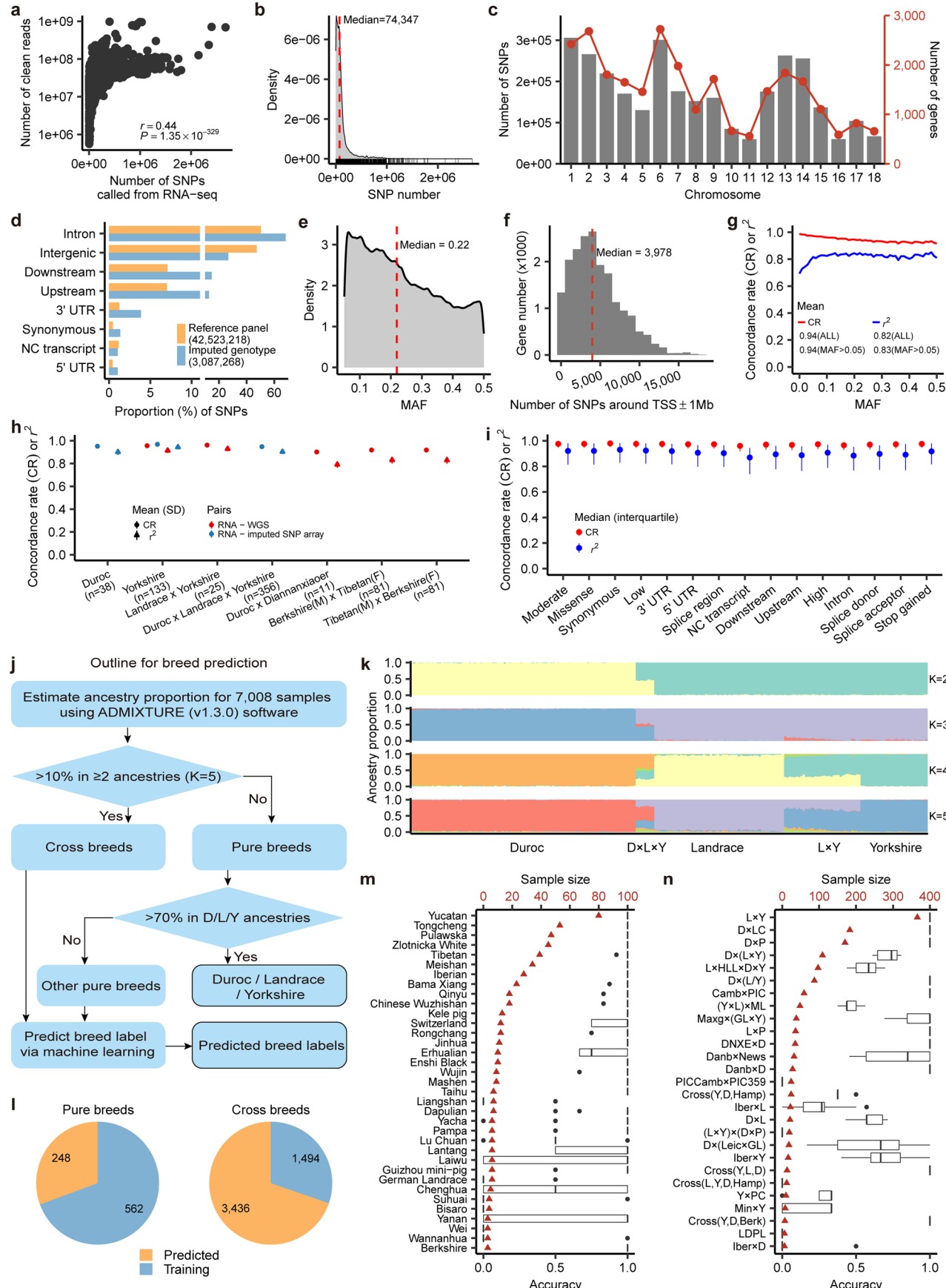

**Extended Data Fig. 1 | See next page for caption.**

**Extended Data Fig. 1 | Genotype calling and imputation and breed prediction.**
**a**, Pearson's correlation ($r$) between number of clean reads and number of called SNPs across 7,095 RNA-Seq samples. The $P$-value is obtained by Pearson's $r$ test.
**b**, Distribution of the number of SNPs called from 7,095 RNA-Seq samples.
**c**, Number of imputed SNPs (left, gray bars) from 7,008 RNA-Seq samples across 18 pig chromosomes after quality control ($DR^2 \geq 0.85$, minor allele frequency $\geq$ 0.05). The red point represents the number of genes (right) in each chromosome in the Sscrofa11.1. assembly (Ensembl v100). **d**, Distribution of 42,523,218 SNPs from the Pig Genomics Reference Panel (PGRP) and 3,087,268 imputed SNPs used for molecular QTL (molQTL) mapping across eight genomic features.
**e**, Minor allele frequency (MAF) of imputed SNPs in 7,008 RNA-Seq samples.
**f**, Distribution of the number of imputed SNPs around 1 Mb of transcript start site (TSS) of 18,911 protein-coding genes. **g**, Concordance rate (CR) and squared correlation ($r^2$) of imputed and observed genotypes in 50 evenly spaced MAF bins based on individuals that are not present in the PGRP. 'ALL' represents the entire variants. **h**, CR and $r^2$ of imputed genotypes from RNA-Seq only and those directly called from whole-genome sequence (WGS) data (red), and imputed genotypes (blue) from SNP array, respectively, in the same individuals. Point and whisker are mean and standard deviation, respectively. Labels of $x$-axis are breeds and number of individuals. **i**, CR and $r^2$ (median and interquartile) of imputed and observed genotypes in different genomic features. Point and whisker are median and interquartile, respectively. **j**, The overall pipeline utilized to predict missing breed labels for RNA-Seq samples. **k**, Estimated ancestry proportion of Duroc (n = 485), Landrace (n = 280), Yorkshire (n = 145), Landrace×Yorkshire (n = 165) and Duroc×Landrace×Yorkshire (n = 40) samples. **l**, Distribution of sample size of training and prediction sets in pure and cross breeds. **m**,**n**, Accuracy of breed prediction for pure breeds (**m**) and cross breeds (**n**) measured by cross-validation. The red triangle represents the sample size of the target breed.

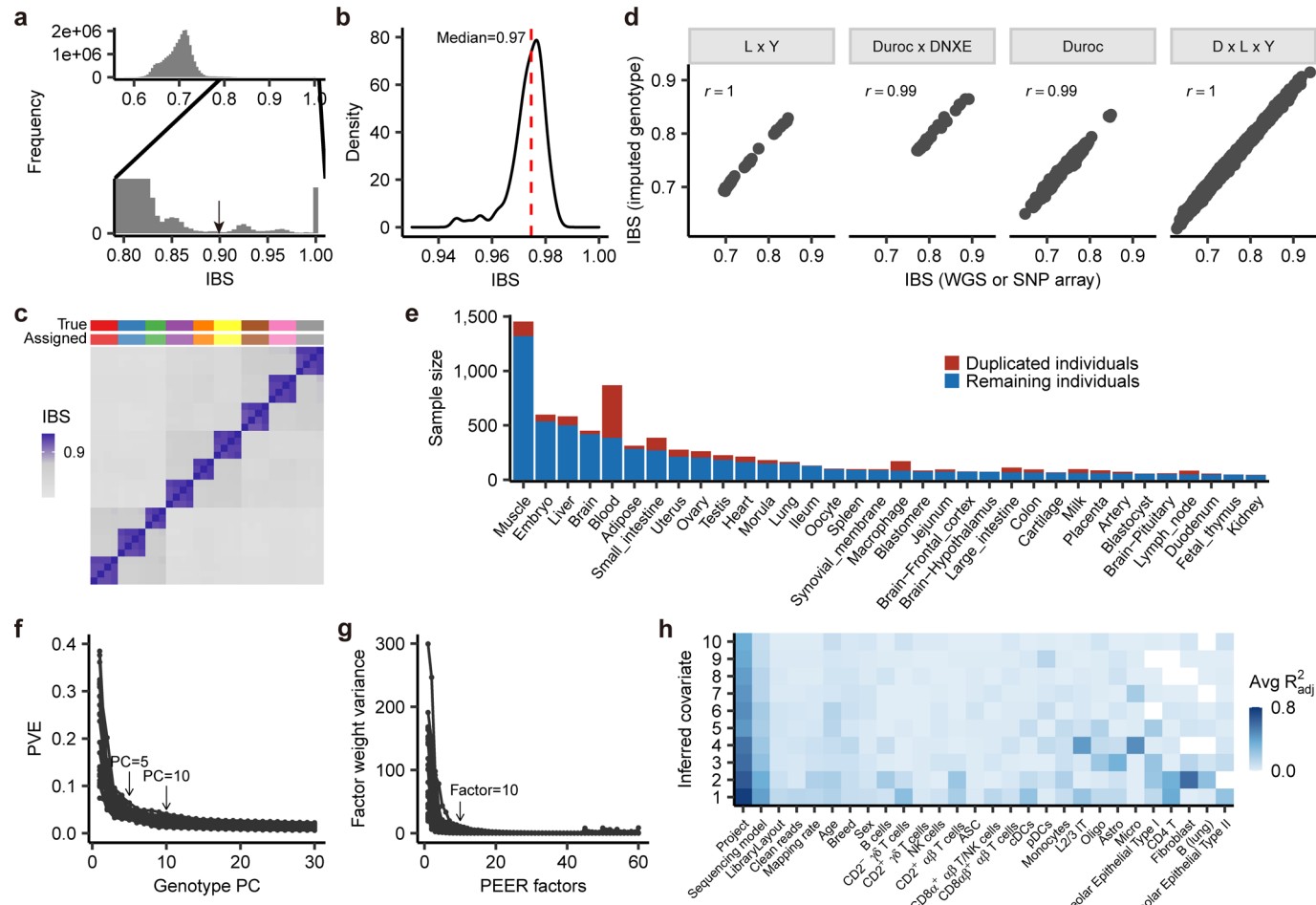

**Extended Data Fig. 2 | Detection of duplicated individuals and confounders of RNA-Seq samples. a**, Distribution of identity-by-state (IBS) distances among 7,008 RNA-Seq samples, which are calculated using 12,207 LD-independent SNPs ($r^2 < 0.2$). **b**, Density of IBS distances that were computed using genotypes derived from RNA-Seq only and whole-genome sequence (WGS) or SNP array data in the same individuals (n = 227). **c**, Heatmap of IBS distance of 25 RNA-Seq samples from 9 individuals. The same color on the top of panel represents samples from the same individuals. True: true individual label; Assigned: assigned individual label using an IBS distance cutoff of 0.9. **d**, Pearson's correlation (*r*) between IBS distance calculated from imputed genotypes and those calculated from WGS or SNP array data across four different populations. L×Y: Landrace and Yorkshire

cross breed (n = 25); Duroc×DNXE: Duroc and Diannanxiaoer cross breed (n = 11); Duroc: Duroc pure breed (n = 37); D×L×Y: composite population with 1/4 Duroc, 1/2 Landrace and 1/4 Yorkshire (n = 179). **e**, Duplicated and remaining individuals in each of the 34 pig tissues used for molecular QTL mapping. Sample pairs with IBS > 0.9 were considered as duplicated individuals. **f**, Proportion of variance explained (PVE) by genotype principal components (PC) in each of 34 tissues (lines). **g**, Factor weight variance of probabilistic estimation of expression residual (PEER) factors in each of 34 tissues (lines). **h**, Proportion of variance (adjusted $R^2$) of known confounders captured by the top 10 inferred PEER factors, calculated using the *lm* function in R (v4.0.2).

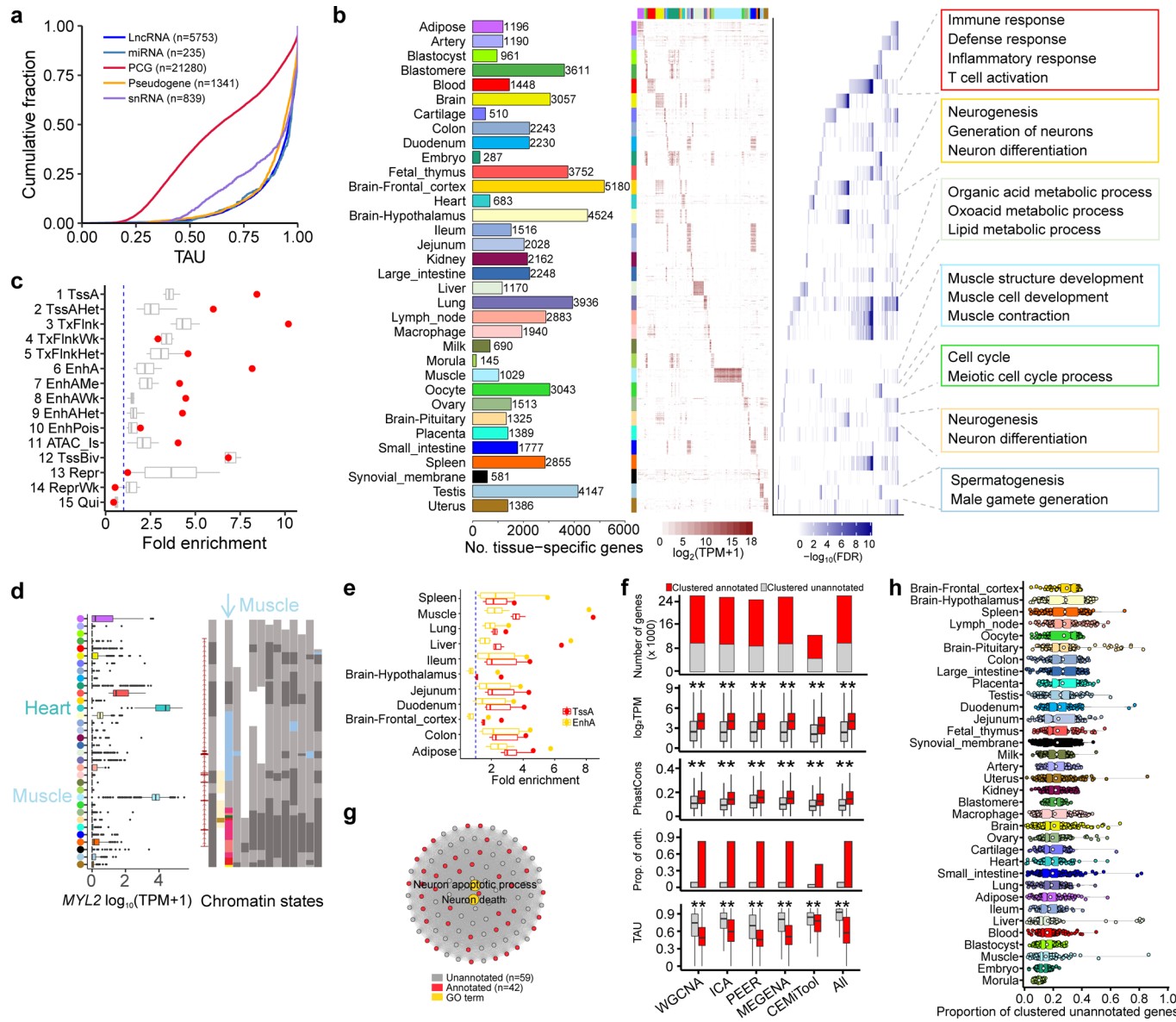

**Extended Data Fig. 3 | The pig gene expression atlas. a**, Tissue-specific expression of five transcript types reflected by the TAU score. PCG: protein-coding genes. **b**, Gene numbers (left), expression pattern (middle, transcripts per million, TPM), and enriched Gene Ontology (GO) terms (right) of tissue-specific genes in 34 tissues. **c**, Enrichment of muscle-specific genes in 15 chromatin states across 14 pig tissues[16]. The red dots represent respective chromatin states in muscle. The blue line indicates enrichment fold = 1. **d**, Expression profiles of *MYL2* gene across 34 tissues (left). The tissue color key is the same as in (**b**). Chromatin state distribution (right) around *MYL2* in 14 pig tissues[16]. In brief, red is for promoters, yellow for enhancers, blue for open chromatin and gray for repressed regions. **e**, Enrichment of tissue-specific genes for two active chromatin states across 11 tissues, which have both chromatin states and gene expression data. The dots represent enrichments from matching tissues. TssA is for active TSS (promoter), and EnhA for active enhancers. **f**, Comparison of genes with and without functional annotation (referred to as 'annotated

genes' and 'unannotated genes', respectively) in gene co-expression modules at different biological layers. The gene co-repression analysis was conducted using five complementary methods, including WGCNA, ICA, PEER, MEGENA and CEMiTool. 'All' shows the combined results from the five methods. The functional annotation was based on the Gene Ontology database (version 2022-01-18). The plots from top to bottom include gene counts, expression level, PhastCons score from 100 vertebrate genomes, proportion of orthologous genes in humans and TAU values. Significant differences between annotated and unannotated genes were obtained using a two-sided Student *t*-test. ** means *P* < 0.01. **g**, An example of gene co-expression module in the pituitary, which includes 59 unannotated and 42 annotated genes, respectively. The functional annotated genes are significantly (*P* = 8 × 10⁻³) enriched in neuron apoptotic processes. The gray edges between genes represent Pearson's correlations of expression across all 53 samples in the pituitary. **h**, The proportion of unannotated genes in each gene co-expression modules across 34 tissues.

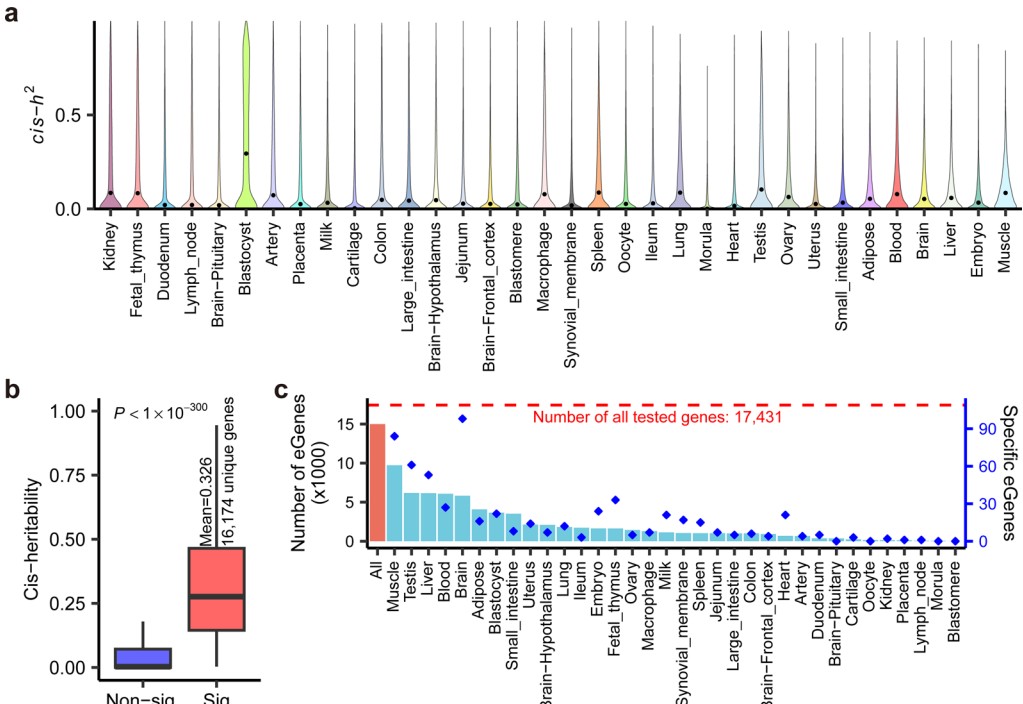

**Extended Data Fig. 4 | *Cis*-heritability of gene expression across 34 pig tissues. a**, Distribution of estimated *cis*-heritability (*cis-h²*) of gene expression across 34 tissues. The black point represents the median of *cis-h²* of all tested genes in a tissue. **b**, Box plot showing the *cis-h²* estimates of genes across 34 tissues that are significant (likelihood ratio test *P* < 0.05) or non-significant,

where 16,174 (93%) unique genes have significant *cis*-heritability in at least one tissue. The *P* value was calculated by two-sided Student *t*-test. **c**, The number of eGenes in each tested tissue, with 86% of the tested genes (red bar, left) are eGenes in at least one tissue. The blue points represent the number of tissue-specific eGenes.

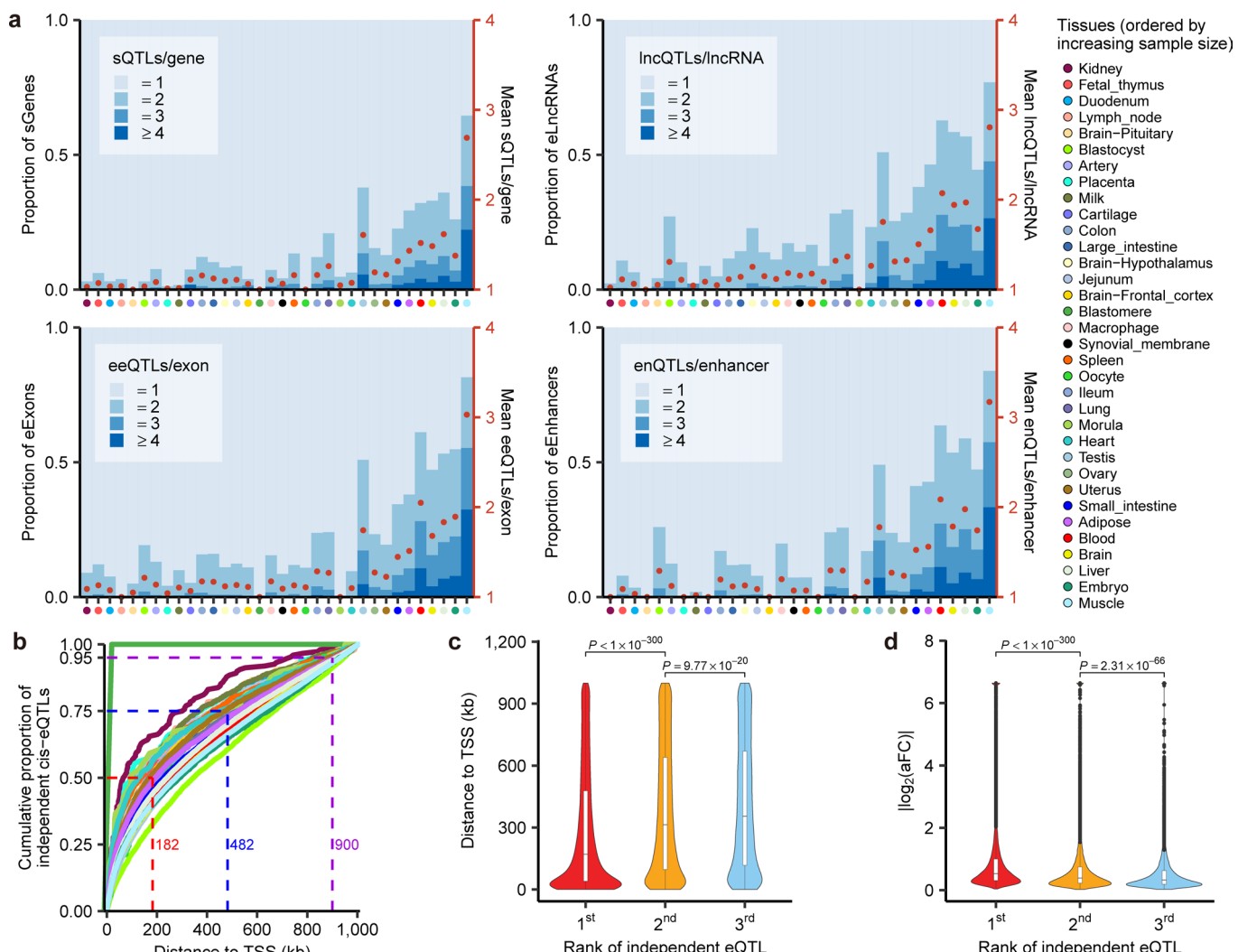

**Extended Data Fig. 5 | Conditionally independent molecular QTLs (molQTL).** **a**, Distribution and average number (red dots, right *y*-axis) of conditionally independent *cis*-QTL per eMolecules across 34 tissues. Tissues (*x*-axis) are ordered by increasing sample size. **b**, Cumulative proportion of distance to the transcription start site (TSS) of target genes for conditionally independent *cis*-eQTL in each of 34 tissues. The meanings of the colors of curved lines are the same as the color key in panel (**a**). **c,d**, Comparison of distance to TSS (**c**) and effect size (|log₂(aFC)|) (**d**) among top three independent *cis*-eQTL per eGene across 34 tissues. The aFC is for allelic fold change. The *P* values were obtained by the two-sided Wilcoxon rank-sum test.

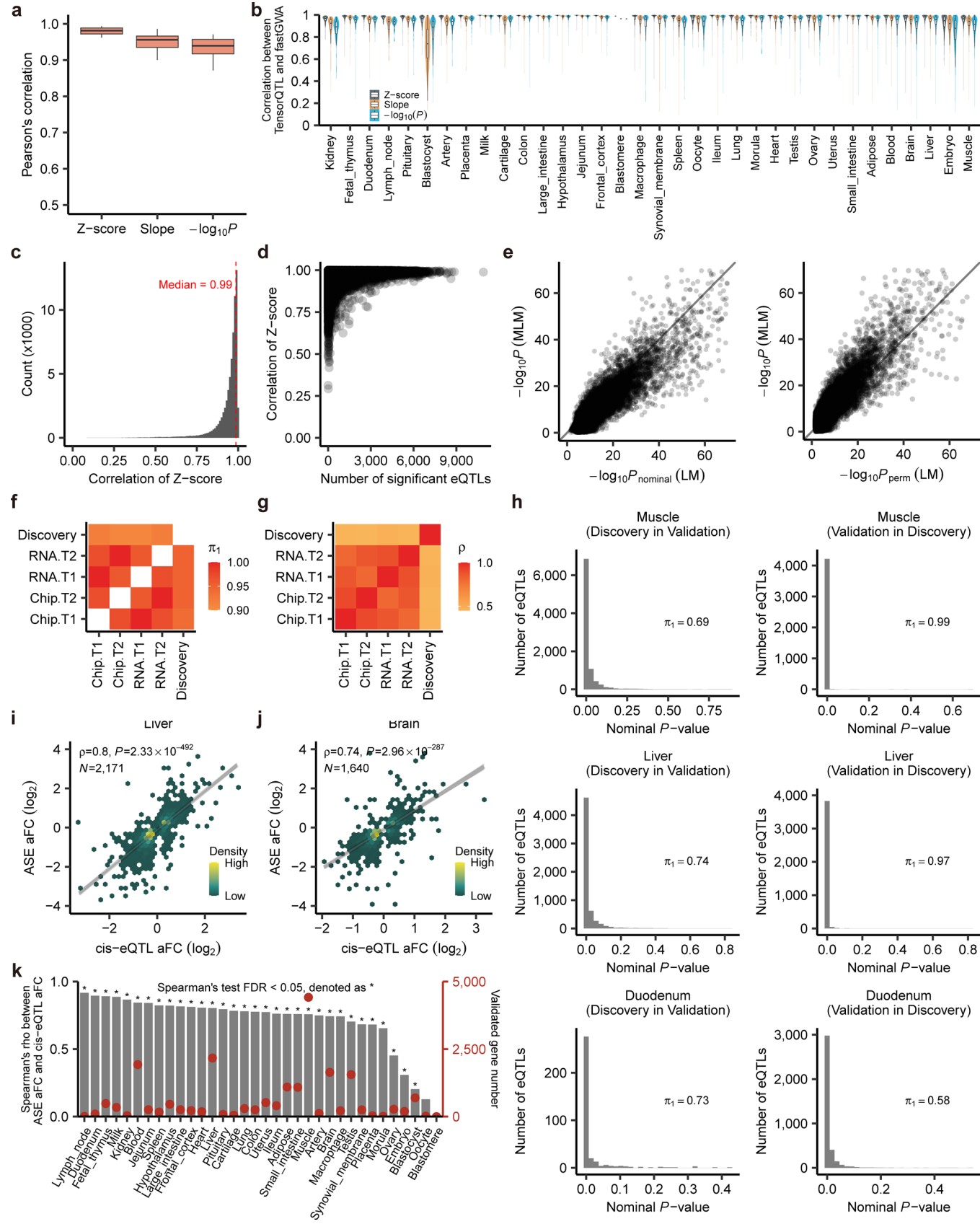

**Extended Data Fig. 6 | See next page for caption.**

**Extended Data Fig. 6 | Validation of *cis*-eQTL. a**, Pearson's correlation of combined summary statistics (for example, Z-score, slope and *P*-value (-$\log_{10}$ scale)) of *cis*-eQTL for all the eGenes across 34 tissues between TensorQTL (linear model, LM) and fastGWA (mixed linear model, MLM). **b**, Pearson's correlation of summary statistics for each eGene in each tissue between LM and MLM. **c**, Distribution of the Pearson's correlations of Z-score between LM and MLM. **d**, Relationship between correlations of Z-score and the number of significant eQTL across all the eGenes. **e**, Correlation of *P* values derived from MLM and nominal (left) or permutation-corrected (right) *P* derived from LM for the lead eQTL of all the eGenes. **f**, Replication rates ($\pi_1$) of blood *cis*-eQTL between the PigGTEx discovery population (n = 386, Discovery) and the external datasets (n = 179). For $\pi_1$ calculation, rows are discovery populations, and columns are replication populations. The external datasets include whole-blood-cell RNA-Seq data and SNP Chip array (Chip) from 179 animals at two developmental stages (T1 and T2). The prefix 'RNA' and 'Chip' indicate imputed genotypes from RNA-Seq and SNP array, respectively. **g**, Spearman's correlation (ρ) of effect size (z-scores) for blood *cis*-eQTL among the same populations above. **h**, Replication rates ($\pi_1$) of PigGTEx *cis*-eQTL in external validation datasets of three tissues, including muscle ($n_{PigGTEx}$ = 1,321, $n_{external}$ = 100), liver ($n_{PigGTEx}$ = 501, $n_{external}$ = 100) and duodenum ($n_{PigGTEx}$ = 49, $n_{external}$ = 100). The *x*-axis is the nominal *P*-value of *cis*-eQTL detected from dataset$_2$ and is significant in dataset$_1$ (that is, dataset$_1$ in dataset$_2$). **i,j**, Spearman's correlation (ρ) of effect sizes (allelic fold change, aFC in $\log_2$ scale) between *cis*-eQTL and matched allele-specific expression (ASE) loci in the liver (**i**) and brain (**j**). N indicates number of tested loci. The lines are fitted by a linear regression model using the *geom_smooth* function from ggplot2 (v3.3.2) in R (v4.0.2). The shading represents the standard error of the fitting line. **k**, Spearman's correlation (ρ) of effect sizes between *cis*-eQTL and matched ASE loci across 34 tissues. Red dots indicate number of tested loci (right *y*-axis).

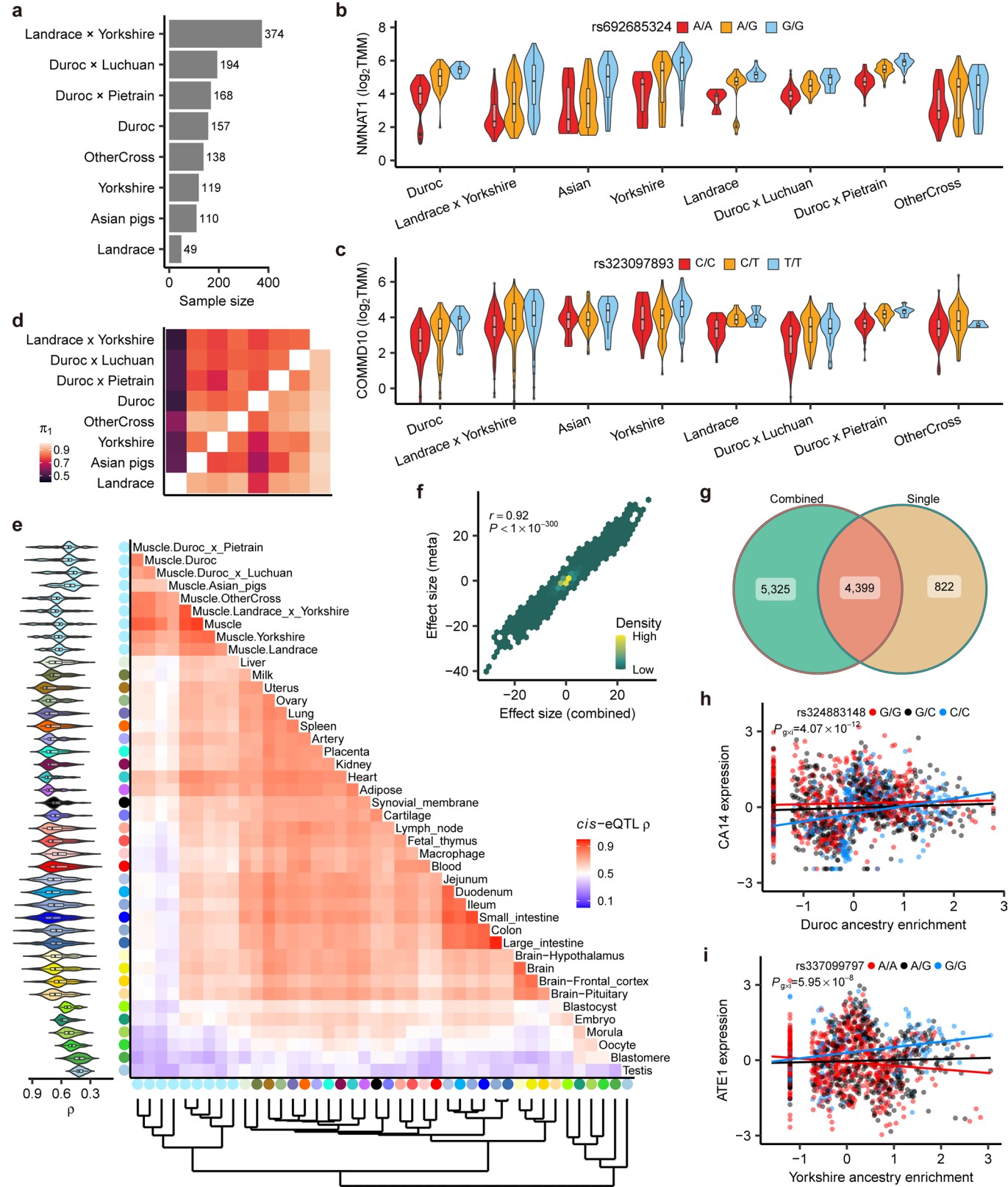

**Extended Data Fig. 7 | See next page for caption.**

**Extended Data Fig. 7 | Breed sharing and interaction *cis*-eQTL (bieQTL).**
**a**, Sample size of muscle RNA-Seq data across eight breed groups. **b**,**c**, Expression levels of *NMNAT1* (**b**) and *COMMD10* (**c**) at three genotypes of *cis*-eQTL in muscle across eight breed groups. **d**, The *cis*-eQTL discovered in each breed group (rows) that can be replicated ($\pi_1$) across all other breed groups (columns). **e**, The heatmap of tissues regarding the pairwise Spearman's correlation (ρ) of *cis*-eQTL effect sizes. Tissues are grouped by hierarchical clustering (bottom). Violin plot (left) represents Spearman's correlation between the target group and the rest. **f**, Pearson's correlation (*r*) of effect size between *cis*-eQTL from the multi-breed

meta-analysis (*y*-axis) and those from the combined muscle population (*x*-axis). The *P* value was obtained from Pearson's *r* test. **g**, Overlap of *cis*-eQTL detected from the combined muscle population (Combined) and those detected in single-breed (Single) *cis*-eQTL mapping (shared in at least two breeds). **h**,**i**, Examples of bieQTL in muscle. Each dot in (**h**, *CA14*) and (**i**, *ATE1*) represents an individual and is colored by three genotypes. Gene expression levels and ancestry enrichment scores are inverse normal transformed. The two-sided *P* value is calculated by the linear regression bieQTL model. The lines are fitted by a linear regression model using the *geom_smooth* function from ggplot2 (v3.3.2) in R (v4.0.2).

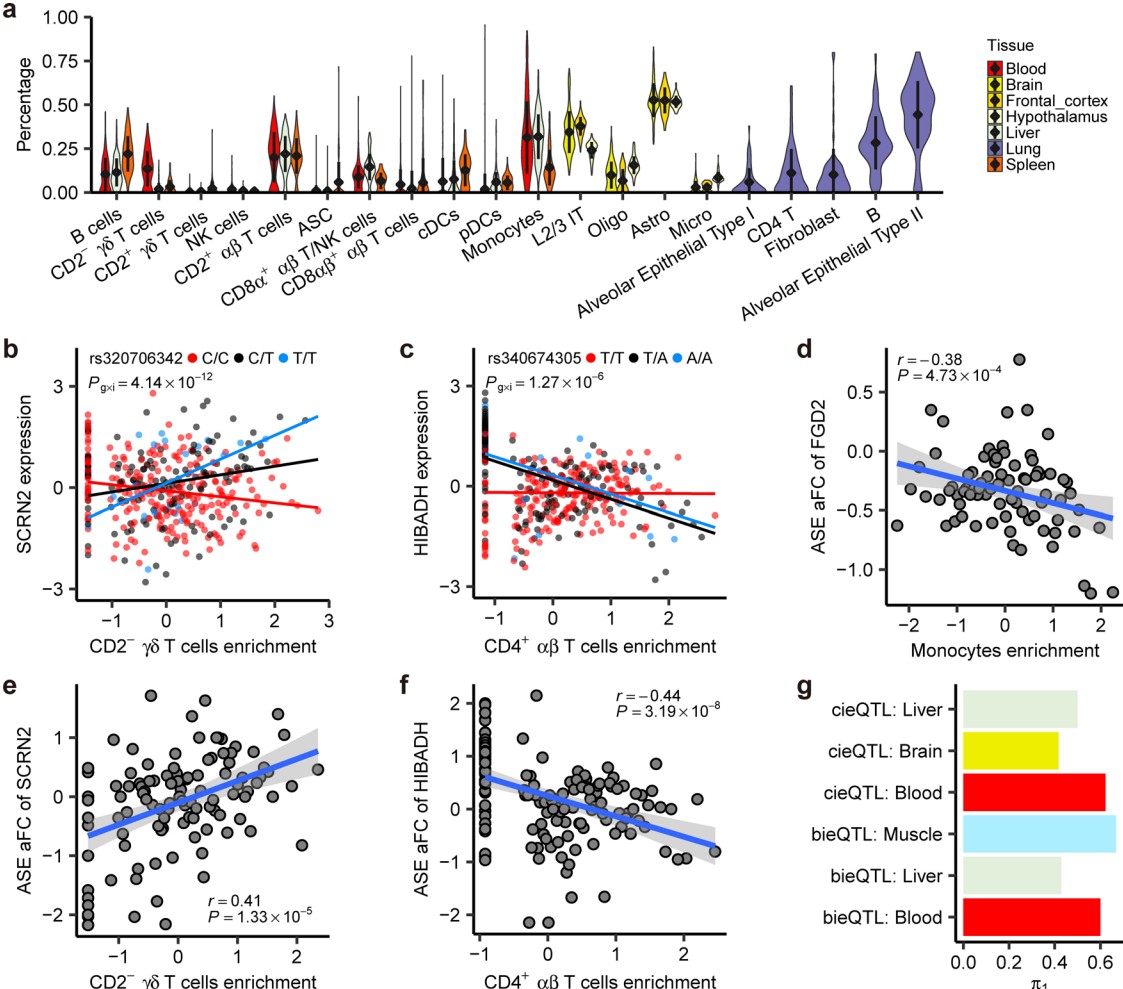

**Extended Data Fig. 8 | Cell-type enrichment and interaction cis-eQTL (cieQTL). a**, Distribution of enrichment scores (percentage) of major cell types in samples of seven tested tissues (brain: n = 415, frontal cortex: n = 75, hypothalamus: n = 73, lung: n = 149, blood: n = 386, liver: n = 501, and spleen: n = 91). Each point and whisker indicate the mean value and standard deviation, respectively. **b,c**, Examples of cieQTL in blood. Each dot in (**b**, *SCRN2*) and (**c**, *HIBADH*) represents an individual and is colored by three genotypes. Gene expression levels and cell-type enrichment scores are inverse normal transformed. The two-sided *P* value was calculated by the linear regression cieQTL model. The lines are fitted by a linear regression model using the *geom_ smooth* function from ggplot2 (v3.3.2) in R (v4.0.2). **d–f**, Pearson's correlation (*r*) between allele-specific expression (ASE) effect sizes (allelic fold change, aFC) and specific cell-type enrichment scores for *FGD2* with monocytes (**d**), *SCRN2* with CD2⁻ γδ T cells (**e**) and *HIBADH* with CD4⁺ αβ T cells in the blood (**f**). The lines are fitted by a linear regression model using the *geom_smooth* function from ggplot2 (v3.3.2) in R (v4.0.2). The shading represents the standard error of the fitting line. **g**, ASE validation rate ($\pi_1$) of breed/cell-type interaction QTL (bieQTL and cieQTL) across tissues with ≥ 5 detectable bieQTL or cieQTL.

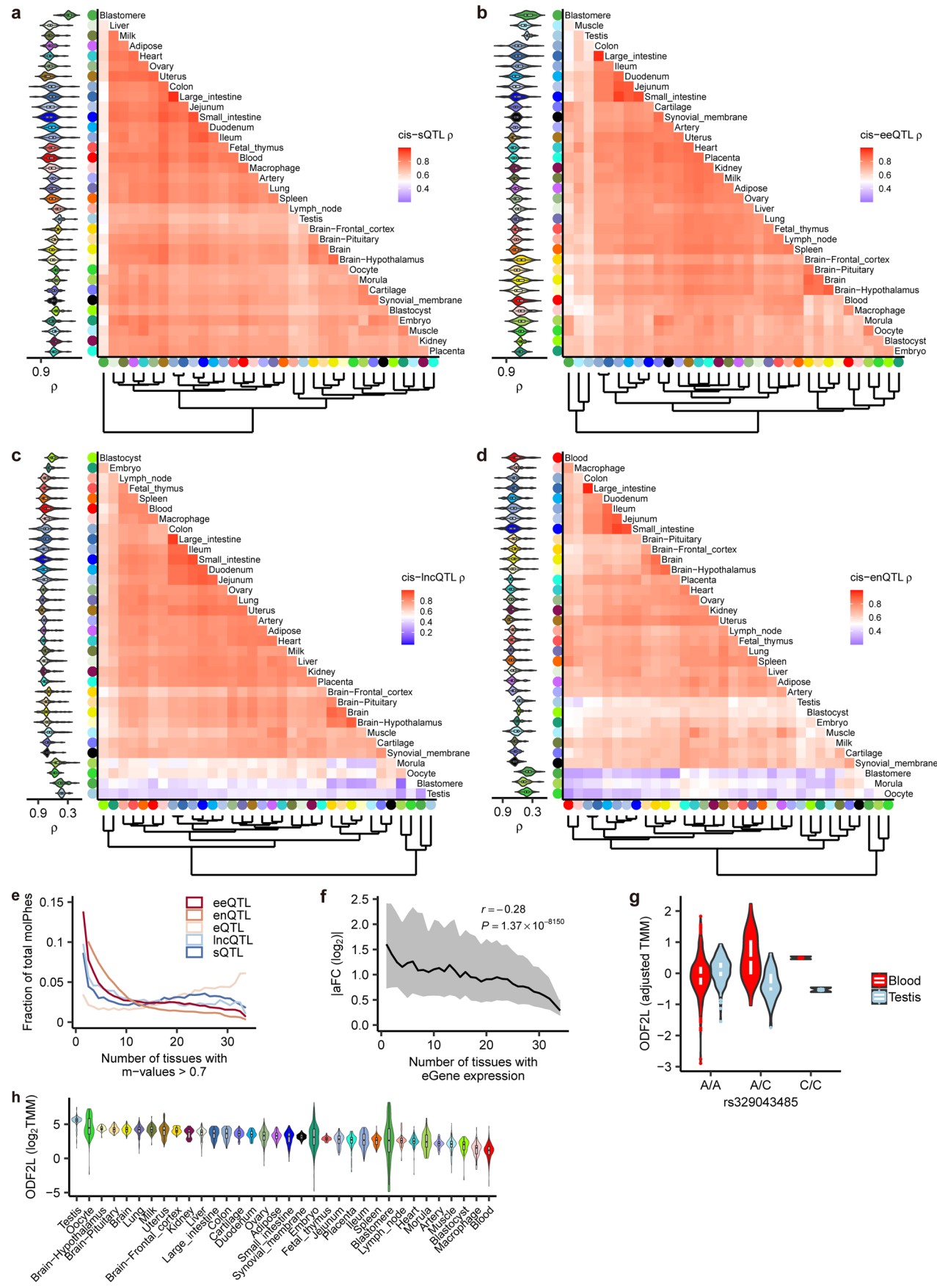

**Extended Data Fig. 9 | See next page for caption.**

**Extended Data Fig. 9 | Tissue-sharing and specificity patterns of molecular QTL (molQTL). a**–**d**, The heatmap of tissues regarding the pairwise Spearman's correlation (ρ) of molQTL effect sizes, that is, *cis*-sQTL (**a**), *cis*-eeQTL (**b**), *cis*-lncQTL (**c**) and *cis*-enQTL (**d**). Tissues are grouped by the hierarchical clustering (bottom). Violin plot (left) represents Spearman's correlations between the target tissue and the rest. **e**, Distribution of number of tissues having METASOFT activity (m-value > 0.7) for each of molQTL. MolPhe: molecular phenotype. **f**, Pearson's correlation (*r*) between number of tissues an eGene expressed in

(transcript per million, TPM > 0.1) and its *cis*-eQTL effect sizes (|aFC(log$_2$)|). The aFC is for allelic fold change. The line and shading indicate the median and interquartile range, respectively. **g**, Expression levels (adjusted TMM) of *ODF2L* at three genotypes of top *cis*-eQTL (rs329043485) in blood and testis. TMM: trimmed mean of M-value normalized expression levels. There are 337, 47 and 2 samples for A/A, A/C and C/C genotypes in blood, respectively, and 148, 34 and 2 in testis, respectively. **h**, Expression levels (log$_2$TMM) of *ODF2L* across 34 tissues. Tissues are ordered (from smallest to largest) by the median expression values.

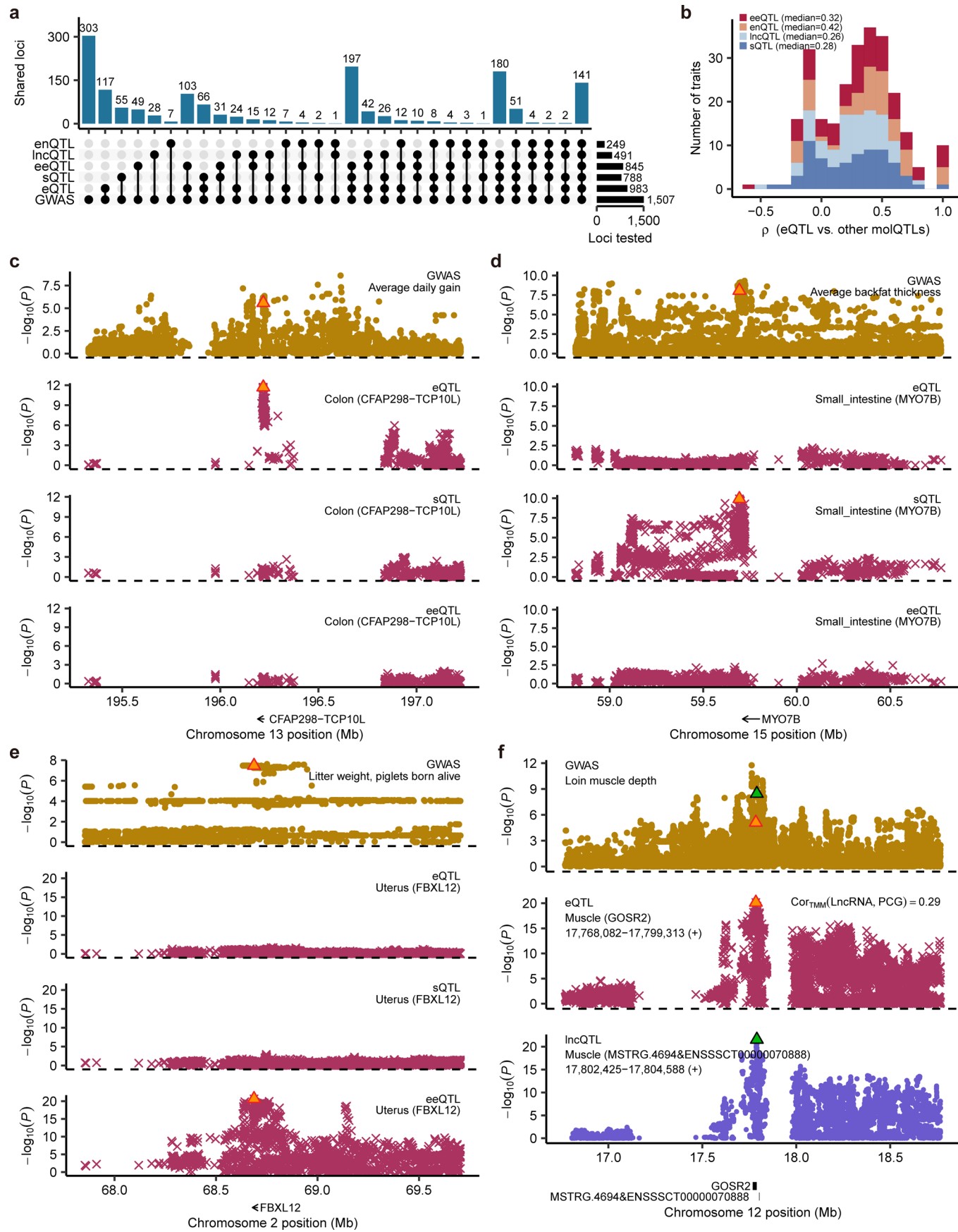

**Extended Data Fig. 10 | See next page for caption.**

**Extended Data Fig. 10 | Complementarity of molecular QTL (molQTL) in interpreting GWAS loci. a**, Number of GWAS loci linked to *cis*-eQTL, *cis*-sQTL, *cis*-eeQTL, *cis*-lncQTL and *cis*-enQTL in 34 tissues based on four different integrative methods, including colocalization (fastEnloc), Mendelian randomization (SMR), single-tissue transcriptome-wide association studies (TWAS, S-PrediXcan) and multi-tissue TWAS (S-MultiXcan). The bottom point-line combinations of the Upset plot represent the intersections of GWAS loci linked to eGenes by different types of molecular phenotypes. **b**, Distribution of rank correlations between tissue-relevance-scores derived from *cis*-eQTL and those from *cis*-sQTL, *cis*-lncQTL, *cis*-eeQTL and *cis*-enQTL across 86 GWAS traits with significant colocalizations for at least one molecular phenotype. **c**, Significant SMR signals ($P_{SMR} = 9.16 \times 10^{-5}$, $P_{HEIDI} = 0.9$) between GWAS loci of average daily gain (ADG) and *cis*-eQTL of *CFAP298-TCP10L* in colon, but not for its *cis*-sQTL or *cis*-eeQTL. The orange triangle represents the top *cis*-eQTL of *CFAP298-TCP10L*. **d**, Significant

SMR signals ($P_{SMR} = 1.78 \times 10^{-5}$, $P_{HEIDI} = 0.07$) between GWAS loci of the average backfat thickness (BFT) and *cis*-sQTL of *MYO7B* in the small intestine, but not for its *cis*-eQTL or *cis*-eeQTL. **e**, Significant SMR signals ($P_{SMR} = 1.78 \times 10^{-6}$, $P_{HEIDI} = 0.97$) between GWAS loci of litter weight (LW, piglets born alive) and *cis*-eeQTL of *FBXL12* in the uterus, but not for its *cis*-eQTL or *cis*-sQTL. **f**, Significant SMR signals ($P_{SMR(lncQTL-GWAS)} = 4.49 \times 10^{-7}$, $P_{SMR(eQTL-GWAS)} = 5.45 \times 10^{-5}$, $P_{SMR(lncQTL-eQTL)} = 4.62 \times 10^{-7}$) among GWAS loci of loin muscle depth (LMD), *cis*-lncQTL of *MSTRG.4694&ENSSSCT00000070888*, and *cis*-eQTL of *GOSR2* in the muscle. *MSTRG.4694&ENSSSCT00000070888* is a lncRNA gene located on the 3112 bp downstream of *GOSR2*, where the Pearson's correlation of their normalized expression levels (trimmed mean of M-value, TMM) is 0.29 in muscle. The orange and green triangles in the top GWAS Manhattan plot represent the top molQTL of *GOSR2* and *MSTRG.4694&ENSSSCT00000070888*, respectively.

# Reporting Summary

## Statistics

For all statistical analyses, confirm that the following items are present in the figure legend, table legend, main text, or Methods section.

| n/a | Confirmed | |
|---|---|---|
| ☐ | ☒ | The exact sample size (*n*) for each experimental group/condition, given as a discrete number and unit of measurement |
| ☐ | ☒ | A statement on whether measurements were taken from distinct samples or whether the same sample was measured repeatedly |
| ☐ | ☒ | The statistical test(s) used AND whether they are one- or two-sided *Only common tests should be described solely by name; describe more complex techniques in the Methods section.* |
| ☐ | ☒ | A description of all covariates tested |
| ☐ | ☒ | A description of any assumptions or corrections, such as tests of normality and adjustment for multiple comparisons |
| ☐ | ☒ | A full description of the statistical parameters including central tendency (e.g. means) or other basic estimates (e.g. regression coefficient) AND variation (e.g. standard deviation) or associated estimates of uncertainty (e.g. confidence intervals) |
| ☐ | ☒ | For null hypothesis testing, the test statistic (e.g. *F*, *t*, *r*) with confidence intervals, effect sizes, degrees of freedom and *P* value noted *Give P values as exact values whenever suitable.* |
| ☐ | ☒ | For Bayesian analysis, information on the choice of priors and Markov chain Monte Carlo settings |
| ☐ | ☒ | For hierarchical and complex designs, identification of the appropriate level for tests and full reporting of outcomes |
| ☐ | ☒ | Estimates of effect sizes (e.g. Cohen's *d*, Pearson's *r*), indicating how they were calculated |

*Our web collection on statistics for biologists contains articles on many of the points above.*

## Software and code

Policy information about availability of computer code

| Data collection | All raw data analyzed in this study are publicly available for download without restrictions from SRA (https://www.ncbi.nlm.nih.gov/sra/) and BIGD (https://bigd.big.ac.cn/bioproject/) databases using the wget function in Linux. Details of RNA-Seq, WGS, WGBS, single-cell RNA-Seq and Hi-C datasets can be found in Supplementary Table 1, 2, 5, 8 and 9, respectively. |
|---|---|
| Data analysis | All the computational scripts and codes (with software version) for RNA-Seq, WGS, WGBS, single-cell RNA-Seq and Hi-C datasets analyses, as well as the respective quality control, molecular phenotype normalization, genotype imputation, molQTL mapping, functional enrichment, colocalization, SMR and TWAS are available at the FarmGTEx GitHub website (https://github.com/FarmGTEx/PigGTEx-Pipeline-v0, https://doi.org/10.6084/m9.figshare.24247771).<br><br>For RNA-Seq data analysis, we used Trimmomatic (v0.39), STAR (v2.7.0), Stringtie (v2.1.1), featureCounts (v1.5.2), Leafcutter (v0.2.9), GATK (v4.0.8.1), phASER (v1.1.1), and Beagle (v5.1) for quality control, mapping, gene expression quantification, alternative splicing, SNP calling, ASE analysis, and genotype imputation, respectively. For sample clustering, we used MEGA (vX) and then visualized with iTOL (v6). For tissue-specific gene expression, we used limma (v3.51.2). For gene co-expression analysis, we used WGCNA (v1.69), ICA (v1.0.2), PEER (v1.3), MEGENA (v1.3.7), and CEMiTool (v1.8.3). For gene functional enrichment analysis, we used clusterProfiler (v4.0) and visualized it using Gephi (v0.9.2).<br><br>For WGS analysis, we used Trimmomatic (v0.39), BWA-MEM (v0.7.5a-r405), Picard (v2.21.2), GATK (v4.1.4.1), and Beagle (v5.1) for quality control, mapping, marked duplicated reads, variants calling, and phasing, respectively. We used used PLINK (v1.90) to do LD pruning.<br><br>For WGBS, we used FastQC (v0.11.9), Trim Galore (v0.4.5), Bismark (v0.19.0), and SMART2 (v2.2.8), Methpipe (v4.1.1), and FastQTL (v2.184) for quality evaluation, quality control, read mapping and DNA methylation level extraction, hypomethylation region detection, allele-specific |

methylation loci analysis and methylation QTL mapping, respectively.

For Hi-C, we used Trim Galore (v0.6.7), BWA (v0.7.17), Juicer (v1.6), Arrowhead (v1.22.01), hicConvertFormat (v3.7.1), pyGenomeTracks (v3.6) for quality control, read mapping, Hi-C contact matrix construction, TAD identification, format conversion, and visualization, respectively.

For single-cell RNA-Seq, we used Seurat (v3.0.2), Azimuth (v0.4.0) and CIBERSORTx online tool (v1) for data processing, cell type annotation and cell type deconvolution, respectively.

We removed SNPs with MAF < 0.01 and/or missing rate > 0.9 using bcftools (v1.9) and employed Beagle (v5.1) to phase the filtered variants and impute sporadically missing genotypes.

For QTL mapping, we used TensorQTL (v1.0.3), aFC (v0.3), dap-g (v1.0.0), METASOFT (v2.0.1), MashR (v0.2-6), and GCTA (v1.93.0) for cis-QTL mapping, effect size estimation, fine-mapping, meta-analysis, tissue-sharing pattern estimation, and cis-QTL mapping with mixed linear model, respectively. We estimated the genetic parameters using the restricted maximum likelihood (REML) method implemented in GCTA (v1.93.0).

We computed genotype PCs based on the filtered SNPs within each of the tissues using SNPRelate (v1.26.0). To account for technical confounders among RNA-Seq samples (e.g., hidden batch effects and other technical or biological factors), we used the Probabilistic Estimation of Expression Residuals (PEER) method, implemented in peer R package (v4.0.2), to estimate a set of latent covariates within each of the 34 tissues based on gene expression matrices. We computed the mappability of each locus in the reference genome using GenMap (v1.3.0). We removed SNPs in repeat regions annotated by the UCSC RepeatMasker track.

We first used imputed genotypes to estimate the ancestry composition of all RNA-Seq samples across tissues using ADMIXTURE (v1.3.0). we estimated the effect size (aFC) of the top ieQTL of ieGenes from ASE data using the script phaser_cis_var.py in phASER (v1.1.1)

For integrative analysis between GWAS and molQTL, we used S-PrediXcan and S-MultiXcan in MetaXcan (v0.6.11) for single-tissue and multi-tissue TWAS analysis, SMR (v1.03) for Mendelian Randomization analysis, and fastENLOC (v1.0) for colocalization. We performed a meta-analysis of molQTL across all 34 tissues using MashR (v0.2-6) and METASOFT (v2.0.1). We calculated the pairwise Rand index to measure the clustering similarity using the rand.index function in the fossil (v0.4.0) R package (v4.0.2).

we performed 2,056 separate GWAS, and conducted the meta-GWAS analysis for the same traits across different populations based on GWAS summary statistics using METAL (v2011-03-25).

For enrichment analysis, we used TORUS (v1) and ClusterProfiler (v4.0) for molQTLs and genes functional annotation, respectively.

For manuscripts utilizing custom algorithms or software that are central to the research but not yet described in published literature, software must be made available to editors and reviewers. We strongly encourage code deposition in a community repository (e.g. GitHub). See the Nature Portfolio guidelines for submitting code & software for further information.

# Data

Policy information about availability of data

All manuscripts must include a data availability statement. This statement should provide the following information, where applicable:
- Accession codes, unique identifiers, or web links for publicly available datasets
- A description of any restrictions on data availability
- For clinical datasets or third party data, please ensure that the statement adheres to our policy

All raw data analyzed in this study are publicly available for download without restrictions from SRA (https://www.ncbi.nlm.nih.gov/sra/) and BIGD (https://bigd.big.ac.cn/bioproject/) databases. Details of RNA-Seq, WGS, WGBS, single-cell RNA-Seq and Hi-C datasets can be found in Supplementary Tables 1, 2, 5, 8 and 9, respectively. All WGS data generated in this study are available under CNCB GSA (https://ngdc.cncb.ac.cn/) accessions: PRJCA016120, PRJCA016130, PRJCA017284, PRJCA016012, and PRJCA016216. All processed data and the full summary statistics of molQTL mapping are available at http://piggtex.farmgtex.org/.

# Human research participants

Policy information about studies involving human research participants and Sex and Gender in Research.

| Reporting on sex and gender | NA |
| --- | --- |
| Population characteristics | NA |
| Recruitment | NA |
| Ethics oversight | NA |

Note that full information on the approval of the study protocol must also be provided in the manuscript.

# Field-specific reporting

Please select the one below that is the best fit for your research. If you are not sure, read the appropriate sections before making your selection.

☒ Life sciences ☐ Behavioural & social sciences ☐ Ecological, evolutionary & environmental sciences

For a reference copy of the document with all sections, see nature.com/documents/nr-reporting-summary-flat.pdf

# Life sciences study design

All studies must disclose on these points even when the disclosure is negative.

| | |
|---|---|
| Sample size | No power calculation was needed in advance in this study. In total, we analyzed all 11,323 RNA-Seq runs (downloaded by March, 2021) from SRA (https://www.ncbi.nlm.nih.gov/sra/), and BIGD databases (https://bigd.big.ac.cn/bioproject/), yielding 9,530 unique RNA-Seq samples. After filtering the samples with low quality (see below), all the remaining samples have been used for analysis. |
| Data exclusions | Full details of data exclusions for each analysis can be found in the Methods as well.<br><br>We filtered out RNA-Seq samples with clean read counts ≤ 500K or uniquely mapping rates < 60%, resulting in 8,262 samples. We further excluded samples with obvious clustering errors (e.g., samples labeled as liver that were not clustered with other liver samples), resulting in 7,095 samples for subsequent analysis.<br><br>For cis-QTLs detection, we excluded tissues with less than 40 individuals, resulting in 34 tissues for cis-QTL mapping. |
| Replication | To validate the cis-eQTLs, we applied four distinct strategies including linear mixed model, internal validation, external validation, and ASE validation.<br><br>First, we observed that the summary statistics of cis-eQTL derived from the linear regression model in TensorQTL had a strong correlation (an average Pearson's r of 0.91 across tissues) with those from a linear mixed model.<br><br>Second, we performed an internal validation in 18 tissues with over 80 samples by randomly dividing samples into two equal groups, and then conducting cis-eQTL mapping separately in both subgroups. We observed a high replication rate (an average $\pi 1$ of 0.92) for cis-eQTL discovery.<br><br>Third, we found that 92%, 74%, 73%, and 69% of cis-eQTL in blood, liver, duodenum, and muscle, respectively, were replicated in independent datasets.<br><br>Fourth, we further found that effects (allelic fold changes, aFC) derived from allele specific expression (ASE) analysis were significantly correlated with those from cis-eQTL mapping consistently across tissues. For instance, in muscle, ASE-derived effects of 4,417 SNPs were significantly correlated (Spearman's $\rho$ = 0.76, P < 1e-300) with their cis-eQTL effects. |
| Randomization | All the datasets are from observation studies and we used all samples publicly available after data exclusions listed above. Therefore, Randomization were not relevant in this study. Samples were grouped by tissue types. |
| Blinding | In this study, we re-analyzed all the publicly available RNA-seq data using a uniform pipeline, followed by the population-based association studies and validated the findings in independent populations. The blinding study design may be not applicable in this study. |

# Reporting for specific materials, systems and methods

We require information from authors about some types of materials, experimental systems and methods used in many studies. Here, indicate whether each material, system or method listed is relevant to your study. If you are not sure if a list item applies to your research, read the appropriate section before selecting a response.

## Materials & experimental systems

| n/a | Involved in the study |
|---|---|
| ☒ | Antibodies |
| ☒ | Eukaryotic cell lines |
| ☒ | Palaeontology and archaeology |
| ☒ | Animals and other organisms |
| ☒ | Clinical data |
| ☒ | Dual use research of concern |

## Methods

| n/a | Involved in the study |
|---|---|
| ☒ | ChIP-seq |
| ☒ | Flow cytometry |
| ☒ | MRI-based neuroimaging |

