## [Peer Review File · Nature Genetics]

Peer Review Information

Manuscript Title: A compendium of genetic regulatory effects across pig tissues

Corresponding author name(s): Dr Lingzhao Fang Professor George (E.) Liu Professor Zhe Zhang Professor Kui Li Professor Albert Tenesa

Editorial Notes:

Transferred manuscripts This document only contains reviewer comments, rebuttal and decision letters for versions considered at Nature Genetics.

Reviewer Comments & Decisions:

Decision Letter, initial version:
--

10th Jan 2023

Dear Lingzhao,

Happy New Year! I hope you've had a good break.

Your Article, "A compendium of genetic regulatory effects across pig tissues" has now been seen by 3 referees. You will see from their comments copied below that while they find your work of considerable potential interest, they have raised quite substantial concerns that must be addressed. In light of these comments, we cannot accept the manuscript for publication, but would be very interested in considering a revised version that addresses these serious concerns.

Overall, the three referees appreciate the aims of pig GTEx but are divided when it comes to the overall novelty and impact.

Reviewer #1 thinks that the novelty falls short, and also suggests there may be unresolved technical issues. They also think that the human-pig cross-species comparison is problematic.

Reviewer #2 is more positive, but notes a range of technical concerns also, some of which overlap with Reviewer #1's comments. Most importantly, they request substantial improvements to data availability.

Reviewer #3 is straightforwardly supportive, and has only minor comments.

We believe that there is a path to publication but it is clear that substantial revisions are needed. There is useful and specific guidance for improvement. We think that the technical comments are all

reasonable and must be fully addressed, particularly the potential for batch effects. We also think that the novelty would have to be improved to persuade Reviewer #1 to be supportive; there are a few suggestions for new analyses that could do so, and we would also encourage you and your co-authors to consider further ideas as well.

We hope you will find the referees' comments useful as you decide how to proceed. If you wish to submit a substantially revised manuscript, please bear in mind that we will be reluctant to approach the referees again in the absence of major revisions.

To guide the scope of the revisions, the editors discuss the referee reports in detail within the team, including with the chief editor, with a view to identifying key priorities that should be addressed in revision and sometimes overruling referee requests that are deemed beyond the scope of the current study. We hope that you will find the prioritised set of referee points to be useful when revising your study. Please do not hesitate to get in touch if you would like to discuss these issues further.

If you choose to revise your manuscript taking into account all reviewer and editor comments, please highlight all changes in the manuscript text file. At this stage we will need you to upload a copy of the manuscript in MS Word .docx or similar editable format.

*2) If you have not done so already please begin to revise your manuscript so that it conforms to our Article format instructions, available [here](http://www.nature.com/ng/authors/article_types/index.html). Refer also to any guidelines provided in this letter.

[redacted]

If you wish to submit a suitably revised manuscript we would hope to receive it within 6 months. If you cannot send it within this time, please let us know. We will be happy to consider your revision so long as nothing similar has been accepted for publication at Nature Genetics or published elsewhere. Should your manuscript be substantially delayed without notifying us in advance and your article is eventually published, the received date would be that of the revised, not the original, version.

Thank you for the opportunity to review your work.

Sincerely,

Michael Fletcher, PhD
Senior Editor, Nature Genetics

ORCID: 0000-0003-1589-7087

Referee expertise:

Referees #1, #3: livestock, including pig, genetics.

Referee #2: computational genetics; QTLs.

Reviewers' Comments:

Reviewer #1:

Remarks to the Author:

This is a GTEx type study in pigs where the RNA-Seq data are collected from the SRA, which is the bulk of the data and analyses. The paper also includes additional datasets such as bisulfite sequencing, single cell RNA-Seq and hi-C, also from public sources. A key resource in making this study possible is public WGS datasets so variants called from RNA-Seq (potentially other datasets as

well) can be imputed to obtain sequence level genotypes. This allows the authors to map QTLs (molQTLs) for different omic measurements. Typical analyses then follow, including mapping of regulatory variants, enrichment analysis, integrative analysis of various kinds. The results are more or less expected. The same research groups have previously published in the same journal similar work in cattle.

The analyses are careful and I have a few technical comments below that need to be addressed. The paper is mostly analysis work and strikes me as a bit superficial. There is a large amount of data analyzed and the authors try hard to be comprehensive. However, the analysis offers relatively few previously unknown insights and some of the claims aren't fully supported by the analysis (e.g. using the pig as a model for complex traits in humans).

Major comments:

1) All analyses in the paper rely on accurate imputation by RNA-Seq variants. I would like to see more comprehensive characterization of imputation accuracy. At the minimum, please provide a) imputation accuracy for all molQTLs in an independent population (individuals used to assess accuracy not in PGRP); b) imputation accuracy based on not just concordance, but also r^2 ; c) clarification that the imputation accuracy reported in extended data figure 1 is based on individuals that are not present in the PGRP. d) when summarizing imputation accuracy, do one for the entire variant set and do another one for common variants ($MAF > 0.05$) only.

2) It appears that the expression estimates used to cluster gene expression (section starting line 242) are not adjusted for covariates (known or PEER). The method describes adjustments later in the mapping section. This is important, any clustering should be performed post batch effect adjustment. Same goes of the cis- h^2 analysis, please clarify if the expression was adjusted for unwanted latent confounding factors.

3) All subsequent analyses following molQTL mapping rely on proper control of inference errors. The analyses as presented adjusted for covariates identified by PEER and genotype PCs, etc. I would like to see a few more pieces of information including a) histogram summarizing correlation between raw and adjusted gene expression; b) a few randomly chosen examples of QQ plots for molQTL mapping to show that the adjustment was adequate. With the small sample size, 86% (!) of all genes tested contain eQTLs. This is a remarkably high proportion even when combined over all tissues and deserves a bit more scrutiny. Put this in the context of the cis- h^2 estimates (mean = 0.14) and I'm really worried about the control of FDR. Figure 2a says very little, even if everything is false discovery, you would still expect to see positive correlation. I'd like to see more analyses done here to help me gauge the level of false discovery in this particular part of analysis. The authors would have to think more and harder about it here but one idea is to perform permutations for some genes to see if a) the adjustment was adequate; b) the FDR reported by tensorQTL is adequate.

4) Line 305: it is good that the tensorQTL estimates are checked against mixed model based estimates. My question is why there are multiple correlation from the same tissue? Is it because the analysis was performed in different datasets (breeds)? Although the average of the correlation was high, it makes me worried that it can go as low as negative. It should be checked to find out why some have low correlation. You would expect if everything is adjusted (especially relatedness) well, these two should yield very similar results (definitely higher than 0.9). In addition, correlation masks the effect of a baseline shift. For example, tensorQTL may report in general lower P value and you

would still observe positive correlation, but that makes FDR estimates much more trickier.

5) Line 401-403: the fact that there are enrichments for these exonic annotations, much stronger than up/downstream makes me worried that there is mapping bias. In other words, the gene expression estimation is influenced by SNPs present in exons.

6) Line 439-446: I wonder if randomly chosen SNPs matched for the molQTLs in terms of MAF would explain similar amounts of heritability given that pigs have high LD.

7) I find the whole section of finding shared genes for complex traits between humans and pigs rather superficial and I will explain why it's not a good idea to do this. It's completely expected that the tissue specificity is conserved (6a). But conservation as measured by PhastCons and LOEUF differing between the different classes of genes based on eGene sharing can be due to a number of reasons, the most trivial one being that eGenes or not is influenced by expression level in general. To say that this is essential functions and purifying selection is an overstretch. The correlation between effect sizes in the two species is very weak (6d), could be even weaker if one variant (lower left corner) is taken out. Lastly, for the correlations between TWAS effect sizes, it's easy to pick among hundreds of such correlations some that have good correlations (6f-h). It's probably hard to perform such analysis, but it would be nice if some "unrelated" traits between the two species can be used as a negative control to see what would have been expected under null. In fact figure 6f would have been a good negative control where high correlation shows up.

Minor comments:

Line 218: I wonder what enhancer expression means. Even if reads map to enhancer regions, it does not mean enhancers are generally transcribed. They can be part of transcripts in terms of genomic coordinates, but their functions are not exerted through transcription. I find it odd to quantify enhancer expression.

Line 218 + Line 1093-1102: It appears that the PCGs and the lncRNAs are normalized separately. This needs justification. They are both transcribed and should be considered in the same pool.

Line 238: unless these are multi-omic data collected from the same samples, call them other omics datasets.

Line 274: I don't think it make sense to cluster h^2 estimates. I have a hard time envisioning a mechanism where the h^2 would cluster based on tissues. What you are clustering may be just expression levels (i.e., genes expressed at higher level tend to have higher h^2).

Line 316: Validation by ASE, this is good. However, looking at extended data figure 6e and f, the majority of the points are around 0 (density high), why is that? These are cis-eQTL to begin with, shouldn't they be mostly farther away from zero?

Figure1: this is not "phylogenetic" clustering, the same term was used throughout the manuscript, please consider revising.

Reviewer #2:

Remarks to the Author:

Comments for NG-A61417-T

Comments for the Author:

In this manuscript, Teng et al. built the most comprehensive Pig Genotype-Tissue Expression Atlas, which contains 1,602 whole-genome sequencing and 9,530 RNA-sequencing samples from multiple pig tissues. The authors also did QTL analyses for five molecular phenotypes (e.g., eQTLs and sQTLs) and compared PigGTEx with the human GTEx data. Overall, the majority of performed analyses are generally convincing and the resources of this study will be widely useful to the community. I have some suggestions to make the PigGTEx more convenient and user-friendly. My detailed questions are listed as follows.

Major concerns:

1. The PigGTEx has 9,530 RNA-Seq samples. After filtering out samples of low quality, only 7,095 RNA-Seq samples were used in the analysis. The authors should provide phenotype information of all RNA-Seq samples in the download page of the website and label which sample were used in the QTL analysis. Do all RNA-Seq samples have matched WGS data? If not, please label which sample have, which not. Please also provide a meta file in the download page to map the RNA-seq ID and WGS ID. The authors also should provide the raw count tables for gene, exon, lncRNA and enhancer and unnormalized alternative splicing values. Confounders are very important for QTL analysis, the authors should provide the values of both known confounders and unknown PEER confounders for each sample in each tissue type in the download page as human GTEx did. Colocalization analyses with GWAS need both significant and non-significant QTLs. Please also provide full summary statistics of QTL results (not only significant items) for users in the download page of the website.
2. The human GTEx use different number of PEER factors for different tissues based on the sample size. Why the authors used the top ten PEER factors for all tissue types? I also found many unknown age/sex items for RNA-Seq samples in Table S1. Did the authors include age/sex/breed as confounders? If yes, how did the authors deal with the unknown items? If not, please clarify or add.
3. Current QTL tissue-sharing patterns are not convincing. For example, Pituitary is more similar as Uterus than Frontal cortex in Fig. 3a. Another example, Muscle is more similar as Adipose than Heart in Extended Data Figure 4. Please clarify. The colors in heatmaps are very similar, which makes it hard to read (e.g., Fig. 3a and b, Extended Fig. 4b, Extended Fig. 7e and Extended Data Figure 9). Please reassign heatmap color range to make it clear.
4. eQTL analysis always include both Protein-coding genes and lncRNAs (such as human GTEx and Brain xQTLServe). It's so strange to split eQTL into PCG cis-eQTL and cis-lncQTL. Please clarify or combine.
5. Trans-QTLs is an important part of QTL studies. Please add trans-QTL analyses for tissue types (such as muscle and brain) which have enough samples ($n \geq 150$).
6. The authors could do fine mapping (e.g. with eCAVIAR, FINEMAP, or SUSIER) of the QTL signal to strengthen their colocalization result (such as the ABCD4 example in Fig. 5i) of QTL and GWAS signal actually overlapping by identifying which exact SNPs that are potentially causal. I also wandering if

SNPs with high fine mapping causal probability will be enriched in sequence ontology in Fig. 4a (just like Fig. 3c-d in PMID 33986536).

7. The TWAS-server on the PigGTEx-Portal does not work. The authors should also provide TWAS results and TWAS trained models (users can use these predictive models in their own studies) in the download webpage.
8. It's not convenient for users to explore QTLs without a genome browser. Please add a IGV Browser like the human GTEx cohort (<https://gtexportal.org/home/browseEqtls>) using IGV.js (<https://github.com/igvteam/igv.js/>).
9. Did the authors consider batch effects of RNA-seq data? Please clarify or add.
10. The author did a good job to compare the similarity of pigs with humans in gene expression and the genetic regulation. Could the author also compare it between pigs and cattles?
11. Are there any trait-associated GWAS loci colocalized with two or more molecular QTLs (eQTL, sQTL, eeQTL, enQTL and lncQTL)? Please give some examples.

Minor concerns:

1. In all figures (such as Fig. 1d), Frontal cortex and Hypothalamus should change to Brain-Frontal cortex and Brain-Hypothalamus (or other suitable names).
2. Colors in Fig. 2a are very similar, please change to make it clear.
3. In Fig. S1, what are the Q1, Q2, Q3 and Q4? Please clarify.
4. There is no Biosample ID or BioProject ID information for 509 new WGS data on Table S2.
5. In Fig. S10 and extended Fig. 5, please add figure legend. In addition, for Fig. S10, could you add another graph (like, boxplot with p-value) to clearly present the difference of spatial distribution across 34 tissues.
6. Fig.5C, Fig. S8 and Fig. S11 are confusing. Such as, Fig.5C, GWAS detected 524 GEAS loci linked to eGenes, the number drops to 1 when fastENlic and S-prediXcan are included. However, the number increases to 225 when all 5 methods are included.
7. The RNA-seq data includes pair-end or single-end and different platforms. The description of RNA-seq trimming with "TruSeq3-PE.fa:2:30:10" is incorrect. For single-end RNA-seq data, the parameter should be "TruSeq2-SE".
8. For extended Fig.8, authors found the variant of cell type composition across bulk tissue samples by a cell-type deconvolution analysis. It's better to draw all the different type of tissue on one diagram to clearly observe cell of origin. In addition, authors should further explain the difference of cell contribution in different tissues.
9. In this manuscript, authors made a lot of analysis and obtained a lot of conclusions. But in some parts, there is no literature or further verification to support conclusions. For example, the gene EMG1 (Fig. 2i), ODF2L (Fig. 3h) and so on, why did authors select these genes for illustration? Are they the most significant? Or have they been reported?

Reviewer #3:

Remarks to the Author:

This manuscript presents the main results of a massive collective effort that is the pilot phase of the Pig Genotype-Tissue Expression atlas (PigGTE_x). PigGTE_x provides a very useful resource for pig genomics research, in particular on gene expression and regulatory effects that underlie complex traits of economic and social importance. The manuscript is very well-written, clear, well-structured, and with suitable references. Results are succinctly described and deeply detailed in the corresponding figures and tables in an appropriate fashion. I only found some small typos here and there, especially in the supplementary files, of which I flag a few below (there may be others). I did not identify any flaws in methodology or in the interpretation of the results, so I do not have any major comments and recommend this manuscript for publication.

Line 181: minimizing

Line 182: Is 'environmental impacts' broad enough to refer to all challenges to reach sustainable food and agriculture production?

L195: comma in 'PigGTE_x, which'

L231: 'similar to the PGRP' except for Duroc x Asian. Maybe worth adding a short sentence stating why that is the case, given that otherwise they are highlighted in colour.

L579-580: 'cis' and 'trans' in italics everywhere?

Fig S1a: The exponential fit deviates a bit from observations for the last years. Is number of samples slowing down or are they just not published yet? I do not think the exponential fit and R² are needed, so they could be removed.

Fig S3: blank space in 'depths (e)'

Fig S3h: chromosome is missing an 'e'

Fig S4c: capital G in 'CpG'

Table S2 and others: Some breeds are not defined in a uniform way. For example: Does 'X bred' mean 'crossbred'? If so, see the later. Another example: 'A x (B x C)' is also referred to as ' $\frac{1}{2}$ A + $\frac{1}{4}$ B + $\frac{1}{4}$ C'. Please use the same criteria for all.

Table S8 (and possibly others): Some typos in header.

Author Rebuttal to Initial comments
--

Dear editors and reviewers,

The authors are very grateful to hear from you for the valuable comments and suggestions on our manuscript entitled "**A compendium of genetic regulatory effects across pig tissues**" (NG-A61417-T). We have carefully and substantially revised the manuscript based on all the comments and suggestions, which have helped us improve the manuscript. We highlighted the revised parts of the manuscript in the revised version, and listed detailed responses to each of the comments below in a point-by-point manner. We hope our revision has satisfactorily addressed all the comments. Thanks a lot for your time.

Sincerely,

Dr. Lingzhao Fang

Center for Quantitative Genetics and Genomics (QGG), Aarhus University, Aarhus, Denmark

E-mail: lingzhao.fang@qgg.au.dk

Prof. Zhe Zhang

College of Animal Science, South China Agricultural University, Guangzhou, China

E-mail: zhezhang@scau.edu.cn

Dr. George E. Liu

Animal Genomics and Improvement Laboratory, USDA-ARS, Building 306, Room 111, BARC-East, Beltsville, MD 20705, USA

E-mail: George.Liu@usda.gov

Prof. Kui Li

Agricultural Genomics Institute at Shenzhen, CAAS, Building A, Room 311, Buxin Road, Dapeng, Shenzhen, Guangdong, China

E-mail: likui@caas.cn

Prof. Albert Tenesa

The Roslin Institute, Royal (Dick) School of Veterinary Studies, The University of Edinburgh, Midlothian EH25 9RG, UK

E-mail: Albert.Tenesa@ed.ac.uk

Comments from Editors:

Your Article, "A compendium of genetic regulatory effects across pig tissues" has now been seen by 3 referees. You will see from their comments copied below that while they find your work of considerable potential interest, they have raised quite substantial concerns that must be addressed. In light of these comments, we cannot accept the manuscript for publication, but would be very interested in considering a revised version that addresses these serious concerns.

Overall, the three referees appreciate the aims of pig GTEx but are divided when it comes to the overall novelty and impact.

Reviewer #1 thinks that the novelty falls short, and also suggests there may be unresolved technical issues. They also think that the human-pig cross-species comparison is problematic. Reviewer #2 is more positive, but notes a range of technical concerns also, some of which overlap with Reviewer #1's comments. Most importantly, they request substantial improvements to data availability.

Reviewer #3 is straightforwardly supportive, and has only minor comments.

We believe that there is a path to publication but it is clear that substantial revisions are needed. There is useful and specific guidance for improvement. We think that the technical comments are all reasonable and must be fully addressed, particularly the potential for batch effects. We also think that the novelty would have to be improved to persuade Reviewer #1 to be supportive; there are a few suggestions for new analyses that could do so, and we would also encourage you and your co-authors to consider further ideas as well.

We hope you will find the referees' comments useful as you decide how to proceed. If you wish to submit a substantially revised manuscript, please bear in mind that we will be reluctant to approach the referees again in the absence of major revisions.

To guide the scope of the revisions, the editors discuss the referee reports in detail within the team, including with the chief editor, with a view to identifying key priorities that should be addressed in revision and sometimes overruling referee requests that are deemed beyond the scope of the current study. We hope that you will find the prioritised set of referee points to be useful when revising your study. Please do not hesitate to get in touch if you would like to discuss these issues further.

If you choose to revise your manuscript taking into account all reviewer and editor comments, please highlight all changes in the manuscript text file. At this stage we will need you to upload a copy of the manuscript in MS Word .docx or similar editable format.

*1) Include a “Response to referees” document detailing, point-by-point, how you addressed each referee comment. If no action was taken to address a point, you must provide a compelling argument. This response will be sent back to the referees along with the revised manuscript.

*2) If you have not done so already please begin to revise your manuscript so that it conforms to our Article format instructions, available here.

*3) Include a revised version of any required Reporting Summary:
<https://www.nature.com/documents/nr-reporting-summary.pdf>

Please be aware of our guidelines on digital image standards.

AU: We appreciate all these valuable and constructive comments and suggestions, and have revised the manuscript accordingly, mainly including 1) addressing all the technical comments raised by the reviewers, particularly in RNA-Seq genotype imputation, batch effect adjustment and control of inference errors; 2) increasing the novelty of the manuscript by conducting additional analyses, particularly in the comparative analysis of pigs and humans; 3) providing all the relevant data/resources (e.g., metadata of all the samples and full summary statistics of molQTL) in the updated web portal (<http://piggtex.farmgtex.org>) as suggested by the reviewer

#2; 4) formatting the manuscript to conform to the Article format instructions. Besides a revised manuscript, we provided our detailed point-by-point responses to all the comments below and updated the Reporting Summary and Checklist files.

Reviewers' Comments:

Reviewer #1

Remarks to the Author:

This is a GTEx type study in pigs where the RNA-Seq data are collected from the SRA, which is the bulk of the data and analyses. The paper also includes additional datasets such as bisulfite sequencing, single cell RNA-Seq and hi-C, also from public sources. A key resource in making this study possible is public WGS datasets so variants called from RNA-Seq (potentially other datasets as well) can be imputed to obtain sequence level genotypes. This allows the authors to map QTLs (molQTLs) for different omic measurements. Typical analyses then follow, including mapping of regulatory variants, enrichment analysis, integrative analysis of various kinds. The results are more or less expected. The same research groups have previously published in the same journal similar work in cattle.

The analyses are careful and I have a few technical comments below that need to be addressed. The paper is mostly analysis work and strikes me as a bit superficial. There is a large amount of data analyzed and the authors try hard to be comprehensive. However, the analysis offers relatively few previously unknown insights and some of the claims aren't fully supported by the analysis (e.g. using the pig as a model for complex traits in humans).

AU: Thank you very much for the overall comments and suggestions. We have revised the manuscript according to the detailed and constructive comments below. We have conducted additional analyses to address the technical comments and to provide novel insights, particularly in the comparison of gene regulation between humans and pigs, as suggested. The detailed analyses, results, and responses to each comment have been listed below, and the respective parts of the manuscript have been revised and highlighted carefully.

Major comments:

Q1) All analyses in the paper rely on accurate imputation by RNA-Seq variants. I would like to see more comprehensive characterization of imputation accuracy. At the minimum, please provide a) imputation accuracy for all molQTLs in an independent population (individuals used to assess accuracy not in PGRP); b) imputation accuracy based on not just concordance, but also r^2 ; c) clarification that the imputation accuracy reported in extended data figure 1 is based on individuals that are not present in the PGRP. d) when summarizing imputation accuracy, do one for the entire variant set and do another one for common variants ($MAF > 0.05$) only.

AU: We agree with the reviewer that the imputation accuracy of RNA-Seq variants is essential for subsequent molQTL mapping. We thus carefully clarified previous analyses on genotype imputation and conducted additional analyses as suggested.

For question **a)**: All the populations (**Extended Data Figure 1**) that have been used in this study to evaluate the imputation accuracy of RNA-Seq variants were independent of PGRP (the genotype imputation reference panel). We have clarified this in the revised manuscript (**Lines 1358-1367**). To further support this, we collected a total of 725 additional samples with both genotypes (from whole-genome sequence [WGS] or 50K SNP array) and RNA-Seq data in seven populations/breeds that were independent (**Revised Table S3**) to PGRP, including Duroc (n = 38, SNP array), Yorkshire (n = 133, WGS and SNP array), Landrace×Yorkshire (n = 25, WGS), Duroc×Landrace×Yorkshire (n = 356, SNP array), Duroc×Diannanxiaoer (n = 11, WGS), Tibetan(♂)×Berkshire(♀) (n = 81, WGS), and Berkshire(♂)×Tibetan(♀) (n = 81, WGS). We then called SNPs from these RNA-Seq samples and imputed them to the sequence level using the PGRP as we did previously.

For question **b)**: We then carefully evaluated the imputation accuracy of RNA-Seq variants in these independent populations mentioned above by comparing imputed genotypes from RNA-Seq to those 1) directly called from WGS data (i.e., RNA - WGS); and 2) imputed from the 50K SNP array (i.e., RNA - imputed SNP array). As suggested by the reviewer, we measured the imputation accuracy using both concordance rate (CR) and genotype correlation (r^2). We observed an average imputation accuracy of 0.94 (CR) and 0.82 (r^2) for all the variants (**Response Figure 1a and Revised Extended Data Figure 1g**). We further summarized the imputation accuracy in each of the populations/breeds separately (**Response Figure 1b, and**

Revised Extended Data Figure 1h), and observed an average CR of 0.95 and r^2 of 0.90 in the common commercial breeds and their crossbreds (i.e., Duroc, Yorkshire, Landrace x Yorkshire, and Duroc x Landrace x Yorkshire). Of note, most (74%) of RNA-Seq samples used in the current study were generated from commercial pig breeds and their crossbreds. In addition, the imputation accuracy was stable across distinct genomic features (e.g., 5'UTR, 3'UTR and intron) (**Response Figure 1c, and Revised Extended Data Figure 1i**).

For question **c**): as suggested, in the revised legend of **Extended Data Figure 1**, we have clarified that *the imputation accuracy reported here is based on individuals that are not present in the PGRP*.

For question **d**): We have summarized the imputation accuracy according to different minor allele frequency (MAF) bins (**Response Figure 1a, and Revised Extended Data Figure 1g**). The average CR and r^2 were 0.94 and 0.82 for all the variants, 0.94 and 0.83 for common variants (MAF > 0.05), and 0.98 and 0.76 for rare variants (MAF < 0.05), respectively.

Altogether, these results demonstrated that the imputation accuracy of common RNA-Seq variants was sufficient for the subsequent molQTL mapping. This was in line with previous findings in the CattleGTEx¹ and other studies^{2,3}. We have added all the above information in the revised manuscript (**Lines 226-228, 1358-1367, Revised Extended Data Figure 1g-i, Table S3**).

Response Figure 1. Imputation accuracy of RNA-Seq SNPs. a) Concordance rate (CR) and squared correlation (r^2) of imputed and observed genotypes in 50 evenly spaced MAF bins based on individuals that are not present in the Pig Genomics Reference Panel (PGRP). “ALL” represents the entire variants. **b)** CR and r^2 of imputed genotypes from RNA-Seq only and those directly called from whole-genome sequence (WGS) data (red), and imputed genotypes (blue) from SNP array, respectively, in the same individuals. Labels of x -axis are breeds and number of individuals. **c)** CR and r^2 of imputed and observed genotypes in different genomic regions annotated by SnpEff (v.4.3)⁴.

Q2) It appears that the expression estimates used to cluster gene expression (section starting line 242) are not adjusted for covariates (known or PEER). The method describes adjustments later in the mapping section. This is important, any clustering should be performed post batch effect adjustment. Same goes of the cis-h2 analysis, please clarify if the expression was adjusted for unwanted latent confounding factors.

AU: As suggested by the reviewer, we re-performed the sample clustering using the gene expression estimates adjusted for unwanted batch effects as did in the molQTL mapping, resulting in a similar clustering pattern regarding tissue types (**Response Figure 2a**), compared

to that obtained using unadjusted expression data (**Revised Fig. 2a**). This indicates that, in general, the potential contribution of unwanted latent confounding factors to the gene expression variance is smaller than that of tissue types in the RNA-Seq samples. In addition, tissue-specific gene expression analysis led to very similar results before and after adjusting gene expression data, where tissue-specific genes clearly reflected the known biology of their respective tissues (**Response Figure 2b, Revised Extended Data Figure 3b**). As done in the molQTL mapping, we have accounted for the unwanted latent confounding factors for gene co-expression analysis and *cis-h²* estimation. We clarified this carefully in the revised manuscript (**Lines 1245-1247, 1443-1449**).

Response Figure 2. Clustering of RNA-Seq samples and tissue-specific genes based on adjusted gene expression. (a) Clustering of 7,095 RNA-Seq samples based on the normalized gene expression (Transcripts Per Million, TPM) of 6,500 highly variable genes, defined as the top 20% of genes with the largest standard deviation of TPM across samples. Gene expressions were adjusted for nuisance batch effects. (b) Gene numbers (left), expression pattern (middle, Transcripts Per Million, TPM), and enriched Gene Ontology (GO) terms (right) of tissue-specific genes in 34 tissues.

Q3) All subsequent analyses following molQTL mapping rely on proper control of inference errors. The analyses as presented adjusted for covariates identified by PEER and genotype PCs, etc. I would like to see a few more pieces of information including a) histogram summarizing correlation between raw and adjusted gene expression; b) a few randomly chosen examples of QQ plots for molQTL mapping to show that the adjustment was adequate. With the small sample size, 86% (!) of all genes tested contain eQTLs. This is a remarkably high proportion even when combined over all tissues and deserves a bit more scrutiny. Put this in the context of the cis-h² estimates (mean = 0.14) and I'm really worried about the control of FDR. Figure 2a says very little, even if everything is false discovery, you would still expect to see positive correlation. I'd like to see more analyses done here to help me gauge the level of false discovery in this particular part of analysis. The authors would have to think more and harder about it here but one idea is to perform permutations for some genes to see if a) the adjustment was adequate; b) the FDR reported by tensorQTL is adequate.

AU: As suggested by the reviewer, we did additional analyses and provided more detailed information below to support the results of molQTL mapping.

a) We summarized Pearson's correlations of raw and adjusted expression for an average of 17,311 expressed genes (ranging from 8,201 in morula to 20,945 in testis) across 34 tissues being analyzed in the molQTL mapping, and observed an average correlation of 0.49 across tissues, ranging from 0.37 in lymph node and 0.76 in morula (**Response Figure 3, and Revised Figure S8b-c**).

b) To assess whether the latent confounding nuisance factors were well controlled for in the molQTL mapping, we calculated the genomic control inflation factor (λ) for each of the tested genes and provided QQ plots for three randomly chosen gene examples below (**Response Figure 4, and Revised Figure S9**). The λ is a statistics to quantify whether the population structure is

adequately adjusted for in association studies⁵, which often requires full summary statistics (χ^2) of associations across the whole genome. We thus performed a genome-wide association studies for each gene in 12 tissues with over 150 samples, accounting for the same covariates as used in the molQTL mapping. As shown in **Response Figure 4a (Revised Figure S9a)**, the mean λ across these 12 tissues is 0.89, ranging from 0.67 in the testis to 1.13 in the muscle. The **Response Figure 4b-d (Revised Figure S9b-d)** shows QQ plots of three randomly chosen genes i.e., *C9orf131* (ENSSSCG00000005298) in the muscle, *CCDC3* (ENSSSCG00000011110) in the liver, and *KMT2C* (ENSSSCG00000020663) in the blood.

c) Across all 34 tissues, we conducted *cis*-eQTL mapping for a total of 17,431 non-overlapping protein-coding genes, resulting in 14,988 genes (86%) with at least one significant eQTL (eGene) in a least one tissue. The numbers of eGenes and tissue-specific eGenes detected in each tissue are summarized in **Response Figure 5a (Revised Extended Data Figure 4d)**. Muscle had the largest sample size and produced 9,724 eGenes (56%), followed by the testis (35.4%) and the liver (35.3%). In addition, the relationships between eGene discovery power (i.e., the proportion of tested genes being discovered as eGenes) and tissue sample size were similar among pigs, cattle, and humans (**Response Figure 5b, and Revised Figure S10a**). The mean *cis*-heritability (*cis*- h^2) of all tested genes (including those with significant or non-significant *cis*- h^2 estimates) across tissues was 0.14. Of note, the mean *cis*- h^2 for 16,174 (93%) genes with significant *cis*- h^2 estimates in at least one tissue was 0.33 (**Response Figure 5c**). These results are in agreement with those in the human GTEx (v8)⁶, which found that 94.7% of all tested genes had at least one significant eQTL across tissues.

d) In this study, to carefully control inference errors in the molQTL mapping, we employed two layers of multiple testing correction based on the permutation approach⁷, as implemented in the TensorQTL software⁸, similar to the approach used in the human GTEx⁶. In the first layer, we applied an adaptive permutation approach to calculate the empirical *P*-values of variants within each gene and then obtained the permutation *P*-value of the lead variant for each gene. In the second layer, we conducted the multiple testing correction for the permutation *P*-values of lead variants across all tested genes by employing the Benjamini-Hochberg method, and considered genes with FDR < 5% as genome-wide significant eGenes. We clarified this in the revised manuscript (**Lines 1362-1368**). In addition, we employed multiple approaches to validate the results of molQTL mapping, including a comparison of molQTL results with those from a linear mixed model (**Revised Extended Data Figure 6a**), internal validation (**Revised Figure 2f**), external validation in independent populations (**Revised Extended Data Figure 6b-d**), and allele-specific expression (ASE) analyses (**Revised Figure 2g, Extended Data Figure 6e-g**). All

these results consistently demonstrated that the molQTL detected in this study had a high replication/validation rate, indicating adequate adjustment of nuisance batch effects and proper control of inference errors for the molQTL mapping.

We have updated all the above information in the revised manuscript (**Lines 276-277, 286-288, and 1443-1449, Revised Figures S8b-c, S9, S10a-b**).

Response Figure 3. Correlations between raw and adjusted expression levels of expressed genes across 34 tissues. (a) Histogram of correlations between raw and adjusted expression levels across 34 tissues. **(b)** Comparison of correlations between raw and adjusted expression levels among 34 tissues.

Response Figure 4. Genomic control for molQTL mapping. (a) Distribution of the genomic control inflation factor (λ) for all tested genes in 12 tissues with over 150 samples. The red line represents the mean λ (0.89) across 12 tissues. (b-d) Quantile-Quantile (QQ) plots of genome-wide associations for *C9orf131* (ENSSSCG00000005298) in the muscle, *CCDC3* (ENSSSCG00000011110) in the liver, and *KMT2C* (ENSSSCG00000020663) in the blood.

Response Figure 5. eGenes and cis-heritability across all tissues. (a) The number of eGenes in each tested tissue, with 86% of the tested genes (red bar) were eGene in at least one tissue. The blue points represent the number of tissue-specific eGenes. (b) Relationship between the proportion of eGenes and the sample size for the tissue in PigGTEX, CattleGTEX, and HumanGTEX (v8). The curves were fitted with $y \sim \log(x)$ by the *geom_smooth* function in ggplot2. (c) Box plot showing the *cis*-heritability estimates of genes across 34 tissues that were significant (likelihood ratio test $P < 0.05$) or non-significant, with 16,174 (93%) unique genes having significant *cis*-heritability in at least one tissue.

Q4) Line 305: it is good that the tensorQTL estimates are checked against mixed model based estimates. My question is why there are multiple correlation from the same tissue? Is it because the analysis was performed in different datasets (breeds)? Although the average of the correlation was high, it makes me worried that it can go as low as negative. It should be checked to find out why some have low correlation. You would expect if everything is adjusted (especially relatedness) well, these two should yield very similar results (definitely higher than 0.9). In addition, correlation masks the effect of a baseline shift. For example, tensorQTL may report in general lower P value and you would still observe positive correlation, but that makes FDR estimates much more trickier.

AU: In each of the 34 tissues, we calculated Pearson's correlations of summary statistics (effect size estimates and *P*-values) of significant SNPs for each of the eGenes obtained from tensorQTL (the linear regression) and GCTA (the mixed linear model). Therefore, there were multiple correlations (as each eGene had two correlation coefficients, one for effect size and the other for *P*-value) in the same tissue (**Revised Extended Data Figure 6a**). As suggested by the reviewer, we have carefully checked the reason why correlations of some eGenes were very low and even negative. First, we noticed that in the previous version, the genotype PCs were not included in the mixed linear model. We thus redid the mixed linear model analysis by considering the same PEER factors and genotype PCs as covariates as used in the tensorQTL analysis. We observed that an average 96% of eGenes had a correlation of > 0.9 across all 34 tissues (**Response Figure 6a-b**). We also found that those eGenes with very low correlations tended to have a small number of significant SNPs (**Response Figure 6c**). In addition, we combined the summary statistics of all eGenes within a tissue to compute one Pearson's correlation for that tissue, and observed a median correlation of 0.98, 0.96 and 0.94 for z-score (i.e., slope/slope_se), slope, and log₁₀-transformed *P*-value, respectively, across the 34 tissues (**Response Figure 6d**). We have updated all the relevant results in the revised manuscript (**Revised Extended Data Figure 6a-d**).

To check whether there is a baseline shift between *P*-values from tensorQTL and GCTA, we considered summary statistics of lead eQTL of all the 14,988 eGenes across the 34 tissues. As shown in the **Response Figure 6e-f (Revised Extended Data Figure 6e)**, although nominal *P*-values from tensorQTL were generally lower than those from GCTA, permutation-based *P*-values from tensorQTL (detailed in **Q3** and ⁷) were more strict than those from GCTA. In this study, when defining eQTL and eGenes, we only considered the permutation-based *P*-values from tensorQTL. We have added this information in the revised manuscript (**Lines 308-310**).

Response Figure 6. Relationship of summary statistics for eGenes between the linear model (LM, TensorQTL) and the mixed linear model (MLM, GCTA) across 34 tissues. (a) Distribution of the Pearson's correlations of Z-score between LM and MLM. **(b)** Pearson's correlation of summary statistics for each eGene in each tissue between LM and MLM. **(c)** Relationship between correlations of Z-score and the number of significant eQTLs. **(d)** Pearson's correlation of combined summary statistics for all eGenes across the 34 tissues between LM and MLM. **(e)** Correlation between nominal P derived from LM and P derived from MLM for the lead eQTL of all the eGenes across the 34 tissues. **(f)** Correlation between permutation-corrected P derived from LM and P derived from MLM.

Q5) Line 401-403: the fact that there are enrichments for these exonic annotations, much stronger than up/downstream makes me worried that there is mapping bias. In other words, the gene expression estimation is influenced by SNPs present in exons.

AU: We double-checked the distribution of all the SNPs that were tested in the molQTL mapping along the genomic annotations. As shown in the **Revised Extended Data Figure 1d**, we did not observe an obviously biased distribution of all these SNPs in exonic variants (e.g., synonymous and missense) as compared to variants in the genotype imputation reference panel (i.e., PGRP). Therefore, we do not think the higher enrichment of molQTL in exonic annotations than in up/downstream regions was due to the biased distribution of SNPs being tested along the genome. In addition, higher enrichments of e/sQTL in exonic annotations than in up/downstream were also observed in human GTEx (v8) (Figure 4a of ⁶) and rat GTEx (Figure 4g of ⁹). A recent review paper on the functional characterization of eQTL also summarized that eQTL can exert their effects on the transcriptome through multiple complex molecular mechanisms, including altering protein structure and function (e.g., variants in or near splice sites can lead to the altered transcript and protein structure, and variants within transcripts can cause the altered transcript stability and translation rates)¹⁰. Although there was significant enrichment (~2.6 fold) of molQTL in exonic annotations (**Revised Figure 4a**), the proportion of such variants over all the identified molQTL was just around 5.4%, i.e., 5.4% for eQTL, 5.5% for sQTL, 5.2% for eeQTL, 5.4% for lncQTL, and 5.8% for enQTL. We have discussed this in the revised manuscript (**Lines 420-424**).

Q6) Line 439-446: I wonder if randomly chosen SNPs matched for the molQTLs in terms of MAF would explain similar amounts of heritability given that pigs have high LD.

AU: To explore whether randomly chosen MAF-matched SNPs could explain similar amounts of heritability for pig complex traits as the molQTL did, we applied two approaches, including LDAK (using individual genotype and phenotype data)¹¹ and MESC (using GWAS summary data)¹², to partition heritability of 16 different complex traits in pigs ($n = 4,127 \sim 4,383$). **1)** For LDAK analysis, we used a mixed linear model with multiple genetic components to partition the heritability of a complex trait into three different SNP sets (i.e., independent molQTL, random MAF-matched SNPs and the remaining SNPs): $y = Xa + Zg_{molQTL} + Zg_{random} + Zg_{remaining} + e$, where y is the vector of phenotypes of the target complex trait, X is the covariates matrix, a is the effects of covariates, Z is the design matrix that allocates phenotypes

to genetic values, $g_{molQTL} \sim N(0, G_{molQTL} \sigma_{g_{molQTL}}^2)$, $g_{random} \sim N(0, G_{random} \sigma_{g_{random}}^2)$, and $g_{remaining} \sim N(0, G_{remaining} \sigma_{g_{remaining}}^2)$ are the polygenic effects of conditionally independent molQTL, random MAF-matched SNPs, and the remaining SNPs, respectively, where G_{molQTL} , G_{random} and $G_{remaining}$ are GRMs calculated based on genotypes of conditionally independent molQTL, random MAF-matched SNPs, and the remaining SNPs, respectively. $\sigma_{g_{molQTL}}^2$, $\sigma_{g_{random}}^2$ and $\sigma_{g_{remaining}}^2$ are genetic variance explained by conditionally independent molQTL, random MAF-matched SNPs, and the remaining SNPs, respectively. e is the residual. 2) For MESC analysis, we employed a previous approach¹² to compute the heritability of complex traits mediated by gene expression that were regulated by independent molQTL or random MAF-matched SNPs. In brief, the contribution of a SNP set to the total heritability of a complex trait can be calculated by $h_{med-SNP}^2 = \frac{\mathcal{L}_{sub}}{\mathcal{L}_{total}} * h_{med}^2$, where $h_{med-SNP}^2$ is the heritability of a complex trait explained by SNPs of interest (conditionally independent molQTL or random MAF-matched SNPs), h_{med}^2 is the expression-mediated heritability by the *cis*-component of all genes, \mathcal{L}_{total} is the sum of expression scores of all tested SNPs, \mathcal{L}_{sub} is the sum of expression scores of SNPs of interest. The expression score of a SNP that was obtained from MESC represents the contribution of that SNP to the total expression *cis*- h^2 of genes¹².

The results from these two approaches consistently demonstrated that the amounts of heritability of complex traits explained by molQTL was significantly higher ($P < 0.05$, paired Student's *t*-test) than that explained by random MAF-matched SNPs (**Response Figure 7, Revised Figure 5b and S13h**). We have added this information in the revised manuscript (**Lines 470-472, 1755-1779**).

Response Figure 7. Heritability of complex traits of pig explained by conditionally independent molQTLs and random MAF-matched SNPs. We estimated the heritability of 16 complex traits with large sample sizes ($n = 4,127\sim 4,383$) using LDAK (a) and MESC (b). The top numerical labels are the P values based on the two-sided paired Student's t -test.

Q7) I find the whole section of finding shared genes for complex traits between humans and pigs rather superficial and I will explain why it's not a good idea to do this. It's completely expected that the tissue specificity is conserved (6a). But conservation as measured by PhastCons and LOEUF differing between the different classes of genes based on eGene sharing can be due to a number of reasons, the most trivial one being that eGenes or not is influenced by expression level in general. To say that this is essential functions and purifying selection is an overstretch. The correlation between effect sizes in the two species is very weak (6d), could be even weaker if one variant (lower left corner) is taken out. Lastly, for the correlations between TWAS effect sizes, it's easy to pick among hundreds of such correlations some that have good correlations (6f-h). It's probably hard to perform such analysis, but it would be nice if some "unrelated" traits between the two species can be used as a negative control to see what would have been expected under null. In fact figure 6f would have been a good negative control where high correlation shows up.

AU: As suggested by the reviewer, we conducted additional analysis to compare gene regulation and complex trait genetics between humans and pigs. As it is expected that tissue-specificity of gene expression is conserved between these two species, we moved this part of the analysis (**Original Figure 6a**) to supplementary files (**Revised Figure S16d**). We agree with the reviewer that sharing of eGenes between two species could be affected by a number of reasons, such as gene expression level. We thus tuned down the statement that the sharing of eGenes between species is due to strong purifying selection in the revised manuscript (**Lines 575-599**), to avoid the overstretch. We repeated the species-sharing analysis of eGenes by using the local false sign rates (LFSR) of eQTL activity that were obtained from the multi-tissue meta-analysis (MashR) in pigs and humans⁶. In each of the 17 matching tissues between pigs and humans, we divided orthologous genes into four groups (i.e., “Neither”, “Human-specific”, “Pig-specific”, and “Shared”), where “Shared” represents genes that were eGenes in both species, “Pig-specific” represents genes that were eGenes in pigs only, “Human-specific” represents genes that were eGenes in humans only, and “Neither” represents genes that were non-eGenes in both species. We indeed observed a significant difference in expression level, tissue-specific expression (measured by Tau value), and tissue-sharing of eQTL (LFSR < 0.05) among the four groups of genes (**Response Figure 8a**). In general, expression levels of genes were negatively correlated with LOEUF scores, which was consistent among the four groups of genes in both humans and pigs (**Response Figure 8b**), indicating that genes with higher expression levels were less likely to be tolerant to loss of function mutations. Among these four groups of genes, ‘Shared’ eGenes had the weakest negative correlation between expression levels and LOEUF scores, while ‘Neither’ eGenes had the strongest negative correlation (**Response Figure 8b**). Of specific note, although they had the highest expression levels, ‘Shared’ eGenes showed the strongest tolerance to loss of function mutations (i.e., the highest LOEUF scores) among the four groups of genes at different expression levels (**Response Figure 8c**). This is also in line with the previous observation in human blood that eGenes with high expression levels were more tolerant to loss of function mutations than non-eGenes with similar expression levels¹³. In addition, we found that the expression level of most genes was weakly or even not correlated with their PhastCons score, eQTL/eGene detection, and *cis-h²* estimate across tissues (**Response Figure 8d-f**). We found a total of 783 eGenes that were active in all tissues in both species and these eGenes were significantly enriched in metabolic processes like organic acid catabolic process (**Revised Table S28**). A total of 194 genes were not eGenes in any tissues in both species and these were significantly enriched in essential biological functions such as cell fate commitment and organ development (**Revised Table S29**).

Due to the challenge of identifying orthologous variants between species, we only analyzed 112 orthologous variants in the **Original Figure 6d**. The correlation of eQTL effects between pigs and humans was thus more likely to be influenced by outliers. Due to the current availability of RatGTEx data⁹, we further explored the sharing of the eQTL effects among humans, pigs, and rats, using an approach proposed previously in the RatGTEx project. For each eGene, we averaged the effect ($|\log_2(\text{aFC})|$) of significant eQTL across 17 matching tissues, separately for humans and pigs. We observed a significant positive correlation ($r = 0.56$) of averaged eQTL effect estimates between humans and pigs (**Response Figure 9a**), which was higher than that ($r = 0.24$) observed between humans and rats previously⁹. In general, matching tissues between pigs and humans had a higher correlation of eQTL effect estimates compared to non-matching tissues (**Response Figure 9b**). Furthermore, we found a significant but weak correlation ($r=0.09$) of $cis-h^2$ of orthologous genes between humans and pigs (**Response Figure 9c**), similar what was observed between humans and rats ($r=0.10$)⁹.

We agree with the reviewer that a proportion of correlations between human and pig traits are expected to be significant when testing many combinations. To properly control the inference errors, we thus performed a permutation-based test (1,000,000 times) to detect significant trait-pairs between pigs and humans. In general, we observed a clear deviation of the observed P -values from the permutation-based null distribution, indicating that some trait pairs were significantly correlated between humans and pigs (**Revised Figure 11a**). After correcting for multiple testing using the Benjamini-Hochberg method, we considered 89 trait-pairs with $\text{FDR} < 10\%$ as significant (**Revised Table S31, Revised Figure 6e**).

To investigate whether GTEx-like resources can facilitate cross-species gene mapping of complex traits by borrowing ‘information’ at the level of orthologous genes instead of individual variants, we performed a cross-species meta-TWAS analysis by modifying a multi-ancestry metaTWAS method used for humans¹⁴. We calculated the Z-stastics of meta-TWAS as: $Z_{meta} = \frac{N_i Z_{TWAS,i} + N_j Z_{TWAS,j}}{\sqrt{N_i^2 + N_j^2}}$, where $Z_{TWAS,i}$ and $Z_{TWAS,j}$ are the Z-stastics from the pig TWAS and human TWAS results, respectively; N_i and N_j are the population sizes of the pig TWAS and human TWAS, respectively. If the tested trait is a case-control study, we adjusted the sample size as $4 / (\frac{1}{N_{cases}} + \frac{1}{N_{controls}})$. We chose several well-recognized homologous trait-pairs between humans and pigs to perform the meta-TWAS and also selected several non-homologous trait-pairs as negative controls. For homologous trait-pairs, cross-species meta-TWAS improved the discovery of trait-associated genes in humans (**Revised Figure 12a**). For instance, cross-species meta-

TWAS analysis of pig average backfat thickness (BFT) and human body weight (BW) revealed eight new genes (FDR < 0.05) associated with BW in humans (**Revised Figure 12b**). Based on GWAS of 3,302 traits in humans (GWAS ATLAS)¹⁵, phenome-wide association studies (PheWAS) showed that five of these eight newly detected genes were also significantly associated with other BW-relevant traits in humans, such as height, birth weight, and BMI (**Revised Table S32**). We further employed LDSC¹⁶ to partition the heritability of human weight into 10 groups of genes that were sorted by their *P*-values (from smallest to largest) based on pig BFT TWAS. The higher heritability for human BW were observed for gene groups that had higher significance in the pig BFT TWAS (Pearson's $r = 0.68$ and $P = 0.03$) (**Response Figure 11c**).

We have added all the above information in the revised manuscript (**Lines 561-631, 1844-1871**).

Response Figure 8. Characteristics of eGenes in pigs and humans. (a) Expression levels (left), TAU values (middle), and tissue-sharing levels (right) for four groups of orthologous genes across 17 tissues in pigs, including non-eGenes in both species (Neither, n = 3,993), human-specific eGenes (Human-specific, n = 8,174), pig-specific eGenes (Pig-specific, n = 3,882), and eGenes in both species (Shared, n = 10,574). We defined the tissue-sharing level of an eGene as the proportion of tissues that the eGene was

active in ($LFSR < 0.05$) across the 34 evaluated pig tissues. *** indicates $P < 0.001$ for the two-sided Wilcoxon rank-sum test. Diamond represents the median value and the error bar represents the upper and lower quartiles. **(b)** Pearson's correlation between tolerance to loss of function mutations (LOEUF) and expression levels in pigs (left) and humans (right). The lines were fitted by *geom_smooth* function in ggplot2. *** indicates $P < 0.001$ for the Pearson's correlation. **(c)** LOEUF in the four groups of orthologous genes in 10 evenly spaced expression level bins, where "0~10%" represents the genes with the lowest 10% expression levels and "90~100%" represents the genes with top 10% expression levels. NS., *, ** and *** indicate one-sided Wilcoxon rank-sum test $P > 0.05$, $P < 0.05$, $P < 0.01$, and $P < 0.001$, respectively. **(d)** Pearson's correlation between PhastCons scores and expression levels of genes. **(e)** Numbers of eGenes divided into four groups based on Pearson's correlation between *cis*-eQTL effect size and eGene expression level across tissues. "Positive *r*" and "Negative *r*" represent genes with significant ($FDR < 0.05$) positive and negative correlations, respectively. "Uncorrelated" represents genes without significant correlations and "Not tested" represents those not tested owing to have not enough observations. **(f)** Pearson's correlation between *cis*-heritabilities and expression levels of genes. The *ALB* gene has a high expression level but its *cis*-heritability is zero in the liver.

Response Figure 9. Similarity of pigs with humans in eQTL effects and *cis*-heritability. **a)** Pearson's correlation of eQTL effect size in orthologous genes between pigs and humans based on the average $|\log_2(aFC)|$ of significant eQTLs across humans and pigs in 17 matching tissues. **b)** Pearson's correlation of eQTL effect size in orthologous genes for each tissue. The red triangle represents the matching tissue between pigs and humans. **c)** Pearson's correlation of *cis*-heritability in orthologous genes between pig and human based on the average *cis*-heritability of each gene across human and pig in matching tissues.

Response Figure 10. Correlation of orthologous gene effect sizes between pig and human traits derived from transcriptome-wide association studies (TWAS). (a) QQ-plot of P -values of TWAS correlations between pig and human (i.e., real data), compared to a permutation control (i.e., permutated data). P values were obtained by the Wilcoxon rank-sum test. (b) Distribution of P -values from the TWAS correlations between pig and human. The blue line is the permutation corrected $P = 0.05$, and the red lines are P -values under different FDR cutoffs to adjust for multiple testing.

Response Figure 11. Cross-species meta-analysis of TWAS (metaTWAS) between pig and human.

(a) Differences in the number of significant genes (FDR < 0.05) from cross-species (pig and human) metaTWAS, compared to those from human TWAS. (b) FDR of discovered genes in human TWAS (RawTWAS) and cross-species metaTWAS in the brain for BFT (pig) and Weight (human). The red points represent genes that are significant (FDR < 0.05) in metaTWAS but not in rawTWAS. (c) Pearson's correlation (r) between the TWAS significances (color bar) of genes and their enrichments of heritability for human weight. The orthologous genes were divided into ten evenly spaced bins by sorting the P -values of TWAS in the brain of pig BFT.

Minor comments:

1) Line 218: I wonder what enhancer expression means. Even if reads map to enhancer regions, it does not mean enhancers are generally transcribed. They can be part of transcripts in terms of genomic coordinates, but their functions are not exerted through transcription. I find it odd to quantify enhancer expression.

AU: Enhancer expression means the expression level of enhancer RNAs (eRNAs). The eRNAs are generally short, non-coding, and bidirectionally transcribed¹⁷⁻¹⁹. We agree with the reviewer that enhancers do not exert their biological functions through transcription. However, several previous studies proposed that the expression level of an enhancer can approximately reflect its functional activity^{17,19-23}. To control for potential contamination of transcribed genes, we only focused on transcribed enhancers that did not overlap with any known gene regions (including protein-coding gene, lncRNA, pseudogene, tRNA, miRNA, and snoRNA) using a previously reported strategy^{17,21,24}, resulting in 3,679 enhancers for the downstream analyses. We have clarified this part in the revised manuscript (**Lines 1209-1212**).

2) Line 218 + Line 1093-1102: It appears that the PCGs and the lncRNAs are normalized separately. This needs justification. They are both transcribed and should be considered in the same pool.

AU: As suggested by the reviewer, we repeated the molQTL mapping using jointly normalized expression data and compared the new results with the previous results that were obtained from separately normalized expression data. We found that the replication rate (π_1) of molQTL was close to 100% and the correlations of summary statistics (effect and P values) were higher than

0.95 across 34 tissues (**Response Figure 12**). We have added this information in the revised manuscript (**Lines 1474-1480, Revised Figure S21a**).

Response Figure 12. The replication rate (π_1) of eQTLs discovery between separate and joint normalization for protein-coding genes and lncRNAs expressions and the Pearson's correlation (r) of their summary statistics across 34 tissues. "Separate in Joint" represents the replication rate of eQTL of "Separate" in "Joint" and "Joint in Separate" represents the replication rate of eQTL of "Joint" in "Separate". "Separate" and "Joint" represent eQTL mapping using separate normalization and joint normalization for the expression of PCGs and lncRNAs, respectively.

3) Line 238: unless these are multi-omic data collected from the same samples, call them other omics datasets.

AU: Corrected throughout the revised manuscript (e.g., **Line 238**).

4) Line 274: I don't think it make sense to cluster h2 estimates. I have a hard time envisioning a mechanism where the h2 would cluster based on tissues. What you are clustering may be just expression levels (i.e., genes expressed at higher level tend to have higher h2).

AU: We found that the cis-heritability ($cis-h^2$) of genes was weakly correlated (Pearson's $r=0.06$) with their expression levels (**Response Figure 8f**). In addition, for the majority of tested genes, the discovery of eQTL was not significantly correlated with their expression levels across tissues (**Response Figure 8e**). For instance, *ALB* is highly expressed in the liver (**Response Figure 13**), which has a $cis-h^2$ close to 0 and no significant eQTL was detected for this gene in pigs or humans. Hence, the tissue clustering based on $cis-h^2$ of genes may not be fully explained

by expression levels. However, as suggested, it is hard to explain the molecular mechanism behind this clustering, so removed this from the revised manuscript to highlight the tissue-sharing analysis of molQTL.

Response Figure 13. Expression levels of the *ALB* gene across 34 pig tissues (a) and 49 human tissues (b).

5) Line 316: Validation by ASE, this is good. However, looking at extended data figure 6e and f, the majority of the points are around 0 (density high), why is that? These are cis-eQTL to begin with, shouldn't they be mostly farther away from zero?

AU: Thanks for the question! Sorry for the mistake. The previous aFC values were actually log₂-transformed. We have corrected this in the revised manuscript (Line 318).

6) Figure1: this is not "phylogenetic" clustering, the same term was used throughout the manuscript, please consider revising.

AU: We have carefully modified it throughout the revised manuscript (**Revised Figures 1 & S2**).

Reviewer #2

Remarks to the Author:

Comments for NG-A61417-T

Comments for the Author:

In this manuscript, Teng et al. built the most comprehensive Pig Genotype-Tissue Expression Atlas, which contains 1,602 whole-genome sequencing and 9,530 RNA-sequencing samples from multiple pig tissues. The authors also did QTL analyses for five molecular phenotypes (e.g., eQTLs and sQTLs) and compared PigGTEx with the human GTEx data. Overall, the majority of performed analyses are generally convincing and the resources of this study will be widely useful to the community. I have some suggestions to make the PigGTEx more convenient and user-friendly. My detailed questions are listed as follows.

AU: Thank you very much for these constructive suggestions and comments. We have modified the manuscript accordingly!

Major concerns:

Q1. The PigGTEx has 9,530 RNA-Seq samples. After filtering out samples of low quality, only 7,095 RNA-Seq samples were used in the analysis. The authors should provide phenotype information of all RNA-Seq samples in the download page of the website and label which sample were used in the QTL analysis. Do all RNA-Seq samples have matched WGS data? If not, please label which sample have, which not. Please also provide a meta file in the download page to map the RNA-seq ID and WGS ID. The authors also should

provide the raw count tables for gene, exon, lncRNA and enhancer and unnormalized alternative splicing values. Confounders are very important for QTL analysis, the authors should provide the values of both known confounders and unknown PEER confounders for each sample in each tissue type in the download page as human GTEx did. Colocalization analyses with GWAS need both significant and non-significant QTLs. Please also provide full summary statistics of QTL results (not only significant items) for users in the download page of the website.

AU: Thank you very much for all these valuable suggestions to help make the PigGTEx web portal more impactful and user-friendly. We have updated all the resources suggested by the reviewer in the download page of PigGTEx-Portal (<http://piggtex.farmgtex.org>) and another open-access database (noted in the download page of PigGTEx-Portal) i.e., ScienceDB (RNA-Seq and Genotype Data: <https://www.scidb.cn/s/eiqy6j>; full summary statistics of molQTL: <https://www.scidb.cn/s/uiABfy>; results of fastENLOC, SMR, TWAS: <https://www.scidb.cn/s/zU7rIj>): including 1) Phenotype information of all 7,095 RNA-Seq samples and sample IDs used in the molQTL mapping; 2) The meta-data file mapping RNA-Seq sample ID and WGS sample ID (Note: not all the public RNA-Seq samples have matched WGS data); 3) Raw counts for gene, exon, lncRNA and enhancer, as well as unnormalized alternative splicing values; 4) Files containing all known and inferred unknown confounders for each sample in each tissue type used in the molQTL mapping; 5) Full summary statistics of all the molQTL.

Q2. The human GTEx use different number of PEER factors for different tissues based on the sample size. Why the authors used the top ten PEER factors for all tissue types? I also found many unknown age/sex items for RNA-Seq samples in Table S1. Did the authors include age/sex/breed as confounders? If yes, how did the authors deal with the unknown items? If not, please clarify or add.

AU: To determine how many PEER factors to include in the molQTL mapping, we explored the weight variance explained by the top 60 PEER factors, as shown in the **Revised Extended Data Figure 2g** and found that PEER factors beyond the 10th captured only small amounts of variance. This was similar across all tissue types. We thus considered the top ten PEER factors for the molQTL mapping in all tissue types. We also found that most known confounders (e.g., sex and age) in the RNA-Seq samples can be explained by the PEER factors (**Extended Data Figure**

2h). To further investigate whether adding known batch factors as covariates influences the molQTL mapping, we took the muscle data, which has the largest sample size ($n = 1,321$), as an example below. We performed the *cis*-eQTL mapping using the same linear regression model, implemented in TensorQTL, but considering the top ten PEER factors, the top 10 genotype PCs, and known confounders (e.g., age, sex and breed) as covariates. We considered sex as a categorical variable and the missing items as “unknown” class, while age was fitted as a continuous variable and its missing values were replaced by the mean value of the known ages. Comparing these results to the previous ones where only PEER factors and genotype PCs were considered in the eQTL mapping, we found that the replication rates of *cis*-eQTL between them was 99.8%, while the Pearson’s correlations of summary statistics were 0.96 for effect size and 0.98 for *P*-value. We have added this information in the revised manuscript (**Lines 1480-1489**).

Q3. Current QTL tissue-sharing patterns are not convincing. For example, Pituitary is more similar as Uterus than Frontal cortex in Fig. 3a. Another example, Muscle is more similar as Adipose than Heart in Extended Data Figure 4. Please clarify. The colors in heatmaps are very similar, which makes it hard to read (e.g., Fig. 3a and b, Extended Fig. 4b, Extended Fig. 7e and Extended Data Figure 9). Please reassign heatmap color range to make it clear.

AU: Thank you very much for this question. After double-checking the tissue-sharing analysis of molQTL with MashR²⁵, we noticed that we did not consider the direction of molQTL effects in our original results. We have repeated the tissue-sharing analysis of molQTL with MashR while accounting for the direction of molQTL effects and observed a more reasonable tissue-clustering pattern than in the original manuscript (**Response Figure 14**). For instance, pituitary was clustered well with other brain regions (e.g., hypothalamus and frontal cortex). We have updated all the relevant results in the revised manuscript (**Revised Figures 3a-b, Extended Data 7e, 9a-d**). In addition, we discuss in the revised manuscript (**Lines 653-658**) that, in the current phase of PigGTE_x, the big difference in sample size and other biological factors (e.g., breed and cell type composition) across tissue types may influence the tissue-clustering patterns. To make the heatmap easy-to-read, we have reassigned the colors to all relevant heatmaps in the revised manuscript (**Revised Figure 3a-b, Extended Data Figure 4c, 7d-e, 9a-d**).

Response Figure 14. Heatmap of tissues depicting the corresponding pairwise Spearman's correlation (ρ) of *cis*-eQTL effect sizes. Tissues are grouped by hierarchical clustering (bottom). Violin plots (left) represent the Spearman's correlations of the target tissue with other tissues.

Q4. eQTL analysis always include both Protein-coding genes and lncRNAs (such as human GTEx and Brain xQTLServe). It's so strange to split eQTL into PCG *cis*-eQTL and *cis*-lncQTL. Please clarify or combine.

AU: Because, as proposed previously, lncRNA may play distinct roles in regulating complex traits²⁶, we conducted eQTL mapping for PCGs and lncRNAs separately to explore their differences in genetic regulation and complex trait complications. In addition, the DNA sequence of lncRNAs is less selectively constraint across species compared to PCGs, as indicated by PhastCons scores (**Response Figure 15a**). We observed that the *cis*-heritabilities (*cis*- h^2) of lncRNAs (mean = 0.12) were significantly (Wilcoxon rank-sum test $P < 1 \times 10^{-300}$) lower than those of PCGs (mean = 0.14), with 92.85% of tested PCGs having significant (likelihood-ratio test $P < 0.05$) *cis*- h^2 estimates, while only 82.44% of tested lncRNAs had significant *cis*- h^2

estimates (**Response Figure 15b**). As lncRNAs often act as regulators of PCGs²⁷, the separate eQTL mapping of PCGs and lncRNAs can also enable us to study the regulatory relationship among lncRNAs, PCGs, and GWAS through the SMR analysis of multiple molecular phenotypes, with one example shown in the **Extended Data Figure 10f**. Of note, results from separate eQTL mapping of PCGs and lncRNAs were almost the same as those from the joint eQTL mapping in terms of both effect size and significance level (for details, please see responses to minor comment #2 of Reviewer #1). We have added this information in the revised manuscript (**Lines 1474-1480**).

Response Figure 15. *Cis*-heritability (a) and PhastCons score (b) for protein-coding genes (PCG) and lncRNAs. *P*-values are obtained by two-sided Wilcoxon rank-sum tests.

Q5. Trans-QTLs is an important part of QTL studies. Please add trans-QTL analyses for tissue types (such as muscle and brain) which have enough samples ($n \geq 150$).

AU: As suggested by the reviewer, we have conducted an exploratory analysis of *trans*-eQTL in 12 tissues with over 150 individuals in this study and detected an average of 80 *trans*-eGenes (FDR < 0.05) across tissues (ranging from 12 in morula to 203 in muscle) (**Response Figure 16a-c**). Effect sizes of *trans*-eQTL were often much smaller and more cell type-specific than those of *cis*-eQTL, which implies that much larger sample sizes (usually tens of thousands of samples) are required for *trans*-eQTL mapping^{6,7,13}. For instance, a recent study on human blood from 31,684 individuals showed a low replication rate of *trans*-eQTL due to the low statistical power and confounding induced by cell type composition¹³. Here, we took muscle, which had the largest sample size ($n = 1,321$), as an example to conduct an internal validation of *trans*-eQTL by randomly and evenly dividing samples into two groups (Group1 and Group2). We then

conducted the *trans*-eQTL mapping separately, and observed that the replication rate ($\pi 1$) between the two groups was 0.4. The Pearson's correlation of effect sizes between significant *trans*-eQTL in Group1 and the effect sizes of matched SNPs in Group2 was 0.5 (**Response Figure 16c**). We only provided significant associations of *trans*-eQTL mapping in the download page of the PigGTEX web portal as was implemented for the human GTEx, because the full summary statistics of *trans*-eQTL mapping are too big. Below are the detailed methods for *trans*-eQTL mapping, similar to those used in human GTEx⁶, which were also added in the revised manuscript (**Lines 323-330 & 15562-1581**).

To reduce the potential false positives in *trans*-eQTL mapping, we applied a stringent standard to filter genetic variants and genes²⁸. We first calculated the mappabilities of genome-wide variants and the cross-mappabilities of genome-wide gene-pairs using crossmap (<https://github.com/battle-lab/crossmap>)²⁸. We kept SNPs with 75 *k*-mer based mappability > 1 , SNPs in non-repeat regions annotated by the UCSC RepeatMasker track²⁹, and SNPs with MAF > 0.05 for subsequent *trans*-eQTL analysis. For genes, we only considered those with an average mappability of at least 0.8 for *trans*-eQTL mapping. For each gene, we only considered SNPs that did not fall in the same chromosome as the target gene or within $\pm 1\text{Mb}$ of its cross-mappable genes. For *trans*-eQTL mapping, we used a linear mixed model, implemented in GCTA, including a genomic relationship matrix and the same covariates as for *cis*-eQTL mapping. For multiple testing correction, we firstly extracted the most significant *P*-value of genome-wide level across all tested genes and multiplied by 10^6 , similar to the approach used in human GTEx⁶, which was consistent with the assumed effective number of tests at the genome-wide level. We then corrected for multiple testing at the gene level using the Benjamini-Hochberg method and defined genes with FDR < 0.05 as *trans*-eGene. For each *trans*-eGene, we employed the Benjamini-Hochberg method to adjust the multiple testing for *P*-values and considered gene-SNP pairs with FDR < 0.05 as significant.

Response Figure 16. Locations and internal validation of *trans*-eQTL. (a) Pearson’s correlation (r) between the number of *trans*-eGenes (FDR < 0.05) and sample size across 12 tested tissues with sample size >150. (b) Locations and association P -values ($-\log_{10}$ scale) of the most significant *trans*-eQTL (FDR < 0.05) for each *trans*-eGene in 12 tested tissues. (c) Pearson’s correlation of effect sizes of 925 significant (FDR < 0.05) *trans*-SNP-gene pairs in muscle in Group1 and those of matched SNPs in Group2, where we conducted the internal validation of *trans*-eQTL by randomly and evenly dividing

samples into two groups (Group1 and Group2). *P* values were obtained based on the Pearson's correlation test.

Q6. The authors could do fine mapping (e.g. with eCAVIAR, FINEMAP, or SUSIER) of the QTL signal to strengthen their colocalization result (such as the ABCD4 example in Fig. 5i) of QTL and GWAS signal actually overlapping by identifying which exact SNPs that are potentially causal. I also wondering if SNPs with high fine mapping causal probability will enriched in sequence ontology in Fig. 4a (just like Fig. 3c-d in PMID 33986536).

AU: As suggested by the reviewer and to strengthen the colocalization of *ABCD4* eQTL and GWAS signal of back fat thickness (BFT), we employed SuSiE-inf (v1.2)³⁰ with default settings to fine-map the potential causal SNPs (defined by 95% credible sets) at the *ABCD4* locus for both eQTL and GWAS signals (**Response Figure 17a**). The fine-mapped lead SNP (*rs1114012229*) of the BFT GWAS was in a high linkage disequilibrium (LD, $r^2=0.85$) with the fine-mapped SNP (*rs1107405934*) for the *ABCD4* eQTL. These two SNPs reside in enhancer regions (EnhAME, EnhA, and EnhAHet) in both intestinal tissues and brain. In addition, *rs1107405934* was specifically and significantly associated with the expression of *ABCD4* in intestinal tissues and brain (**Response Figure 17b-c**). These results together indicate that *rs1114012229* and *rs1107405934* are promising candidate causal variants for BFT by regulating the expression of *ABCD4* in intestine and brain. We have added this information to the revised manuscript (**Lines 506-517**).

We further employed SuSiE-inf (v1.2)³⁰ to perform fine-mapping analysis for all five types of molQTL. We selected the molQTL with the highest posterior inclusion probability to represent the causality score of each molecular phenotype³¹. We then divided the molQTL into three causality groups (high: top 1/3, medium: 1/3-2/3, and low: bottom 1/3) based on the rank of causality scores. Furthermore, we conducted the enrichment analysis for each group of molQTL with sequence ontology and chromatin states. In general, we observed a higher enrichment of molQTL of higher causality scores for functional genomic features (**Response Figure 18a**). For instance, sQTL and lncQTL with high causality scores (high group) were more significantly enriched for spliced related region (splice donor) and non-coding region (NC transcript), respectively, compared to other two groups (**Response Figure 18a**). Among all the five types of molQTL, enQTL with high causality scores had the highest enrichment for enhancer-like chromatin states (EnhA, EnhAME, EnhAWk, EnhAHet, and EnhPois) (**Response Figure 18b**).

We have updated the enrichment results with the fine-mapped molQTL in the revised manuscript (Lines 506-517 & 1709-1714).

Response Figure 17. Fine-mapping of the *ABCD4* locus for the BFT GWAS. (a) The fine-mapped GWAS lead SNP (*rs111401222*) is in high linkage disequilibrium (LD) with the credible eQTL (*rs1107405934*) in small intestine. SNPs *rs111401222* and *rs1107405934* are located in enhancer regions in intestinal tissues and brain. (b-c) Association *P*-values of *rs1107405934* with each gene within its *cis*-region in small intestine (b) and brain (c). The inner panels of (b) and (c) are normalized expression levels of *ABCD4* in the three genotypes of *rs1107405934*. NS., ** and *** indicate two-sided Student's *t*-test significance at $P > 0.05$, $P < 0.01$ and $P < 0.001$, respectively.

Response Figure 18. Functional enrichment of fine-mapped molQTL. (a-b) Fold enrichment and standard deviation (SD) for fine-mapped molQTL in each causality bin for the intersection with sequence ontologies (a) and chromatin states (b), where molQTL were divided into three causality groups (high: top 1/3, medium: 1/3-2/3, and low: bottom 1/3) based on the rank of causality scores calculated using SuSiE-inf (v1.2).

Q7. The TWAS-server on the PigGTEEx-Portal does not work. The authors should also provide TWAS results and TWAS trained models (users can use these predictive models in their own studies) in the download webpage.

AU: We have updated the TWAS-server on the PigGTEEx-Portal and have provided all TWAS results and trained TWAS models in both the download webpage of PigGTEEx-Portal and the ScienceDB database (<https://www.scidb.cn/s/iUniQj> and <https://www.scidb.cn/s/zU7rIj>).

Q8. It's not convenient for users to explore QTLs without a genome browser. Please add a IGV Browser like the human GTEEx cohort (<https://gtexportal.org/home/browseEqtls>) using IGV.js (<https://github.com/igvteam/igv.js/>).

AU: We have added the IGV Browser plug to allow users to explore the molQTL summary statistics easily in the updated version of PigGTEEx-Portal (<http://piggtex.farmgtex.org>).

Q9. Did the authors consider batch effects of RNA-seq data? Please clarify or add.

AU: We have considered batch effects of RNA-Seq data. We inferred the potential confounders from gene expression data using PEER³² and found that the inferred PEER factors captured the known batch effects (e.g., project, sequencing module, and library layout) (**Revised Extended Data Figure 2h**). We included these inferred PEER factors and genotype PCs to account for both known and unknown confounders in the molQTL mapping (**Lines 1401-1419**). In addition, we reconducted the molQTL mapping by considering both known (e.g., sex and age) and inferred confounders (i.e., PEER factors and genotype PCs), and observed very similar results (**details see responses to Q2**). We have clarified this in the revised manuscript (**Lines 1480-1489**).

Q10. The author did a good job to compare the similarity of pigs with humans in gene expression and the genetic regulation. Could the author also compare it between pigs and cattles?

AU: As suggested by the reviewer, we compared the similarity of expression and regulation of 14,583 orthologous genes among pigs, cattle, and humans in the revised manuscript (**Lines 571-574, 1844-1848**). In general, the tissue-specific expression of genes (measured by TAU values) was more similar between pigs and humans than between cattle and humans (**Response Figure 19a-c**). Similarly, eQTL effects of orthologous genes in pigs were more highly correlated with those in humans than with those in cattle (**Response Figure 19d-f**). These findings provided molecular evidence that compared to cattle, pigs may serve as a better animal model for certain human biology research and disease treatment³³.

Response Figure 19. Similarity of gene expression and eQTL genetic regulator effects across pigs, cattle, and humans. (a-c) Pearson's correlation (r) of TAU values (measuring the tissue-specificity of gene expression) of 14,583 one-to-one orthologous genes between pigs, cattle, and humans. Dots represent genes and colors show the density of dots. (d-f) Spearman's correlation (ρ) of eQTL effect size $|\log_2(aFC)|$ in orthologous genes of 12 matched tissues between pigs, cattle, and humans.

Q11. Are there any trait-associated GWAS loci colocalized with two or more molecular QTLs (eQTL, sQTL, eeQTL, enQTL and lncQTL)? Please give some examples.

AU: Yes. In the previous version, **Extended Data Figure 10f** shows an example of colocalization among lncQTL, eQTL, and GWAS loci of loin muscle depth. We have demonstrated another two examples in the revised manuscript (**Response Figure 20, Revised Figure S16**). For instance, GWAS loci of loin muscle depth were significantly colocalized with eQTL, sQTL, and eeQTL of the *TRPT1* gene in muscle (**Response Figure 20a**). GWAS loci of backfat thickness were significantly colocalized with eQTL, sQTL, and eeQTL of the *PIGL* gene in liver (**Response Figure 20b**). We have updated these examples in the **Revised Figure S16** in the revised manuscript.

Response Figure 20. Examples of GWAS loci colocalized with multiple types of molQTL. (a) Significant SMR signals between GWAS loci of loin muscle depth and *cis*-eQTL, *cis*-sQTL, and *cis*-eeQTL of the *TRPT1* gene in muscle on chromosome 2. **(b)** Significant SMR signals between GWAS loci of backfat thickness and *cis*-eQTL, *cis*-sQTL, and *cis*-eeQTL of the *PIGL* gene in liver on chromosome 12.

Minor concerns:

1) In all figures (such as Fig. 1d), Frontal cortex and Hypothalamus should change to Brain-Frontal cortex and Brain-Hypothalamus (or other suitable names).

AU: Done (**Revised Figure 1d, 2a, 3a, 5h, 6a**)!

2) Colors in Fig. 2a are very similar, please change to make it clear.

AU: Done! We provided another plot below (Response Figure 21) to make it clear in the revised manuscript (Revised Figure S9b).

Response Figure 21. Proportion of eMolecules for five types of molQTL in each of 34 tissues in pigs. Tissues are ordered by increasing sample sizes.

3) In Fig. S1, what are the Q1, Q2, Q3 and Q4? Please clarify.

AU: Done. Q1-Q4 represent the four quarters of a year. We have updated this in the revised figure legend of Figure S1.

4) There is no Biosample ID or BioProject ID information for 509 new WGS data on Table S2.

AU: We have uploaded all these 509 newly generated WGS data to the public database and added the Biosample ID and BioProject ID in the revised Table S2.

5) In Fig. S10 and extended Fig. 5, please add figure legend. In addition, for Fig. S10, could you add another graph (like, boxplot with p-value) to clearly present the difference of spatial distribution across 34 tissues.

AU: Done (Revised Figure S10 and Extended Data Figure 5)! We have added a graph below (Response Figure 22) to clearly present the difference in spatial distribution of lead SNPs between eGenes and non-eGenes across 34 tissues (Revised Figure S10c).

Response Figure 22. Box plot showing the distances of the most significant associated variants to the transcriptional start site (TSS) of the target genes for eGenes and non-eGenes. *** represents $P < 0.001$ based on the two-sided Wilcoxon rank-sum test.

6) Fig.5C, Fig. S8 and Fig. S11 are confusing. Such as, Fig.5C, GWAS detected 524 GEAS loci linked to eGenes, the number drops to 1 when fastEnloc and S-prediXcan are included. However, the number increases to 225 when all 5 methods are included.

AU: Sorry for the confusion. We have more clearly described the Upset plots used in the revised manuscript (e.g., Revised Figure 5c, S8d, S12f and Extended Data Figure 10a), which were alternatives to the often-used Venn diagrams for overlapping many datasets. The bottom point-line combinations of Upset plot represent the intersections of different combinations of methods. For instance, 524 in the top barplot of Figure 5c means that 524 out of 1,507 GWAS loci were not linked to any eGenes by any of the four integrative methods. While, 1 means that there was only one GWAS locus that was linked to eGenes by S-PrediXcan and fastEnloc but not the other two methods. The number 225 means that there were 225 GWAS loci that were linked to eGenes by all four integrative methods (S-PrediXcan, S-MultiXcan, SMR, and fastEnloc)

simultaneously. We have updated the respective figure legends to make them easy to read in the revised manuscript (**Revised Figure 5c, S8d, S12f and Extended Data Figure 10a**).

7) The RNA-seq data includes pair-end or single-end and different platforms. the description of RNA-seq trimming with “TruSeq3-PE.fa:2:30:10” is incorrect. For single-end RNA-seq data, the parameter should be “TruSeq2-SE”.

AU: Thanks for pointing out this! Corrected! (**Lines 1189-1191**)

8) For extended Fig.8, authors found the variant of cell type composition across bulk tissue samples by a cell-type deconvolution analysis. It’s better to draw all the different type of tissue on one diagram to clearly observe cell of origin. In addition, authors should further explain the difference of cell contribution in different tissues.

AU: As suggested by the reviewer, we provided the plot below (**Response Figure 23**) to show the enrichment percentage of cell types across seven tested tissues. As expected, we observed that lymphocytes (i.e., B cells and CD4⁺ αβ T cells) are the major cells in spleen³⁴, while Astro cells and alveolar epithelial type 2 cells are the major cells in brain and lung, respectively. We have added this information to the revised manuscript (**Line 361-363**).

Response Figure 23. Distribution of enrichment scores (percentage) of major cell types across all samples in seven tested tissues.

9) In this manuscript, authors made a lot of analysis and obtained a lot of conclusions. But in some parts, there is no literature or further verification to support conclusions. For example, the gene *EMG1* (Fig. 2i), *ODF2L* (Fig. 3h) and so on, why did authors select these genes for illustration? Are they the most significant? Or have they been reported?

AU: For most gene examples, we have chosen them mainly based on the statistical significance and their biological function. Where possible, we have provided literature support for gene examples in the revised manuscript (**Lines 409-411, 339-340**). For instance, *ODF2L*, which showed an opposite direction of eQTL effects between blood and testis, is involved in negative regulation of cilium assembly and spermatogenesis³⁵. In the breed-sharing eGene analysis, we chose three genes (i.e., *EMG1*, *NMNATI*, and *COMMD10*) as examples mainly based on their statistical significance, where *EMG1* in the revised **Figure 2i** is the most significant one. *NMNATI* is involved in nucleotide biosynthetic process³⁶ and *COMMD10* enables protein binding³⁷.

Reviewer #3

Remarks to the Author:

This manuscript presents the main results of a massive collective effort that is the pilot phase of the Pig Genotype-Tissue Expression atlas (PigGTE_x). PigGTE_x provides a very useful resource for pig genomics research, in particular on gene expression and regulatory effects that underlie complex traits of economic and social importance. The manuscript is very well-written, clear, well-structured, and with suitable references. Results are succinctly described and deeply detailed in the corresponding figures and tables in an appropriate fashion. I only found some small typos here and there, especially in the supplementary files, of which I flag a few below (there may be others). I did not identify any flaws in methodology or in the interpretation of the results, so I do not have any major comments and recommend this manuscript for publication.

AU: Thank you very much for the very positive comments! We have carefully modified the manuscript according to the detailed comments below.

Line 181: minimizing

AU: Done (**Line 181**)!

Line 182: Is ‘environmental impacts’ broad enough to refer to all challenges to reach sustainable food and agriculture production?

AU: No, it is not broad enough. We thus rephrased this sentence in the revised manuscript (**Line 181-182**) as below:

To sustain food and agriculture production while minimizing associated negative environmental impacts and maximizing the genetic potential of animals, it is crucial to identify molecular mechanisms that underpin complex traits of economic importance in farm animals to enable biology-driven breeding biotechnologies.

L195: comma in ‘PigGTE_x, which’

AU: Done (**Line 194**)!

L231: ‘similar to the PGRP’ except for Duroc x Asian. Maybe worth adding a short sentence stating why that is the case, given that otherwise they are highlighted in colour.

AU: We have highlighted Duroc x Asian in the PGRP panel in the **Revised Figure 1c**.

L579-580: ‘cis’ and ‘trans’ in italics everywhere?

AU: We carefully corrected them throughout the revised manuscript.

Fig S1a: The exponential fit deviates a bit from observations for the last years. Is number of samples slowing down or are they just not published yet? I do not think the exponential fit and R² are needed, so they could be removed.

AU: Done! We have removed the fitting line in the revised manuscript (**Revised Figure S1a**).

Fig S3: blank space in ‘depths (e)’

AU: Corrected (**Revised Fig. S3**).

Fig S3h: chromosome is missing an ‘e’

AU: Corrected (**Revised Fig. S3h**).

Fig S4c: capital G in ‘CpG’

AU: Corrected (**Revised Fig. S4c**).

Table S2 and others: Some breeds are not defined in a uniform way. For example: Does ‘X bred’ mean ‘crossbred’? If so, see the later. Another example: ‘A x (B x C)’ is also referred to as ‘ $\frac{1}{2}$ A + $\frac{1}{4}$ B + $\frac{1}{4}$ C’. Please use the same criteria for all.

AU: Done (**Revised Table S2, Line 1526**)!

Table S8 (and possibly others): Some typos in header.

AU: Corrected (**Revised Table S8**).

References

1. Liu, S. *et al.* A multi-tissue atlas of regulatory variants in cattle. *Nat. Genet.* (2022) doi:10.1038/s41588-022-01153-5.

2. Jehl, F. *et al.* RNA-Seq Data for Reliable SNP Detection and Genotype Calling: Interest for Coding Variant Characterization and Cis-Regulation Analysis by Allele-Specific Expression in Livestock Species. *Front. Genet.* **12**, 1–17 (2021).
3. Deelen, P. *et al.* Calling genotypes from public RNA-sequencing data enables identification of genetic variants that affect gene-expression levels. *Genome Med.* **7**, 1–13 (2015).
4. Cingolani, P. *et al.* A program for annotating and predicting the effects of single nucleotide polymorphisms, SnpEff: SNPs in the genome of *Drosophila melanogaster* strain w1118; iso-2; iso-3. *Fly (Austin)*. **6**, 80–92 (2012).
5. Yang, J. *et al.* Genomic inflation factors under polygenic inheritance. *Eur. J. Hum. Genet.* **19**, 807–812 (2011).
6. Aguet, F. *et al.* The GTEx Consortium atlas of genetic regulatory effects across human tissues. *Science (80-.)*. **369**, 1318–1330 (2020).
7. Aguet, F. *et al.* Molecular quantitative trait loci. **0123456789**, (2023).
8. Taylor-Weiner, A. *et al.* Scaling computational genomics to millions of individuals with GPUs. *Genome Biol.* **20**, 228 (2019).
9. Munro, D. *et al.* The regulatory landscape of multiple brain regions in outbred heterogeneous stock rats. *Nucleic Acids Res.* **50**, 10882–10895 (2022).
10. Flynn, E. & Lappalainen, T. Functional Characterization of Genetic Variant Effects on Expression. *Annu. Rev. Biomed. Data Sci.* **5**, 119–139 (2022).
11. Speed, D., Hemani, G., Johnson, M. R. & Balding, D. J. Improved heritability estimation from genome-wide SNPs. *Am. J. Hum. Genet.* **91**, 1011–1021 (2012).
12. Yao, D. W., O’Connor, L. J., Price, A. L. & Gusev, A. Quantifying genetic effects on disease mediated by assayed gene expression levels. *Nat. Genet.* **52**, 626–633 (2020).
13. Võsa, U. *et al.* Large-scale cis- and trans-eQTL analyses identify thousands of genetic loci and polygenic scores that regulate blood gene expression. *Nat. Genet.* **53**, 1300–1310 (2021).
14. Bhattacharya, A. *et al.* Best practices for multi-ancestry, meta-analytic transcriptome-wide association studies: Lessons from the Global Biobank Meta-analysis Initiative. *Cell Genomics* **2**, 100180 (2022).
15. Watanabe, K. *et al.* A global overview of pleiotropy and genetic architecture in complex traits. *Nat. Genet.* **51**, 1339–1348 (2019).
16. Bulik-Sullivan, B. *et al.* LD score regression distinguishes confounding from polygenicity in genome-wide association studies. *Nat. Genet.* **47**, 291–295 (2015).
17. Ren, B. Enhancers make non-coding RNA. *Nature* **465**, 173–174 (2010).
18. de Santa, F. *et al.* A large fraction of extragenic RNA Pol II transcription sites overlap enhancers. *PLoS Biol.* **8**, (2010).
19. Mikhaylichenko, O. *et al.* The degree of enhancer or promoter activity is reflected by the levels and directionality of eRNA transcription. *Genes Dev.* **32**, 42–57 (2018).
20. Kim, T. K. *et al.* Widespread transcription at neuronal activity-regulated enhancers. *Nature* **465**, 182–187 (2010).
21. Chen, H. *et al.* A Pan-Cancer Analysis of Enhancer Expression in Nearly 9000 Patient Samples. *Cell* **173**, 386–399.e12 (2018).
22. Murakawa, Y. *et al.* Enhanced Identification of Transcriptional Enhancers Provides Mechanistic Insights into Diseases. *Trends Genet.* **32**, 76–88 (2016).
23. Hah, N., Murakami, S., Nagari, A., Danko, C. G. & Lee Kraus, W. Enhancer transcripts mark active estrogen receptor binding sites. *Genome Res.* **23**, 1210–1223 (2013).
24. Zhang, Z. *et al.* HeRA: an atlas of enhancer RNAs across human tissues. *Nucleic Acids Res.* **49**, D932–D938 (2021).
25. Urbut, S. M., Wang, G., Carbonetto, P. & Stephens, M. Flexible statistical methods for estimating and testing effects in genomic studies with multiple conditions. *Nat. Genet.* **51**, 187–195 (2019).

26. de Goede, O. M. *et al.* Population-scale tissue transcriptomics maps long non-coding RNAs to complex disease. *Cell* **184**, 2633–2648.e19 (2021).
27. Bridges, M. C., Daulagala, A. C. & Kourtidis, A. LNCcation: lncRNA localization and function. *J. Cell Biol.* **220**, 1–17 (2021).
28. Saha, A. & Battle, A. False positives in trans-eQTL and co-expression analyses arising from RNA-sequencing alignment errors. *F1000Research* **7**, 1–27 (2019).
29. Lee, B. T. *et al.* The UCSC Genome Browser database: 2022 update. *Nucleic Acids Res.* **50**, D1115–D1122 (2022).
30. Cui, R. *et al.* Improving fine-mapping by modeling infinitesimal effects. *bioRxiv* 2022.10.21.513123 (2022).
31. Li, L. *et al.* An atlas of alternative polyadenylation quantitative trait loci contributing to complex trait and disease heritability. *Nat. Genet.* **53**, 994–1005 (2021).
32. Stegle, O., Parts, L., Piipari, M., Winn, J. & Durbin, R. Using probabilistic estimation of expression residuals (PEER) to obtain increased power and interpretability of gene expression analyses. *Nat. Protoc.* **7**, 500–507 (2012).
33. Lunney, J. K. *et al.* Importance of the pig as a human biomedical model. *Sci. Transl. Med.* **13**, (2021).
34. Pillai, S. & Cariappa, A. The follicular versus marginal zone B lymphocyte cell fate decision. *Nat. Rev. Immunol.* **9**, 767–777 (2009).
35. de Saram, P., Iqbal, A., Murdoch, J. N. & Wilkinson, C. J. BCAP is a centriolar satellite protein and inhibitor of ciliogenesis. *J. Cell Sci.* **130**, 3360–3373 (2017).
36. Berger, F., Lau, C., Dahlmann, M. & Ziegler, M. Subcellular compartmentation and differential catalytic properties of the three human nicotinamide mononucleotide adenylyltransferase isoforms. *J. Biol. Chem.* **280**, 36334–41 (2005).
37. Burstein, E. *et al.* COMMD proteins, a novel family of structural and functional homologs of MURR1. *J. Biol. Chem.* **280**, 22222–32 (2005).

Decision Letter, first revision:

24th Jul 2023

Dear Lingzhao,

Thank you for submitting your revised manuscript "A compendium of genetic regulatory effects across pig tissues" (NG-A61417R). It has now been seen by the original referees and their comments are below. The reviewers find that the paper has improved in revision, and therefore we'll be happy in principle to publish it in Nature Genetics, pending minor revisions to satisfy the referees' final requests and to comply with our editorial and formatting guidelines.

Sincerely,

Michael Fletcher, PhD
Senior Editor, Nature Genetics

ORCID: 0000-0003-1589-7087

Reviewer #1 (Remarks to the Author):

The authors have addressed my concerns adequately. I have no more comments.

Reviewer #2 (Remarks to the Author):

the authors have addressed all my concerns

Reviewed by Wei LI and Ya Allen Cui.

Reviewer #3 (Remarks to the Author):

The authors have addressed all my previous comments and improved on the analyses flagged by the other reviewers. I have nothing else to add.

Final Decision Letter:

In reply please quote: NG-A61417R1 Fang

13th Oct 2023

Dear Lingzhao,

I am delighted to say that your manuscript "A compendium of genetic regulatory effects across pig tissues" has been accepted for publication in an upcoming issue of Nature Genetics.

Your paper will be published online after we receive your corrections and will appear in print in the next available issue. You can find out your date of online publication by contacting the Nature Press Office (press@nature.com) after sending your e-proof corrections. Now is the time to inform your Public Relations or Press Office about your paper, as they might be interested in promoting its publication. This will allow them time to prepare an accurate and satisfactory press release. Include your manuscript tracking number (NG-A61417R1) and the name of the journal, which they will need when they contact our Press Office.

Please note that *Nature Genetics* is a Transformative Journal (TJ). Authors may publish their research with us through the traditional subscription access route or make their paper immediately open access through payment of an article-processing charge (APC). Authors will not be required to make a final decision about access to their article until it has been accepted. [Find out more about Transformative Journals](https://www.springernature.com/gp/open-research/transformative-journals)

Authors may need to take specific actions to achieve [compliance with funder and institutional open access mandates](https://www.springernature.com/gp/open-research/funding/policy-compliance-faqs). If your research is supported by a funder that requires immediate open access (e.g. according to [Plan S principles](https://www.springernature.com/gp/open-research/plan-s-compliance)) then you should select the gold OA route, and we will direct you to the compliant route where possible. For authors selecting the subscription publication route, the journal's standard licensing terms will need to be accepted, including [self-archiving and license to publish](https://www.nature.com/nature-portfolio/editorial-policies/self-archiving-and-license-to-publish). Those licensing terms will supersede any other terms that the author or any third party may assert apply to any version of the manuscript.

If you have not already done so, we invite you to upload the step-by-step protocols used in this manuscript to the Protocols Exchange, part of our on-line web resource, natureprotocols.com. If you complete the upload by the time you receive your manuscript proofs, we can insert links in your article that lead directly to the protocol details. Your protocol will be made freely available upon publication of your paper. By participating in natureprotocols.com, you are enabling researchers to more readily reproduce or adapt the methodology you use. [Natureprotocols.com](https://natureprotocols.com) is fully searchable, providing your protocols and paper with increased utility and visibility. Please submit your protocol to <https://protocolexchange.researchsquare.com/>. After entering your [nature.com](https://www.nature.com) username and password you will need to enter your manuscript number (NG-A61417R1). Further information can be found at <https://www.nature.com/nature-portfolio/editorial-policies/reporting-standards#protocols>

Sincerely,

Michael Fletcher, PhD
Senior Editor, Nature Genetics

ORCID: 0000-0003-1589-7087